# Tensor Programs VI: Feature Learning in Infinite Depth Neural Networks

**Greg Yang**[*]
xAI

**Dingli Yu**[*]
Princeton Language
and Intelligence,
Princeton University

**Chen Zhu**
Nvidia

**Soufiane Hayou**[†]
Simons Institute,
UC Berkeley

## Abstract

Empirical studies have consistently demonstrated that increasing the size of neural networks often yields superior performance in practical applications. However, there is a lack of consensus regarding the appropriate scaling strategy, particularly when it comes to increasing the *depth* of neural networks. In practice, excessively large depths can lead to model performance degradation. In this paper, we introduce Depth-$\mu$P, a principled approach for depth scaling, allowing for the training of arbitrarily deep networks while maximizing feature learning and feature diversity among nearby layers. Our method involves dividing the contribution of each residual block and the parameter update by the square root of the depth. Through the use of Tensor Programs, we rigorously establish the existence of a limit for infinitely deep neural networks under the proposed scaling scheme. This scaling strategy ensures more stable training for deep neural networks and guarantees the transferability of hyperparameters from shallow to deep models. To substantiate the efficacy of our scaling method, we conduct empirical validation on neural networks with depths up to $2^{10}$.

## 1 Introduction

Deep neural networks have showcased remarkable performance across a broad range of tasks, including image classification, game playing exemplified by AlphaGo (Silver et al., 2016), and natural language processing demonstrated by GPT-4 (OpenAI, 2023). A prevailing trend in developing these networks is to increase their size and complexity, with empirical evidence indicating that using the same computation resources, models with more parameters tend to exhibit better performance. There are two ways to increase any network size: *width* and *depth*. The properties of the width (given a fixed depth) have been extensively studied in the literature: recent work by Yang et al. (2022) identified the *Maximal Update Parametrization* ($\mu$P) that guarantees maximal feature learning in the infinite width limit.[1] Another benefit of $\mu$P is hyperparameter transfer which enables hyperparameter tuning on smaller models; the optimal hyperparameter choice for the smaller model remains optimal for larger models (i.e., models with larger width). However, despite the achievements of large-scale deep models and the theoretical understanding of scaling width, increasing the depth of neural networks still has both practical limitations and theoretical difficulties. In practice, increasing depth beyond some level often results in performance degradation and/or significant shifts in the optimal hyperparameters. In theory, unlike increasing width, increasing depth introduces new parameters that significantly change the training dynamics. In this paper, we aim to solve this problem by extending $\mu$P to include depth scaling. We call the depth scaling Depth-$\mu$P.

The issue of depth scaling has persisted over time. A decade ago, deep neural networks experienced significant degradation problems — having more than a few dozen layers would increase the training error instead of improving the model's performance. This was partly due to the vanishing

---

[*]Equal contribution.

[†]Work partially done at the National University of Singapore.

[1]Here maximal feature learning refers to $\Theta(1)$ change in features in the infinite width limit. This should be contrasted with the lazy training regime where the change in features is of order $\Theta(n^{-1/2})$.

or exploding gradient problem that affects the efficient propagation of information through the network. The introduction of residual networks (ResNet) (He et al., 2016a;b; Srivastava et al., 2015) has partially resolved this issue, allowing for the training of deeper networks with improved performance. ResNet is constructed by layering *residual blocks*, which are composed of a series of convolutional layers and then an element-wise addition with the input. This element-wise addition (commonly referred to as *skip connection*) is a significant innovation of ResNet and remains an important ingredient in modern architectures including Transformers (Vaswani et al., 2017).

Specifically, in a residual architecture, the $l$-th residual block is formulated as

$$x^l = x^{l-1} + g^l(x^{l-1}; W^l),$$

where $x^{l-1}$ is the input, $x^l$ is the output, $W^l$ are the parameters of the block, and $g^l$ (often called the *residual branch*) is a mapping that defines the layer (e.g. a stack of convolutions in ResNet, or SelfAttention and MLP in a Transformer). In this work, we focus on the case where $g^l$ is a biasless perceptron with (or without) activation.

The stacking of many residual blocks causes an obvious issue even at the initialization — the norm of $x^l$ grows with $l$, so the last layer features do not have a stable norm when increasing the depth. Intuitively, one can stabilize these features by scaling the residual branches with a depth-dependent constant. However, scaling the residual branches with arbitrarily small constants might result in no feature learning in the large depth limit since the gradients will also be multiplied with the scaling factor. In this paper, we propose Depth-$\mu$P, a principled approach to scaling up the depth of residual networks, enabling the training of arbitrarily deep networks while achieving *feature learning* and maximizing *feature diversity* among nearby layers. Our framework extends the previous results on $\mu$P which deals with optimal width scaling (Yang et al., 2022). It completes the width scaling and hence provides a full width and depth scaling recipe that guarantees maximal feature learning and hyperparameter transfer across width and depth. Depth-$\mu$P contains the following modifications to the standard practice:

1. There is a multiplier for each residual branch before adding to its input, which is inversely proportional to the square root of $L$ (where $L$ is the depth). Formally, with a constant $a$ independent from $L$,

$$x^l = x^{l-1} + \frac{a}{\sqrt{L}} \cdot g^l(x^{l-1}; W^l). \quad (1)$$

2. We set the learning rate of $W^l$ so that the update of $W^l$ during training is proportional to $1/\sqrt{L}$. We derive different learning rate schemes for different optimization algorithms based on this principle. For Adam, because it is scale-invariant to the gradient, the learning rate of $W^l$ is set to be $\eta/\sqrt{L}$. On the other hand, the learning rate of $W^l$ for SGD is set as a constant $\eta$ because the gradient of $W^l$ is already of size $1/\sqrt{L}$ due to the multiplier.

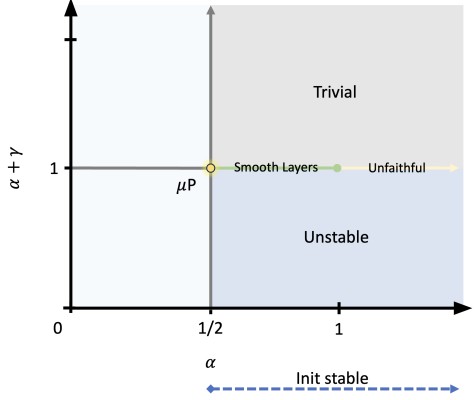

Figure 1: Behaviors of scaling strategies with a branch multiplier $L^{-\alpha}$ and parameter update size proportional to $L^{-\gamma}$.

Figure 1 compares Depth-$\mu$P with general branch multiplier scaling $L^{-\alpha}$ and update size scaling $L^{-\gamma}$ (where Depth-$\mu$P has $\alpha = \gamma = \frac{1}{2}$; see Section 4 for more details), showing the uniqueness of Depth-$\mu$P. Specifically, limiting our analysis to block depth 1 (i.e., $g^l$ is a biasless perceptron, $W^l$ is a single matrix)[2], we show that Depth-$\mu$P leads to the following properties: **a)** At the initialization, each one of the $L$ residual blocks contributes $\Theta(1/\sqrt{L})$ to the main branch. These $L$ contributions are independent of each other, so the sum of them is of size $\Theta(1)$; **b)** During training, the contribution of the update of each residual block is $\Theta(1/L)$ due to the combining effect of the learning rate and multiplier. The contributions of the updates are highly correlated, so they sum up to $\Theta(1)$; **c)** More

---

[2]In Appendix K, we include a thorough discussion on the scaling strategy for block depth 2 and above. We find limitations in all possible infinite-depth limits of such parametrizations, which we illustrate on simple networks as well as Megatron transformer trained on Common Crawl (see Appendix L.3).

importantly, we classify all depth limits and show Depth-$\mu$P yields the optimal limit. This implies that the optimal hyperparameters of the networks in Depth-$\mu$P are approximately invariant wrt depth. With Depth-$\mu$P, we successfully train networks comprising thousands of residual blocks, while also showcasing the transferability of hyperparameters across depth.

## 2   RELATED WORKS

Recently, $\mu$P (Maximal Update Parametrization), introduced in Yang et al. (2022), has emerged as a promising approach for maximizing feature learning while simultaneously preventing feature explosion as the network width increases, given a fixed depth. Notably, $\mu$P facilitates hyperparameter transfer across varying network widths. This means that instead of tuning hyperparameters directly on large models, one can optimize them on smaller models and utilize the same set of hyperparameters for larger models.

While increasing the width can lead to improved performance, increasing the depth of the network also yields significant performance gains, and most state-of-the-art models use deep architectures. To address the challenges associated with training deep networks, several studies have proposed scaling the network blocks using a depth-dependent scaler to ensure stability of features and gradients at initialization (Hanin & Rolnick, 2018; Zhang et al., 2019; Hayou et al., 2021; Noci et al., 2022; Hayou, 2023; Hayou & Yang, 2023; Noci et al., 2023). For Multi-Layer Perceptrons (MLPs) (no skip connections), Jelassi et al. (2023) showed that a learning rate scaling of $depth^{-3/2}$ guarantees stability after the initial gradient step. However, it remains unclear how the learning rate should be adjusted beyond the first step, and this scaling is not suitable for architectures with residual connections. We refer the reader to Appendix A for a more comprehensive literature review.

## 3   WARM-UP: DEEP LINEAR RESIDUAL NETWORKS

Let us begin with a simple example that provides the necessary intuition underpinning our depth scaling strategy. Given a depth $L$, width $n$, consider a linear residual network of the form

$$x^l = x^{l-1} + \frac{1}{\sqrt{L}} W^l x^{l-1}, \qquad \text{for } l = 1, \dots, L,$$

where $x^0 = U\xi$ is the input layer, $f(\xi) = V^\top x^L$ is the output layer, $W^l \in \mathbb{R}^{n \times n}$ are the weight matrices, and $U, V$ are input and output weight matrices assumed to be fixed during training.

### 3.1   OPTIMAL SCALING OF THE LEARNING RATE

To simplify the analysis, we consider gradient updates based on a single datapoint. The first gradient step is given by $W_1^l = W_0^l - \eta G_0^l$, where $\eta$ is the learning rate, and $G_0^l$ is a matrix with update directions. For instance, we have the following expressions for $G_0^l$ with SGD and Adam:

- SGD: $G_0^l = \frac{1}{\sqrt{L}} dx^l \otimes x^{l-1}$, where $dx^l \stackrel{\text{def}}{=} \frac{\partial \ell}{\partial x^l}$ for some loss function $\ell$.

- Adam: $G_0^l = \text{sign}\left(\frac{1}{\sqrt{L}} dx^l \otimes x^{l-1}\right)$, where $\otimes$ is the Kronecker product.[3]

In both cases, $dx^l$ and $x^{l-1}$ are computed for a single training datapoint $\xi_0$. The last layer features $x^L$ (for some input $\xi$) are given by $x^L = \prod_{l=1}^L \left(I + \frac{1}{\sqrt{L}} W^l\right) x^0$.[4] We use the subscript $t$ to refer to training step. After the first gradient step, we have the following

$$x_1^L = \prod_{l=1}^L \left(I + \frac{1}{\sqrt{L}} W_1^l\right) x^0 = x_0^L - \frac{\eta}{\sqrt{L}} A_L + \mathcal{O}(\eta^2), \tag{2}$$

where $A_L = \sum_{l=1}^L \left[\prod_{k>l} \left(I + \frac{1}{\sqrt{L}} W_0^k\right)\right] G_0^l \left[\prod_{k<l} \left(I + \frac{1}{\sqrt{L}} W_0^k\right)\right] x^0$. We argue that $A_L$ behaves as $\Theta(L)$ (in second norm). This is the due to the $1/\sqrt{L}$ scaling factor. To see this, we

---

[3]The first step of Adam can be viewed as the SignSGD update. Our analysis for Adam is valid for any training algorithm that gives $\Theta(1)$ gradients.

[4]To avoid any confusion, here we define the matrix product by $\prod_{l=1}^L A_l = A_L \times A_{L-1} \cdots \times A_1$.

further simplify the analysis by considering the case $d_{in} = n = d_{out} = 1$ (single neuron per layer) and the squared loss.

**Scaling for SGD.** With SGD, we have that $G_0^l = \frac{1}{\sqrt{L}} \prod_{k \neq l} \left(1 + \frac{1}{\sqrt{L}} W_0^k\right) x_0 dx^L$, where $dx^L = (V x^L - y(\xi_0))$ and $y(\xi_0)$ is the target output. Therefore, it is easy to see that

$$\mathbb{E} A_l^2 = \frac{1}{L} \mathbb{E} \left( \sum_{l=1}^{L} \prod_{k \neq l} \left(1 + \frac{1}{\sqrt{L}} W_0^k\right)^2 dx^L x_0^2 \right)^2 = \Theta \left( \frac{1}{L} L^2 \right) = \Theta(L),$$

where we have used the fact that $\mathbb{E} \left(1 + \frac{1}{\sqrt{L}} W_0^k\right)^{2p} = 1 + \Theta(L^{-1})$, for any positive integer $p$. Hence, the magnitude of the first order term in eq. (2) is given by $\mathbb{E} \left[ \left(\frac{\eta}{\sqrt{L}} A_l\right)^2 \right] = \Theta(\eta^2)$, which shows that the update is stable in depth as long as $\eta = \Theta(1)$ in depth. More precisely, this is the maximal choice of learning rate that does not lead to exploding features as depth increases.

**Scaling for Adam.** With Adam, we have $G_0^l = \pm 1$, and therefore we obtain

$$\mathbb{E} A_l^2 = \mathbb{E} \left( \sum_{l=1}^{L} \prod_{k \neq l} \left(1 + \frac{1}{\sqrt{L}} W_0^k\right) x_0 \right)^2 = \Theta \left( L^2 \right),$$

where we have used the same arguments as before. In this case, the first order term in eq. (2) is given by $\mathbb{E} \left[ \left(\frac{\eta}{\sqrt{L}} A_l\right)^2 \right] = \Theta(\eta^2 L^{-1})$, and therefore, the maximal learning rate that one can choose without exploding the features is given by $\eta = \Theta(L^{-1/2})$.

*Summary*: By ensuring that parameter update is $\Theta(1/\sqrt{L})$, the features remain stable while feature update is $\Theta(1)$. This $\Theta(1)$ update is due to the accumulation of $\Theta(1/L)$ correlated terms across depth. We refer the reader to Appendix B for a more in-depth analysis of this simple model.

## 3.2 SGD TRAINING DYNAMICS BEYOND ONE TRAINING STEP AND $n = 1$

The intuitive example above assumes there is only one single neuron per layer ($n = 1$). However, rigorous extension to the general finite width setting ($n > 1$) is hard. This can be solved by introducing Tensor Programs framework (Yang & Littwin, 2023), which allows us to study the infinite-width network and be worry-free from noise caused by finite width. Tensor Programs also enable analysis of training dynamics beyond the first training step. Particularly, we rigorously derive the training dynamics of SGD for the linear residual network when the width and the depth sequentially go to infinity. Our analysis of the linear case reveals our main technical tools with minimal notations and prerequisite knowledge, so it is more friendly to readers who are interested in the technical details. Below we provide our technical road map. The full version of the analysis can be found in Appendix C.

1. We first take the width of the network to infinity by the Tensor Program framework. As a result, instead of tracking vectors and matrices along the training trajectory, we track random variables that correspond to the vectors, that is, for a vector $x \in \mathbb{R}^n$ that appears in the computation of the training, the coordinates of $x$ can be viewed as iid copies of random variable $\|x\rangle$ (called a *ket*) when $n \to \infty$.

2. Since the network is linear, every random variable can be written as a linear combination of a set of zero-mean "base" random variables by the Master Theorem of Tensor Programs (Yang & Littwin, 2023). Then we can track the random variables by analyzing the coefficients of their corresponding linear combinations, and the covariance between the "base" random variables.

3. We aggregate the coefficients of all random variables and represent them by a six-dimensional tensor $\mathbf{\Gamma}$, where two of the dimensions have shape $L$ because the number of random variables and the number of "base" random variables scale linearly with $L$. The next step is to find if $\mathbf{\Gamma}$ converges when $L$ grows. However, any convergence directly based on $\mathbf{\Gamma}$ is impossible because the shape of $\mathbf{\Gamma}$ is different when $L$ grows. We thus convert $\mathbf{\Gamma}$ to a set of functions $\Gamma$ whose input domain is $[0, 1] \times (0, 1]$, where the two inputs to $\Gamma$ corresponds to the fractional index of the two

dimensions of $\mathbf{\Gamma}$ of shape $L$. Similarly, the covariance between the "base" random variables can represented by another set of functions $C$ whose input domain is $(0, 1]$. Finally, we claim that the functions $\Gamma$ and $C$ converge when $L \to \infty$, and identify their limits as the solution of a set of functional integrals in Proposition C.4.

In Appendix L.1, we conduct a thorough empirical verification of our theory in the linear case. The experiments clearly show the convergence of deep linear residual networks under Depth-$\mu$P.

## 4 PRELIMINARIES FOR THE GENERAL CASE

**Notation** Let $L$ be the depth of the network, i.e., the number of residual blocks, and $n$ be the width of the network, i.e. the dimension of all hidden representations $x^0, \ldots, x^L$. Let $\xi \in \mathbb{R}^{d_{\text{in}}}$ be the input of the network, $U \in \mathbb{R}^{n \times d_{\text{in}}}$ be the input layer, and $V \in \mathbb{R}^{n \times e}$ be the output layer, so that $x^0 = U\xi$ and the model output w.r.t. $\xi$ is $f(\xi) \triangleq V^\top x^L$. Let $\ell$ be the loss function absorbing the label, and $dx^l$ be the gradient of $x^l$ w.r.t. the loss. We denote variables at $t$-th training step by adding $t$ as a subscript, e.g., the input at step $t$ is $\xi_t$[5], the hidden representation of $l$-th layer at step $t$ is $x_t^l$, and the model output at step $t$ is $f_t$. Let $T$ be the number of training steps. Let $I$ be the identity matrix, and $\mathbf{1}$ be the full one vector. For $m \in \mathbb{N}^+$, let $[m] = \{1, \ldots, m\}$.

### 4.1 UNIFIED SCALING FOR SGD AND ADAM

We extend the definition of entrywise update (Yang & Littwin (2023)) for depth scaling, allowing us to study the unified depth scaling for SGD, Adam, and other optimization algorithms that perform only entrywise operations.

**Definition 4.1.** A gradient-based update of parameter $w$ with both width and depth scaling is defined by a set of functions $\boldsymbol{Q} = \{Q_t : \mathbb{R}^{t+1} \to \mathbb{R}\}_{t \geq 0}$, and $c, d, \delta, \gamma, \eta$. The update at time $t$ of the optimization is
$$w \leftarrow w - \eta n^{-c} L^{-\gamma} Q_t(n^d L^\delta g_0, \ldots, n^d L^\delta g_t),$$
where $g_s, s = 0, \ldots, t$, are the gradients of $w$ at time $s$.

For SGD, $Q_t(n^d L^\delta g_0, \ldots, n^d L^\delta g_t) = n^d L^\delta g_t$, and the "true" learning rate is $\eta n^{-c+d} L^{-\gamma+\delta}$. For Adam, $Q_t(n^d L^\delta g_0, \ldots, n^d L^\delta g_t) = \dfrac{\frac{1-\beta_1}{1-\beta_1^{t+1}} \sum_{s=0}^t \beta_1^{t-s} n^d L^\delta g_s}{\sqrt{\frac{1-\beta_2}{1-\beta_2^{t+1}} \sum_{s=0}^t \beta_2^{t-s} (n^d L^\delta g_s)^2 + \epsilon}}$, and the "true" learning rate is just $\eta n^{-c} L^{-\gamma}$.

The purpose of multiplying the gradients $n^d L^\delta$ before $Q_t$ is to make sure the inputs to $Q_t$ are $\Theta(1)$ w.r.t. $n$ and $L$[6]; otherwise, the update might be trivial when $n$ and $L$ become large. For example, if gradients are $o(1)$ entrywise, then, in Adam, directly feeding gradients to $Q_t$ will always give an output of 0 because of the constant $\epsilon > 0$. In this paper, we will only consider $d, \delta$ such that $n^d L^\delta g$ is $\Theta(1)$.[7] As a result, the output of $Q_t$ is also $\Theta(1)$ in general. Therefore, $n^{-c} L^{-\gamma}$ decides the scale of the update and should be our focus. We call $\eta n^{-c} L^{-\gamma}$ the *effective learning rate*.

### 4.2 SETUP

We consider an $L$-hidden-layer residual network with biasless perceptron blocks:
$$\forall l \in [L], \quad h^l = W^l x^{l-1}, \quad x^l = x^{l-1} + L^{-\alpha} \text{MS}(\phi(h^l)),$$
where $x^0 = U\xi$, the network output $f = V^\top x^L$, $\phi$ is the nonlinearity/activation function and MS refers to Mean Substraction and is given by $\text{MS}(x) = x - \langle x, \mathbf{1} \rangle / n = Gx$ with $G = I - \mathbf{1}\mathbf{1}^\top / n$. We follow $\mu$P (Yang, 2020b) for the *widthwise* scaling, i.e., the initialization of $U, V, W^l$ are i.i.d. zero-mean Gaussian with variance $1, n^{-2}, n^{-1}$ respectively, and the $c$ in the effective learning rate $\eta n^{-c} L^{-\gamma}$ for $U, V$ and $W^l$ are $0, 1, 1$ respectively, i.e., the learning rate of $W^l$ is $\eta n^{-1} L^{-\gamma}$.

In sum, the $(\alpha, \gamma)$ pair decides the *depthwise* scaling of our whole setup.

---

[5]Here, the input is used to perform one gradient step at training step $t$. We will see later that our claims should in principle hold for batched versions of the training algorithm.

[6]It is called faithfulness in Yang & Littwin (2023).

[7]Note $c, d, \delta, \gamma, \eta$ in Definition 4.1 can be different for parameters, so it is possible to make every parameter to satisfy the condition. In this paper, we focus on matrices $W^l$, which share the same set of $c, d, \delta, \gamma, \eta$.

**Maximal update parametrization ($\mu$P)** (Yang, 2020b) considers the change of initialization and learning rate of each weight matrix in the network when *width* scales up.[8] It provides a unique initialization and learning rate of each weight matrix as a function of width $n$ that makes the update of each weight matrix maximal (up to a constant factor). The benefit of $\mu$P is not only the theoretical guarantee but also the hyperparameter stability when scaling up the width (Yang & Hu, 2021).

**Mean Substraction** (MS). The use of mean substraction is key in our analysis. First of all, it acts in a similar fashion to LayerNorm without the normalization part. We believe that mean substraction in LayerNorm plays a more important role than the division by the norm. This is because with the right depth scaling, it is generally the case that features will remain stable as depth grows (no exploding/vanishing), however, the correlation will induce non-zero means, hence the need for mean substraction to avoid exploding behaviour. Secondly, substracting the mean only decreases the dimension of the layers by 1, which is insignificant in the large width limit. Lastly, one can think of a generalized parametrization in which means of the activations are computed and added back after scaling the centered block. This will result in similar scaling patterns to the ones we propose in our setup. For the sake of simplification, we do not consider this general setup in this paper.

## 5  CLASSIFICATION OF DEPTHWISE PARAMETRIZATIONS

In this section, we provide a comprehensive description of the impact of depth parametrization on stability and update size. For this purpose, we only have two scalings to keep track of: the branch multiplier and the learning rate scaling because the initialization scale is fixed by the faithfulness property (defined below). Requiring that the features don't blow up at initialization means that the branch multipliers must be at most $\Theta(1/\sqrt{L})$. Assuming the updates are faithful (i.e., input to gradient processing functions are $\Theta(1)$ entrywise), the update size can be at most $1/L$ for the hidden layers, by an (Jacobian) operator-norm argument, but potentially much less. Naively speaking, there can be a trade-off between update size and initialization: if initialization is large, then the update may need to be small so as not to blow up the other parts of the network; likewise if the initialization is small, then the update size can be larger. But one may be surprised that a careful calculation shows that there is no trade-off: we can maximize both initialization and update size at the same time.

Before delving into the details, let us first define the notions of training routine, stability, faithfulness, and non-triviality. Hereafter, all the aymptotic notations such as $\mathcal{O}$, $\Omega$ and $o$ should be understood in the limit "$n \to \infty$, then $L \to \infty$". For random variables, such notations should be understood in the sense of weak convergence (convergence in distribution). When we use the notation $x = \mathcal{O}(1)$ for some vector $x = (x_1, \ldots, x_n) \in \mathbb{R}^n$, it should understood in the sense that for all $i \in [n], x_i = \mathcal{O}(1)$. Lastly, we will use bold characters (e.g. $\boldsymbol{h}$ instead of $h$) to denote 'batched' versions of the quantities. This is just to emphasize that the following claims should hold for batched quantities as well.

*Remark:* in this section, we state the results as "claims" instead of theorems. In Appendix G.4, we provide "heuristic" proofs that can be made rigorous under non-trivial technical conditions. We believe this additional layer of complexity is unneeded and do not serve the purpose of this paper. To showcase the correctness, we provide rigorous proof for all the claims in the linear case in Appendix H, which also serves as intuitions for the general case.

**Definition 5.1** (Training routine). A training routine is the package of $\eta$, $\boldsymbol{Q}$, and the input batches.

**Definition 5.2** (Stability). We say a parametrization is

1. *stable at initialization* if $\boldsymbol{h}_0^l, \boldsymbol{x}_0^l = \mathcal{O}(1), \forall l \in [L]$, and $\boldsymbol{f}_0 = \mathcal{O}(1)$.

2. *stable during training* if for any training routine, any time $t \geq 0$, $l \in [L]$, we have $\Delta\boldsymbol{h}_t^l, \Delta\boldsymbol{x}_t^l = \mathcal{O}(1), \forall l \in [L]$, and $\Delta\boldsymbol{f}_t = \mathcal{O}(1)$, where the symbol '$\Delta$' refers to the change after one gradient step.

We say the parametrization is *stable* if it is stable both at initialization and during training.

**Definition 5.3** (Faithful). We say a parametrization is *faithful at step $t$* if $\boldsymbol{h}_t^l = \Theta(1)$ for all $l \in [L]$. We say the parametrization is *faithful* if it is faithful for all $t$. We also say it is *faithful at initialization* (resp. faithful during training) if this is true at $t = 0$ (resp. for $t \geq 1$).

---

[8]Reparametrization is also included in the original $\mu$P, but it is not necessary for the purpose of this paper.

Note faithfulness here refers to "faithfulness to $\phi$", meaning the input to $\phi$ is $\Theta(1)$. This is different from the definition of faithfulness in Yang & Littwin (2023), where faithfulness refers to "faithfulness to $Q$" meaning the input to $Q$ is $\Theta(1)$. "faithfulness to $Q$" is already assumed in this work as mentioned in Section 4.1.

**Definition 5.4** (Nontriviality). We say a parametrization is *trivial* if for every training routine and any time $t \geq 1$, $\boldsymbol{f}_t - \boldsymbol{f}_0 \xrightarrow{\text{a.s.}} 0$ in the limit "$n \to \infty$, then $L \to \infty$" (i.e., the function does not evolve in the infinite-width-then-depth limit). We say the parametrization is *nontrivial* otherwise.

**Definition 5.5** (Feature Learning). We say a parametrization induces *feature learning* in the limit "$n \to \infty$, then $L \to \infty$", if there exist a training routine, and $t \geq 1$, and any $\lambda > 0$, we have $\Delta \boldsymbol{h}_t^{\lfloor \lambda L \rfloor} = \Theta(1)$.

### 5.1 MAIN CLAIMS

We are now ready to state the main results. The next claim provides a necessary and sufficient condition under which a parametrization is stable at initialization.

**Claim 5.1.** *A parametrization is stable at initialization iff $\alpha \geq 1/2$.*

Claim 5.1 is not new and similar results were reported by Hayou et al. (2021). However, Hayou et al. (2021) focuses on initialization and lacks a similar stability analysis during training. In the next result, we identify two different behaviours depending on the scaling of the learning rate.

**Claim 5.2.** *Consider a parametrization that is stable at initialization. Then the following hold (separately from each other): a) It is stable during training as well iff $\alpha + \gamma \geq 1$; b) It is nontrivial iff $\alpha + \gamma \leq 1$. Therefore, it is both stable and nontrivial iff $\alpha + \gamma = 1$.*

From Claim 5.1 and Claim 5.2, having $\alpha + \gamma = 1$ and $\alpha \geq 1/2$ is a necessary and sufficient condition for a parametrization to be stable and nontrivial throughout training. In the next result, we therefore restrict our analysis to such parametrizations and study their faithfulness.

**Claim 5.3.** *Consider a stable and nontrivial parametrization. The following hold (separately from each other).*

- *It is faithful at initialization iff $\alpha \geq 1/2$. As a result, $\alpha = 1/2$ is the minimal choice of $\alpha$ that guarantees faithfulness.*

- *It is faithful during training iff $\alpha \leq 1$.*

*Therefore, a stable and nontrivial parametrization is faithful iff $\alpha \in [1/2, 1]$.*

The first claim follows from well-known calculations of randomly initialized residual networks Hayou et al. (2021). For the second claim, the intuition here is just that if $\alpha + \gamma = 1$ and $\alpha > 1$ then $\gamma < 0$, i.e., the update size blows up with depth. This would then cause the input to the nonlinearities to blow up with size.

Now by focusing on stable, nontrivial, and faithful parametrizations, we have narrowed down the space of $(\alpha, \gamma)$ to $\alpha + \gamma = 1$ and $\alpha \in [1/2, 1]$. However, in order to achieve *optimal* hyperparameter transfer, we need to identify the *optimal* parametrization among them. (See Appendix I for detailed explanation on what causes hyperparameter transfer.) To accomplish this, we first define the notion of feature diversity exponent, which relates to the similarity in the features of adjacent layers.

**Definition 5.6** (Feature Diversity Exponent). We say a parametrization has feature diversity exponent $\kappa \geq 0$ if $\kappa$ is the maximal value such that for all $\lambda \in [0, 1]$ and sufficiently small $\epsilon > 0$, and all time $t$,

$$\frac{1}{\sqrt{n}} \left\| \boldsymbol{x}_t^{\lfloor (\lambda + \epsilon) L \rfloor} - \boldsymbol{x}_t^{\lfloor \lambda L \rfloor} \right\| = \Omega(\epsilon^{1-\kappa}),$$

where $\Omega(1)$ should be interpreted in the limit "$n \to \infty$, then $L \to \infty$, then $\epsilon \to 0$". We say a parametrization is *redundant* if $\kappa = 0$.

In other words, the feature diversity exponent $\kappa$ is a measure of how different the outputs are in layers that are close to each other. With $\kappa = 0$, the output of each layer is essentially the same as the output of the previous layer in the sense that the rate of change from one layer to the next is bounded (at least locally), and hence the network is intuitively "wasting" parameters.

**Claim 5.4.** *Consider a stable and nontrivial parametrization that is furthermore faithful during training (but not necessarily at initialization). Then it is redundant if $\alpha \in (1/2, 1]$.*

To understand the intuition behind Claim 5.4, let us see what happens when $\alpha > 1/2$. In this case, the randomness of the initialization weights will have no impact on training trajectory as depth increases. To see this, consider some layer index $\lfloor \lambda L \rfloor$. The blocks are divided by $L^\alpha$ which is larger than the magnitude of accumulated randomness (of order $(\lambda L)^{1/2}$). This basically destroys all the randomness from initialization and therefore the randomness in the learned features will consist only of that coming from $U$ and $V$ (input and output matrices). When depth goes to infinity, the contribution of the randomness in two adjacent layers becomes less important, we end up with adjacent layers becoming very similar because the gradients to these layers are highly correlated.

In contrast, we have the following result, which defines Depth-$\mu$P.

**Claim 5.5** (Depth-$\mu$P). *$\alpha = \gamma = 1/2$ is the unique parametrization that is stable, nontrivial, faithful, induces feature learning, and achieves maximal feature diversity with $\kappa = 1/2$.*

Note feature learning is not the unique property of Depth-$\mu$P[9], but in terms of feature diversity, a phase transition phenomenon occurs when $\alpha = 1/2$. More precisely, for Depth-$\mu$P, we can show that $n^{-1/2} \left\| \boldsymbol{x}_t^{\lfloor (\lambda + \epsilon) L \rfloor} - \boldsymbol{x}_t^{\lfloor \lambda L \rfloor} \right\| = \mathcal{O}(\epsilon^{1/2})$ while the same quantity is $\mathcal{O}(\epsilon)$ for all $\alpha \in (1/2, 1]$, which suggests that Depth-$\mu$P yields *rough* path for $\boldsymbol{x}_t$. This allows the features to change significantly from one layer to the next, hence efficiently using the parameters. For readers who are familiar with rough path theory, the $1/2$ continuity exponent is a result of Brownian increments in the path.[10]

Moreover, with $\alpha = 1$, there is a phenomenon of feature collapse in the sense that the features will be contained in the $\sigma$-algebra generated by the input and output layers, but contains no randomness from the hidden layers (see Appendix G.2). Intuitively, the case of $\alpha = 1$ is analogous to width situation, where deep mean field collapses to a single neuron (all neurons become essentially the same). For depth, the features (layers) are still relatively different but the redundancy does not allow significant variety in these features.

In Appendix J, we provide more insight into feature diversity and shed light on its importance. We show that the choice of nonlinearity and placement of nonlinearity can affect feature diversity, and indeed the best architecture that maximizes feature diversity also performs better in the experiments. We also provide empirical verification of our claims about feature diversity in Appendix L.4.

## 6 EXPERIMENTS

In this section, we provide empirical evidence to show the optimality of Depth-$\mu$P scaling and the transferability of hyperparameters across depth. We train vanilla residual network with block depth 1 (1 MLP layer in each residual block) on CIFAR10 dataset using Adam optimizer, batch size $64$, for 50 epochs. The parametrization of the network and the weight update rule are as follows,

$$x^l = x^{l-1} + a \times L^{-\alpha} \mathrm{MS}(\phi(W^l x^{l-1})), \quad W^l \leftarrow W^l - \eta \times n^{-1} L^{-\gamma} Q_t^l(nL^\delta g_0, \ldots, nL^\delta g_t),$$

where the learning rate $\eta$ and the block multiplier $a$ are the *hyperparameters*.[11] The nonlinearity $\phi$ is ReLU. The values of $\alpha, \gamma$ depend on the parametrization of choice. For Depth-$\mu$P, we have $\alpha = \gamma = 1/2$, and for standard parametrization, we have $\alpha = 0, \gamma = 1$.[12]

**Learning rate transfer ($\eta$).** In Figure 2, we show the training loss versus learning rate for depths $2^k$, for $k \in \{3, 4 \ldots, 10\}$. For Depth-$\mu$P, a convergence pattern can be observed for the optimal learning rate as depth grows. Optimal learning rates for small depths (e.g. $L = 2^3$) exhibit a mild shift which should be expected, as our theory shows convergence in the large depth limit. However, starting from depth $L = 2^6$, the optimal learning rate is concentrated around $10^{-3}$. With standard

---

[9]Stable, nontrivial, and faithful parametrizations all induce feature learning

[10]The reader might ask whether we can obtain an exponent smaller than $1/2$. This is indeed possible, but it will entail using correlated weights. We leave this question for future work.

[11]Note that $\eta$ here is constant across depths and widths, and the effective learning rate is given by $\eta n^{-1} L^{-\gamma}$.

[12]In standard parametrization, there is generally no rule to scale the learning rate with depth, and the optimal learning rate is typically found by grid search. Here, we assume that in standard parametrization, the learning rate is scaled by $L^{-1}$ to preserve faithfullness.

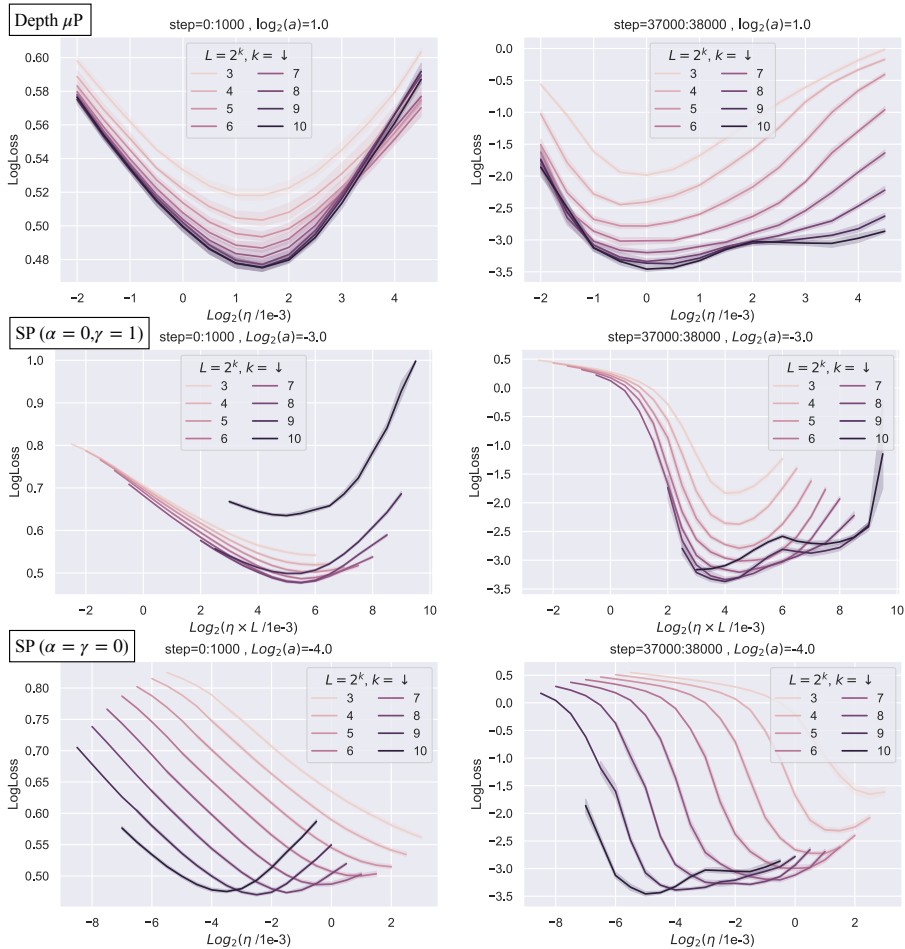

Figure 2: Training loss versus learning rate at varying depths for Depth-$\mu$P (top row) and two Standard Parametrizations (bottom two rows). The network consists of MLP blocks with width $n = 256$ and block depth 1. We fix its input/output layers and train its hidden layers for 50 epochs with batch size 64 on CIFAR10 using Adam. Each curve represent the average training loss over a time slice of 1000 steps for depths $2^k$ for $k \in \{3, 4, \ldots, 10\}$. Confidence intervals are based on 5 seeds. The results show that Depth-$\mu$P preserves optimal learning rate while consistently improving the training loss as depth increases.

parametrization with learning rate scaling ($\alpha = 0, \gamma = 1$), the optimal learning rate exhibits a significant shift with depth and training loss degrades when the depth is too large. Here we have already set $a = 2^{-3}$, making the curves look better compared to the standard practice with $a = 1$. For standard parametrization without any depth scaling ($\alpha = \gamma = 0$), the optimal learning rate exhibits a significant shift as depth grows (even with smaller $a = 2^{-4}$), suggesting that standard parametrization is not suitable for depth scaling. Additional figures with multiple time slices are provided in Appendix M.

**Block multiplier transfer** ($a$). In Figure 14, we investigate the stability of the hyperparameter $a$ in Depth-$\mu$P as depth increases. The results suggest that the optimal value of this constant converges as depth grows, which suggest transferability. Additional experiments with multiple time slices are provided in Appendix M.

## 7 DISCUSSION

This research contributes a foundational understanding of the optimal depth scaling for neural networks, showcasing the effectiveness of Depth-$\mu$P through rigorous theoretical justifications and practical experiments. As the field of deep learning continues to evolve, our work serves as a valuable guidepost for developing more efficient and powerful neural network architectures while also paving the way for exciting future investigations into the scaling of advanced architectures.

While our current theory establishes the optimality of depth-$\mu$P in the case of MLP blocks, the applicability of this scaling principle to more advanced architectures, such as Transformers, should in-principle yield favorable outcomes. We leave this for future work.

ACKNOWLEDGEMENT

We thank Huishuai Zhang, Jeremy Bernstein, Edward Hu, Michael Santacroce, Liyuan Liu for their helpful comments and discussion. D. Yu was supported by NSF and ONR. Part of this work was done during D. Yu's internship at Microsoft.

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

# A  RELATED WORKS

## A.1  WIDTH SCALING AND $\mu$P

The infinite-width limit of neural networks has been a topic of extensive research in the literature. Numerous studies have predominantly focused on examining the behavior of various statistical quantities at initialization. Some works have gone beyond the initialization stage to explore the dynamics of feature learning in neural networks.

**Lazy Training.**  With standard parametrization, a learning rate of order $\mathcal{O}(n^{-1})$,[13] $n$ being the width, yields the so-called lazy training regime in the infinite-width limit, where the features remain roughly constant throughout training (Chizat et al., 2020; Yang et al., 2022). This regime is also known as the Neural Tangent Kernel (NTK) regime and its convergence properties have been extensively studied in the literature (Jacot et al., 2020; Allen-Zhu et al., 2019; Chizat & Bach, 2018; Zou et al., 2018).

**Feature learning and $\mu$P.**  Recent empirical studies (e.g. Yang et al. (2022)) have provided compelling evidence that feature learning plays a crucial role in the success of deep learning. It is widely acknowledged that the remarkable performance achieved by deep neural networks can be attributed to their ability to acquire meaningful representations through the process of training. Consequently, scaling the network architecture emerges as a natural choice to enhance the performance of such models.

In this context, $\mu$P (Maximal Update Parametrization), introduced in Yang et al. (2022), has emerged as a promising approach for maximizing feature learning while simultaneously preventing feature explosion as the network width increases, given a fixed depth. Notably, $\mu$P facilitates hyperparameter transfer across varying network widths. This means that instead of tuning hyperparameters directly on large models, one can optimize them on smaller models and utilize the same set of hyperparameters for larger models.

The derivation of $\mu$P leverages the Tensor Programs framework (Yang, 2021; 2020a;b; Yang & Hu, 2021; Yang et al., 2022), which provides valuable tools for capturing the behavior of neural networks in the infinite-width regime during the training process.

## A.2  DEPTH SCALING

While increasing the width of neural networks can lead to improved performance, increasing the depth of the network also yields significant performance gains, and most state-of-the-art models use deep architectures. The introduction of skip connections (He et al., 2016a;b) played a pivotal role in enabling the training of deep networks. However, it became apparent that even with skip connections and normalization layers, training deep networks remains a challenging task (Liu et al., 2020). Moreover, tuning hyperparameters for large depth networks is a time-and-resource-consuming task.

To address the challenges associated with training deep networks, several studies have proposed scaling the network blocks using a depth-dependent scaler to ensure stability of features and gradients at initialization (Hanin & Rolnick, 2018; Zhang et al., 2019; Hayou et al., 2021; Noci et al., 2022; Hayou, 2023; Hayou & Yang, 2023; Noci et al., 2023). However, these works primarily focus on stability at initialization and lack insights into the dynamics during the training process. For instance, one might argue that features can still experience explosive growth if the learning rate is not properly chosen. Therefore, an effective depth scaling approach should not only ensure stability at initialization but also provide guidelines for scaling the learning rate.

This motivation underlies the development of Depth-$\mu$P, which offers a comprehensive framework for depth scaling. Depth-$\mu$P encompasses block multipliers and learning rate scaling, providing a complete recipe for training deep networks. In the case of Multi-Layer Perceptrons (MLPs) (no skip connections), Jelassi et al. (2023) showed that a learning rate scaling of $depth^{-3/2}$ guarantees stability after the initial gradient step. However, it remains unclear how the learning rate should be adjusted beyond the first step, and this scaling is not suitable for architectures with residual connections.

---

[13]We also obtain the lazy infinite-width limit with the NTK paramterization and a $\mathcal{O}(n^{-1/2})$ learning rate.

# B CONTINUATION OF THE INTUITIVE EXPLANATION WITH LINEAR NETWORKS

Recall the network model

$$\forall l \in [L], \quad x^l = \frac{1}{\sqrt{L}} W^l x^{l-1} + x^{l-1},$$

where $x^0 = U\xi$, resp. $f(\xi) = V^\top x^L$ is the input, resp. output, layer, and $W^l \in \mathbb{R}^{n \times n}$ are the weight matrices, and $U, V$ are input and output weight matrices that we assume to be fixed during training.

## B.1 CONVERGENCE WHEN DEPTH GOES TO $\infty$

Let us look at $x_1^L$ again in the simple case $d_{in} = d_{out} = n = 1$ and analyze its behaviour when $L \to \infty$. This paragraph is only intended to give an intuition for the convergence. A rigorous proof of such convergence will be later presented in the paper. Let us consider the case with SGD training with learning rate $\eta = 1$ and let $M_{L,l} = \prod_{k \neq l} \left(1 + \frac{1}{\sqrt{L}} W_0^k\right)$ and $\tau = (V x_0^L - y(\xi_0)) x^0$. With this, we have the following

$$x_1^L = \prod_{l=1}^{L} \left(1 + \frac{1}{\sqrt{L}} W_0^l - \frac{1}{L} \tau M_{L,l}\right) x^0. \tag{3}$$

WLOG, let us assume that $x_0^0 > 0$. Then, with high probability (the event that $W_0^l \ll \sqrt{L}$, for some notion of "$\ll$", occurs with a probability of at least $1 - e^{-L^\alpha}$ for some $\alpha > 0$)[14], we have that $x_1^L > 0$. We can therefore look at $\log(x_1^L)$ which simplifies the task. Taking the log and using Taylor expansion under a high probability event, we obtain

$$\log(x_1^L/x^0) = \frac{1}{\sqrt{L}} \sum_{l=1}^{L} W_0^l - \frac{1}{L} \sum_{l=1}^{L} \tau M_{L,l} + \frac{\sum_{l=1}^{L} (W_0^l)^2}{L} + \mathcal{O}(L^{-1+\epsilon})$$

$$= \frac{1}{\sqrt{L}} \sum_{l=1}^{L} W_0^l - \tau x_0^L \frac{1}{L} \sum_{l=1}^{L} \frac{1}{1 + \frac{1}{\sqrt{L}} W_0^l} + \frac{\sum_{l=1}^{L} (W_0^l)^2}{L} + \mathcal{O}(L^{-1+\epsilon}),$$

for some $\epsilon > 0$. The first and third terms $\frac{1}{\sqrt{L}} \sum_{l=1}^{L} W_0^l$ and $\frac{\sum_{l=1}^{L} (W_0^l)^2}{L}$ converge (almost surely) to a standard Gaussian and 1, respectively. The second term also converges naturally, since $x_0^L$ converges in $L_2$ to a Log-Normal random variable (Hayou (2023)) and with a delicate treatment (involving high probability bounds), one can show that the term $\frac{1}{L} \sum_{l=1}^{L} \frac{1}{1 + \frac{1}{\sqrt{L}} W_0^l}$ converges (in $L_2$ norm) at large depth. This implies that one should expect $x_1^L$ to have some notion of weak convergence as depth grows. Note that the same analysis becomes much more complicated for general width $n > 0$. To avoid dealing with high probability bounds, a convenient method consists of taking the width to infinity first $n \to \infty$, then analyzing what happens as depth increases. We discuss this in the next section.

## B.2 A DISCUSSION ON THE GENERAL CASE

**Difficulty of generalizing to the nonlinear case.** The extension to the general width scenario ($n > 1$) necessitates a more intricate treatment of the term $A_l$ to find optimal scaling rules, yet the proposed scaling remains optimal for general width. This preliminary analysis lays the groundwork for proposing a specific learning rate scaling scheme that maximizes feature learning. Moreover, demonstrating the optimality of this scaling strategy in the presence of non-linearities is a nontrivial task. The primary challenge stems from the correlation among the post-activations induced during the training process. Overcoming these challenges requires a rigorous framework capable of addressing the large depth limit of crucial quantities in the network.

For this purpose, we employ the Tensor Program framework to investigate the behavior of essential network quantities in the infinite-width-then-depth limit. By leveraging this framework, our

---

[14]This follows from simple concentration inequalities for sub-exponential random variables.

theoretical findings establish that the aforementioned scaling strategy remains optimal for general networks with skip connections. Our framework considers the setup where the width is taken to infinity first, followed by depth. This represents the case where $1 \ll depth \ll width$, which encompasses most practical settings (e.g. Large Language Models).

**The critical role of Initialization.** A naive approach to depth scaling can be as follows: since the weights $W_t^k$ might become highly correlated during training, one has to scale the blocks with $1/L$. To understand this, let us assume a block multiplier of $L^{-\alpha}$ and consider the scenario of perfect correlation where all weights are equal, i.e., $W_t^k = W$ for every $k \in 1, \ldots, L$. In this case, the last layer features can be expressed as $x^L = (I + L^{-\alpha}W)^L x_0$. When $\alpha = 1/2$, the features are likely to exhibit an explosive growth with increasing depth, while opting for $\alpha = 1$ is guaranteed to stabilize the features.

However, in this paper, we demonstrate that this intuition does not align with practical observations. Contrary to expectations, the features do not undergo an explosive growth as the depth increases when $\alpha = 1/2$. This phenomenon is attributed to two key factors: random initialization and learning rate scaling with depth. These factors ensure that the weight matrices never become highly correlated in this particular fashion during the training process.

In summary, while a naive depth scaling strategy based on scaling blocks might suggest the need for $\alpha = 1$ to stabilize the features, our findings reveal that in practice, this is not the case. The interplay of random initialization and learning rate scaling effectively prevents the features from experiencing explosive growth, even with the choice of $\alpha = 1/2$.

## C  CONTINUATION OF SGD TRAINING DYNAMICS

In this section, we continue to study the linear neural network with residual connections under Depth-$\mu$P. We derive the training dynamics of SGD for this linear network when the width and the depth sequentially go to infinity. The road map of our analysis consists the following three steps.

1. We first take the width of the network to infinity by the Tensor Program framework (Yang & Littwin, 2023). As a result, instead of tracking vectors and matrices along the training trajectory, we track random variables that correspond to the vectors, that is, for a vector $x \in \mathbb{R}^n$ that appears in the computation of the training, the coordinates of $x$ can be viewed as iid copies of random variable $\|x\rangle$ (called a *ket*) when $n \to \infty$. [15]

2. Since the network is linear, every random variable can be written as a linear combination of a set of zero-mean "base" random variables by the Master Theorem of Tensor Programs (Yang & Littwin, 2023). Therefore, we can track the random variables by analyzing the coefficients of their corresponding linear combinations, along with the covariance between the "base" random variables.

3. Since the number of random variables and the number of "base" random variables scale linearly with $L$, the coefficients of all random variables can be represented by a six dimensional tensor, where two of the dimensions have shape $L$. We then map the tensor to a set of functions whose input domain is $[0,1] \times [0,1]$. Finally, we claim that the functions converge when $L \to \infty$, and identify their limits as the solution of a set of integral functional.

Formally, under mild assumptions[16], we first prove that for any depth $L$, when $n \to \infty$, the training dynamics of SGD can be characterized by two sets of functions $\{\Gamma_{t,s,a,b} : [0,1] \times (0,1] \to \mathbb{R}\}_{t \in \{0,\ldots,T-1\}, s \in \{-1,\ldots,t\}, a,b \in \{0,1\}}$ and $\{C_{t,s,a} : (0,1] \to \mathbb{R}\}_{t,s \in \{-1,\ldots,T-1\}, a \in \{0,1\}}$, where $T$ is the total number of training steps. The network output at time $t$ is $\Gamma_{t,-1,0,1}(1,1)$ in the infinite width limit, and the formal definition of $\Gamma$ and $C$ can be found in Appendix C.

---

[15]The definition of $\|x\rangle$ requires the coordinates of $x$ is $\mathcal{O}(1)$ w.r.t. $n$, and $\|x\rangle$ is trivial if the coordinates of $x$ is $o(1)$ w.r.t. $n$. Therefore, for $x$ whose coordinates are not $\Theta(1)$, we normalize $x$ by multiplying polynomial of $n$ so the resulting vector has coordinates $\Theta(1)$.

[16]Other than the assumptions of Tensor Programs (e.g. the loss function is well-conditioned), the rest assumptions (e.g., the input and output dimension is 1; the batch size is 1) are only for the convenience of the notations and can be easily generalized within the Tensor Program framework.

Then we take the depth $L$ to infinity, and find $\Gamma$ and $C$ is the solution of the following functional integral in the limit as in Proposition C.4.

**Assumptions and Notations**    Recall the linear network is given by

$$x^0 = U\xi,$$

$$\forall l \in [L], \quad x^l = \frac{a}{\sqrt{L}}W^l x^{l-1} + x^{l-1},$$

$$f = V^\top x^L.$$

For convenience, we assume $a = 1$, the SGD learning rate of $W^l$ is 1. We add $t$ as a subscript to any notation to denote the same object but at $t$-th training step, e.g., the input at step $t$ is a single datapoint $\xi_t$, the hidden output of $l$-th layer at step $t$ is $x_t^l$, and the model output at step $t$ is $f_t$. Let $T$ be the number of training steps. Let $\ell_t$ be the loss function absorbing the label at time $t$, and $\chi_t$ be the derivative of the loss at time $t$, i.e., $\chi_t = \ell_t'(f_t)$. Let $dx_t^l = \partial\ell_t/\partial x_t^l$, and its normalized version $\delta x_t^l = n \cdot dx_t^l$.

The Tensor Program analysis heavily depends on the scaling of initialization and learning rate of $U, V, W$ w.r.t $n$. In this paper, we use $\mu$P as the scaling w.r.t. $n$ since it maximizes feature learning in the large width limit (Yang & Hu, 2021). Without loss of generality, we follow Yang & Hu (2021) and assume the input and output dimension is 1, i.e., $\xi \in \mathbb{R}$, $f \in \mathbb{R}$. For a clean presentation, we additionally assume $U, V$ are frozen during training in this section and each coordinate of $W$ is initialized with i.i.d. Gaussian of variance $1/n$.

## C.1    Width Limit under $\mu$P

As the first step, we take the width of the network $n$ to infinity using Tensor Programs (TP). As briefly mentioned in the road map of the section, the TP framework characterizes each vector involved in the training procedure by a random variable when $n \to \infty$. For a vector $x \in \mathbb{R}^n$ that has roughly iid coordinates, we write $\|x\rangle \in \mathbb{R}$ (called a *ket*) to denote a random variable such that $x$'s entries look like iid copies of $\|x\rangle$. Then for any two vector $x, y \in \mathbb{R}^n$ that have roughly iid coordinates, their limiting inner product by $n$ can be written as $\lim_{n\to\infty} \frac{x^\top y}{n} = \mathbb{E}\|x\rangle \cdot \|y\rangle$, which we write succinctly as $\langle x\|y\rangle$. Deep linear network with SGD is a simple example for this conversion from vectors to random variables. As shown in Program 1, we define a series of scalars ($\mathring{f}_t$ and $\mathring{\chi}_t$) and random variables ($\|U\rangle, \|nV\rangle, \|x_t^l\rangle, \|\delta x_t^l\rangle, \|W_t^l x_t^{l-1}\rangle, \|W_t^{l\top}\delta x_t^l\rangle$) using the ket notations. For better understanding, we provide a brief introduction to TP below.

**Tensor Programs (TP) in a nutshell.**    When training a neural network, one can think of this procedure as a process of successively creating new vectors and scalars from an initial set of random vectors and matrices (initialization weights), and some deterministic quantities (dataset in this case). In the first step, the forward propagation creates the features $x_0^l$ where the subscript 0 refers to initialization, and the scalar $f_0$, which is the network output. In the first backward pass, the output derivative $\chi_0$ is computed, then the gradients $dx_0^l$ are backpropagated. (Since the coordinates of $dx_0^l$ vanish to 0 when $n \to \infty$, TP instead tracks its normalized version $\delta x_0^l \stackrel{\text{def}}{=} n \cdot dx_0^l$.) New vectors are created and appended to the TP as training progresses. When the width $n$ goes to infinity, vectors of size $n$ in the TP (e.g., the features $x_t^l$, and normalized gradients $\delta x_t^l$) see their coordinates converge to roughly iid random variables (e.g., $\|x_t^l\rangle$ and $\|\delta x_t^l\rangle$ in Program 1), and other scalar quantities (e.g., $f_t$ and $\chi_t$) converge to deterministic values (e.g., $\mathring{f}_t$ and $\mathring{\chi}_t$ in Program 1) under proper parametrization ($\mu$P). The Master Theorem (Yang et al., 2022) captures the behaviour of these quantities by characterizing the *infinite-width* limit of the training process. For more in-depth definitions and details about TP, we refer the reader to Yang et al. (2022).

Now when we look back to Program 1, the definitions of scalars and random variables should be clear (except for $\|W_0^l x_t^{l-1}\rangle$ and $\|W_0^{l\top}\delta x_t^l\rangle$). One can find straightforward correspondence between those and their finite counterpart, for example:

- $\mathring{f}_t$ corresponds to $f_t$, and $\mathring{\chi}_t$ corresponds to $\chi_t$;
- $\|x_t^l\rangle$ corresponds to $x_t^l$ and $\|\delta x_t^l\rangle$ corresponds to $\delta x_t^l$. (Recall $\delta x_t^l = n \cdot dx_t^l$ is the normalized version of $dx_t^l$.)

---

**Program 1:** Random Variables induced from Tensor Program for the Linear Network with LR $\eta = 1$ and frozen $U, V$.

---

**Initial random variables:** $[\![U]\!\rangle, [\![nV]\!\rangle$ are independent standard Gaussian.

**for** $t = 0, \dots, T-1$ **do**

$\quad [\![x_t^0]\!\rangle \stackrel{\text{def}}{=} \xi_t [\![U]\!\rangle$;

$\quad$ **for** $l = 1, \dots, L$ **do**

$\quad\quad [\![W_t^l x_t^{l-1}]\!\rangle \stackrel{\text{def}}{=} [\![W_0^l x_t^{l-1}]\!\rangle - \frac{1}{\sqrt{L}} \sum_{s=0}^{t-1} [\![\delta x_s^l]\!\rangle \langle x_s^{l-1} [\![x_t^{l-1}]\!\rangle$;

$\quad\quad [\![x_t^l]\!\rangle \stackrel{\text{def}}{=} [\![x_t^{l-1}]\!\rangle + \frac{1}{\sqrt{L}} [\![W_t^l x_t^{l-1}]\!\rangle$;

$\quad$ **end**

$\quad \mathring{f}_t \stackrel{\text{def}}{=} \langle x_t^L [\![nV]\!\rangle$;

$\quad \mathring{\chi}_t \stackrel{\text{def}}{=} \ell'_t(\mathring{f}_t)$;

$\quad [\![\delta x_t^L]\!\rangle \stackrel{\text{def}}{=} \mathring{\chi}_t [\![nV]\!\rangle$;

$\quad$ **for** $l = L, \dots, 1$ **do**

$\quad\quad [\![W_t^{l\top} \delta x_t^l]\!\rangle \stackrel{\text{def}}{=} [\![W_0^{l\top} \delta x_t^l]\!\rangle - \frac{1}{\sqrt{L}} \sum_{s=0}^{t-1} [\![x_s^{l-1}]\!\rangle \langle \delta x_s^l [\![\delta x_t^l]\!\rangle$;

$\quad\quad [\![\delta x_t^{l-1}]\!\rangle \stackrel{\text{def}}{=} [\![\delta x_t^l]\!\rangle + \frac{1}{\sqrt{L}} [\![W_t^{l\top} \delta x_t^l]\!\rangle$;

$\quad$ **end**

**end**

where $[\![W_0^l x_t^{l-1}]\!\rangle$ and $[\![W_0^{l\top} \delta x_t^l]\!\rangle$ are defined in Definition C.1.

---

- By SGD, $W_t^l = W_0^l - \frac{1}{\sqrt{L}} \sum_{s=0}^{t-1} dx_s^l \otimes x_s^{l-1}$, which corresponds to $[\![W_t^l x_t^{l-1}]\!\rangle = [\![W_0^l x_t^{l-1}]\!\rangle - \frac{1}{\sqrt{L}} \sum_{s=0}^{t-1} [\![\delta x_s^l]\!\rangle \langle x_s^{l-1} [\![x_t^{l-1}]\!\rangle$.

Now we can dive into the definition of $[\![W_0^l x_t^{l-1}]\!\rangle$ and $[\![W_0^{l\top} \delta x_t^l]\!\rangle$. Let $\mathcal{W}$ be the set of initial random matrices of size $n \times n$, i.e., $\{W_0^1, \dots, W_0^L\}$, and $\mathcal{W}^\top \stackrel{\text{def}}{=} \{W^\top : W \in \mathcal{W}\}$. Let $\mathcal{V}_W$ denote the set of all vectors in training of the form $Wy$ for some $y$. Then for every $W \in \mathcal{W} \cup \mathcal{W}^\top$, and $Wy \in \mathcal{V}_W$, we can decompose $[\![Wy]\!\rangle$ into the sum of $[\![W\widehat{y}]\!\rangle$ and $[\![W\dot{y}]\!\rangle$, where $[\![W\widehat{y}]\!\rangle$ is a random variable that act as if $W$ were independent of $y$, and $[\![W\dot{y}]\!\rangle$ is the random variable capturing the correlation part between $W$ and $y$. Specifically, let us briefly track what happens to $W_0^l x_t^{l-1}$ during training. In the first step, we have $W_0^l x_0^{l-1}$ which has roughly Gaussian coordinates (in the large width limit). In this case, we have $[\![W_0^l x_0^{l-1}]\!\rangle = 0$. After the first backprop, we have $dx_0^{l-1} = dx_0^l + \frac{1}{\sqrt{L}} W_0^{l\top} dx_0^l$, which means that the update in $W^{l-1}$ will contain a term of the form $W_0^{l\top} z$ for some vector $z$. This implies that $W_0^l x_1^{l-1}$ will contain a term of the form $W_0^l W_0^{l\top} z'$ for some vector $z'$. This term induces an additional correlation term that appears when we take the width to infinity. The $[\![W_0^l x_1^{l-1}]\!\rangle$ is defined by isolating this additional correlation term from $W_0^l W_0^{l\top} z'$. The remaining term is Gaussian in the infinite-width limit, which defines the term $[\![W_0^l x_1^{l-1}]\!\rangle$. Formally, we present the following definition.

**Definition C.1.** We define $[\![Wy]\!\rangle \stackrel{\text{def}}{=} [\![W\widehat{y}]\!\rangle + [\![W\dot{y}]\!\rangle$ for every $W \in \mathcal{W} \cup \mathcal{W}^\top$ and $Wy \in \mathcal{V}_W$, where

- $[\![W\widehat{y}]\!\rangle$ is a Gaussian variable with zero mean. $\forall W \in \mathcal{W} \cup \mathcal{W}^\top, Wy, Wz \in \mathcal{V}_W$,

$$\text{Cov}\left([\![W\widehat{y}]\!\rangle, [\![W\widehat{z}]\!\rangle\right) \stackrel{\text{def}}{=} \langle y [\![z]\rangle.$$

  $\forall W, W' \in \mathcal{W} \cup \mathcal{W}^\top, Wy \in \mathcal{V}_W, W'z \in \mathcal{V}_{W'}$, $[\![W\widehat{y}]\!\rangle$ and $[\![W'\widehat{z}]\!\rangle$ are independent if $W \neq W'$. $[\![W\widehat{y}]\!\rangle$ is also independent from $[\![U]\!\rangle$ and $[\![nV]\!\rangle$.

- $[\![W\dot{y}]\!\rangle$ is defined to be a linear combination of $\{[\![z]\!\rangle : W^\top z \in \mathcal{V}_{W^\top}\}$. Then we can unwind any $[\![y]\!\rangle$ inductively as a linear combination of $[\![\bullet \widehat{\ }]\!\rangle, [\![U]\!\rangle$ and $[\![nV]\!\rangle$, which allows us to fully

define

$$\llbracket Wy \rangle \overset{\text{def}}{=} \sum_{W^\top z \in \mathcal{V}_{W^\top}} \llbracket z \rangle \cdot \frac{\partial \llbracket y \rangle}{\partial \widehat{\llbracket W^\top z \rangle}}.$$

## C.2 Depthwise Scaling of Random Variables

As mentioned in Definition C.1, both $\llbracket x_t^l \rangle$ and $\llbracket \delta x_t^{l-1} \rangle$ can be written as linear combination of "base" random variables: $\{\widehat{\llbracket W_0^m x_s^{m-1} \rangle}\}_{s \in \{0,\dots,t\}, m \in [L]}, \{\widehat{\llbracket W_0^{m\top} \delta x_s^m \rangle}\}_{s \in \{0,\dots,t\}, m \in [L]}, \llbracket U \rangle$ and $\llbracket nV \rangle$. Moreover, the coefficients of the linear combinations can be calculated in a recursive way: by expanding $\llbracket W_0^l x_t^{l-1} \rangle$ using Definition C.1, we have

$$\llbracket x_t^l \rangle = \llbracket x_t^{l-1} \rangle + \frac{1}{\sqrt{L}} \widehat{\llbracket W_0^l x_t^{l-1} \rangle} + \frac{1}{\sqrt{L}} \sum_{s=1}^{t-1} \llbracket \delta x_s^l \rangle \left( \frac{\partial \llbracket x_t^{l-1} \rangle}{\partial \widehat{\llbracket W_0^{l\top} \delta x_s^l \rangle}} - \frac{1}{\sqrt{L}} \langle x_s^{l-1} \llbracket x_t^{l-1} \rangle \right).$$

The recursive formula for $\llbracket \delta x_t^l \rangle$ is similar.

Using this induction, we claim in the linear combinations, the coefficient of every $\llbracket \widehat{\bullet} \rangle$ is $\mathcal{O}(1/\sqrt{L})$, and the coefficient of $\llbracket U \rangle$ and $\llbracket nV \rangle$ is $\mathcal{O}(1)$. We also claim the covariance between any pairs of random variables in the form of $\llbracket x_t^l \rangle$ and $\llbracket \delta x_t^{l-1} \rangle$ is $\mathcal{O}(1)$.

**Proposition C.2.** $\forall t, \forall s \le t, \forall l, m, \forall \llbracket y \rangle \in \{\llbracket x_t^l \rangle, \llbracket \delta x_t^l \rangle\}$,

$$\frac{\partial \llbracket y \rangle}{\partial \widehat{\llbracket W_0^m x_s^{m-1} \rangle}} = \mathcal{O}\left(\frac{1}{\sqrt{L}}\right), \frac{\partial \llbracket y \rangle}{\partial \widehat{\llbracket W_0^{m\top} \delta x_s^m \rangle}} = \mathcal{O}\left(\frac{1}{\sqrt{L}}\right), \frac{\partial \llbracket y \rangle}{\partial \llbracket U \rangle} = \mathcal{O}(1), \frac{\partial \llbracket y \rangle}{\partial \llbracket nV \rangle} = \mathcal{O}(1).$$

$\forall t, s, l, m, \forall \llbracket y \rangle \in \{\llbracket x_t^l \rangle, \llbracket \delta x_t^l \rangle\}, \forall \llbracket z \rangle \in \{\llbracket x_s^m \rangle, \llbracket \delta x_s^m \rangle\}$,

$$\langle y \llbracket z \rangle = \mathcal{O}(1).$$

The reasoning of Proposition C.2 is provided in Appendix D. Note the computation of covariance can also be written as a recursive formula. The reasoning relies essentially on an inductive argument.

## C.3 Infinite Depth Limit

Now we formalize our argument above and obtain the formula describing the dynamics of the network when $L \to \infty$. We first write the coefficients of the linear combinations as a six dimensional tensor $\boldsymbol{\Gamma}_{t,s,a,b,l,m}$, where $t, s \in \{0, \dots, T-1\}, a, b \in \{0, 1\}, l, m \in [L]$. Specifically, $\boldsymbol{\Gamma}_{t,s,a,b,l,m}$ represents the derivative of $\llbracket x_t^l \rangle$ and $\llbracket \delta x_t^l \rangle$ w.r.t. $\widehat{\llbracket W_0^m x_s^{m-1} \rangle}$ and $\widehat{\llbracket W_0^{m\top} \delta x_s^m \rangle}$. Here, we use 0 to denote kets appears in the forward pass ($\llbracket x_t^l \rangle$ and $\widehat{\llbracket W_0^m x_s^{m-1} \rangle}$), and 1 to denote kets in the backward pass ($\llbracket \delta x_t^l \rangle$ and $\widehat{\llbracket W_0^{m\top} \delta x_s^m \rangle}$). Formally, $\boldsymbol{\Gamma}_{t,s,0,0,l,m} = \frac{\partial \llbracket x_t^l \rangle}{\partial \widehat{\llbracket W_0^m x_s^{m-1} \rangle}}, \boldsymbol{\Gamma}_{t,s,0,1,l,m} = \frac{\partial \llbracket x_t^l \rangle}{\partial \widehat{\llbracket W_0^{m\top} \delta x_s^m \rangle}},$

$\boldsymbol{\Gamma}_{t,s,1,0,l,m} = \frac{\partial \llbracket \delta x_t^l \rangle}{\partial \widehat{\llbracket W_0^m x_s^{m-1} \rangle}}, \boldsymbol{\Gamma}_{t,s,1,1,l,m} = \frac{\partial \llbracket \delta x_t^l \rangle}{\partial \widehat{\llbracket W_0^{m\top} \delta x_s^m \rangle}}.$

However, it is hard to describe the limit of $\boldsymbol{\Gamma}$ because its size increases along with $L$. Therefore, we define the following set of functions $\{\Gamma_{t,s,a,b} : [0, 1] \times (0, 1] \to \mathbb{R}\}_{t \in \{0,\dots,T-1\}, s \in \{-1,\dots,t\}, a,b \in \{0,1\}}$: For $s \ge 0$,

$$\Gamma_{t,s,a,b}(p,q) = \sqrt{L} \cdot \boldsymbol{\Gamma}_{t,s,a,b,\lceil Lp \rceil, \lceil Lq \rceil}$$

For $s = -1$, $\Gamma_{t,-1,0,0}(p,q) = \frac{\partial \llbracket x_t^{\lceil Lp \rceil} \rangle}{\partial \llbracket U \rangle}, \Gamma_{t,-1,0,1}(p,q) = \frac{\partial \llbracket x_t^{\lceil Lp \rceil} \rangle}{\partial \llbracket nV \rangle}, \Gamma_{t,-1,1,0}(p,q) = \frac{\partial \llbracket \delta x_t^{\lceil Lp \rceil} \rangle}{\partial \llbracket U \rangle}, \Gamma_{t,-1,1,1}(p,q) = \frac{\partial \llbracket \delta x_t^{\lceil Lp \rceil} \rangle}{\partial \llbracket nV \rangle}.$

Here $l, m$ are normalized to $[0, 1]$ so the input domain of $\Gamma$s are identical for different $L$; $\boldsymbol{\Gamma}_{t,s,a,b,l,m}$ is multiplied by $\sqrt{L}$ because $\boldsymbol{\Gamma}_{t,s,a,b,l,m} = \mathcal{O}(1/\sqrt{L})$ by Proposition C.2; and the extra $s = -1$ case helps us also capture the derivative w.r.t. $\llbracket U \rangle$ and $\llbracket nV \rangle$.

Similarly, we can also define another set of function $\{C_{t,s,a} : (0, 1] \to \mathbb{R}\}_{t,s \in \{-1,\dots,T-1\}, a \in \{0,1\}}$ to describe the covariance between the "base" random variables: $\forall p \in (0, 1]$, let $l = \lceil Lp \rceil$,

- $C_{t,s,0}\,(p) \stackrel{\text{def}}{=} \text{Cov}(\llbracket W_0^l x_t^{l-1}\widehat{\rangle}, \llbracket W_0^l x_s^{l-1}\widehat{\rangle}) = \langle x_t^{l-1} \llbracket x_s^{l-1}\rangle,$

- $C_{t,s,1}\,(p) \stackrel{\text{def}}{=} \text{Cov}(\llbracket W_0^{l\top} \delta x_t^l\widehat{\rangle}, \llbracket W_0^{l\top} \delta x_s^l\widehat{\rangle}) = \langle \delta x_t^l \llbracket \delta x_s^l\rangle,$

For $t = -1$, $C_{-1,-1,0}\,(p) \stackrel{\text{def}}{=} \text{Cov}(\llbracket U\rangle, \llbracket U\rangle) = 1$, and $C_{-1,-1,1}\,(p) \stackrel{\text{def}}{=} \text{Cov}(\llbracket nV\rangle, \llbracket nV\rangle) = 1$, By Definition C.1, the "base" random variables of different "groups" are independent, so we only tracks the covariance listed above.

Using this definition of $\Gamma$ and $C$, it is convenient to write their recursive formula in the following lemma.

**Lemma C.3** (Finite depth recursive formula for $\Gamma$ and $C$ (Informal version of Lemma D.1)). *$\Gamma$ and $C$ can be computed recursively as follows:*

$$\Gamma_{t,r,0,b}\left(\frac{l}{L},q\right) = \Gamma_{t,r,0,b}\left(\frac{l-1}{L},q\right) + \mathbb{I}_{[(t=r)\wedge(b=0)\wedge(l=\lceil Lq\rceil)]}$$
$$+ \frac{1}{L}\sum_{s=0}^{t-1}\Gamma_{s,r,1,b}\left(\frac{l}{L},q\right)\left(\Gamma_{t,s,0,1}\left(\frac{l-1}{L},\frac{l}{L}\right) - C_{t,s,0}\left(\frac{l}{L}\right)\right).$$

$$\Gamma_{t,r,1,b}\left(\frac{l-1}{L},q\right) = \Gamma_{t,r,1,b}\left(\frac{l}{L},q\right) + \mathbb{I}_{[(t=r)\wedge(b=1)\wedge(l=\lceil Lq\rceil)]}$$
$$+ \frac{1}{L}\sum_{s=0}^{t-1}\Gamma_{s,r,0,b}\left(\frac{l-1}{L},q\right)\left(\Gamma_{t,s,1,0}\left(\frac{l}{L},\frac{l}{L}\right) - C_{t,s,1}\left(\frac{l}{L}\right)\right).$$

$$C_{t,s,a}(p) = \sum_{t'=-1}^{t}\sum_{s'=-1}^{s}\sum_{b\in\{0,1\}}\int_0^1 \Gamma_{t,t',a,b}(l/L,q)C_{t',s',b}(q)\Gamma_{s,s',a,b}(l/L,q)\mathrm{d}q,$$

*where $l = \lceil Lp\rceil - 1$ if $a = 0$, and $l = \lceil Lp\rceil$ if $a = 1$.*

The proof of Lemma C.3 is straightforward from Program 1. In Appendix D, we also give a formal proof that $\Gamma$ and $C$ converge when $L$ grows to infinity, in the case where $L$ is powers of 2. The restriction on $L$ being powers of 2 is imposed for the convenience of the proof, and the convergence of $\Gamma$ and $C$ is true in the general case. Moreover, we derive the infinite depth behavior based on the recursion of $\Gamma$ and $C$ in Lemma C.3.

**Proposition C.4** (Infinite depth limit of $\Gamma$ and $C$ (Informal version of Proposition D.2)). *In the limit $L \to \infty$, we have*

$$\Gamma_{t,r,0,b}(p,q) = \mathbb{I}_{[(t=r)\wedge(b=0)\wedge(p\geq q)]} + \int_0^p \sum_{s=0}^{t-1}\Gamma_{s,r,1,b}(p',q)\cdot(\Gamma_{t,s,0,1}(p',p') - C_{t,s,0}(p'))\mathrm{d}p';$$

$$\Gamma_{t,r,1,b}(p,q) = \mathbb{I}_{[(t=r)\wedge(b=1)\wedge(p\leq q)]} + \int_p^1 \sum_{s=0}^{t-1}\Gamma_{s,r,0,b}(p',q)\cdot(\Gamma_{t,s,1,0}(p',p') - C_{t,s,1}(p'))\mathrm{d}p';$$

$$C_{t,s,a}(p) = \sum_{t'=-1}^{t}\sum_{s'=-1}^{s}\sum_{b\in\{0,1\}}\int_0^1 \Gamma_{t,t',a,b}(p,q)C_{t',s',b}(q)\Gamma_{s,s',a,b}(p,q)\mathrm{d}q.$$

The proof of Proposition C.4 follows from Lemma C.3. A rigorous proof requires first showing the existence of a solution of the integral functional satisfied by the couple $(\Gamma, C)$. The solution is typically a fixed point of the integral functional in Proposition C.4. After showing the existence, one needs to show that $(\Gamma, C)$ converges to this limit. This typically requires controlling the difference between finite-depth and infinite-depth solutions and involves obtaining upper-bounds on error propagation. The existence is guaranteed under mild conditions on the integral functional. We omit here the full proof for existence and assume that the functional is sufficiently well-behaved for this convergence result to hold. The formal proof of the convergence of $\Gamma$ and $C$ for $L = 2^k$ ($k \in \mathbb{N}$) in Appendix D is a showcase of the correctness of the proposition.

# D DETAILS OF THE LINEAR CASE

## D.1 PROOF SKETCH OF PROPOSITION C.2

Here we provide a proof sketch of Proposition C.2, the formal prove is implied by the existence of $\Gamma$ and $C$ in the infinite depth limit.

*Proof sketch.* The claims can be reasoned by induction on $t$ and $l$. Let us take $\|x_t^l\rangle$ as an example, since $\|\delta x_t^{l-1}\rangle$ is symmetric with $\|x_t^l\rangle$. By expanding the definition of $\|x_t^l\rangle$, we have

$$\|x_t^l\rangle = \|x_t^{l-1}\rangle + \frac{1}{\sqrt{L}}\|W_0^l x_t^{l-1}\widehat{\rangle} + \frac{1}{\sqrt{L}}\sum_{s=1}^{t-1}\|\delta x_s^l\rangle\left(\frac{\partial\|x_t^{l-1}\rangle}{\partial\|W_0^{l\top}\delta x_s^l\widehat{\rangle}} - \frac{1}{\sqrt{L}}\langle x_s^{l-1}\|x_t^{l-1}\rangle\right).$$

Note by induction, $\langle x_s^{l-1}\|x_t^{l-1}\rangle = \mathcal{O}(1)$ and $\frac{\partial x_t^{l-1}}{\partial\|W_0^{l\top}\delta x_s^l\widehat{\rangle}} = \mathcal{O}(1/\sqrt{L})$, so

$$\|x_t^l\rangle = \|x_t^{l-1}\rangle + \frac{1}{\sqrt{L}}\|W_0^l x_t^{l-1}\widehat{\rangle} + \mathcal{O}\left(\frac{1}{L}\right)\sum_{s=1}^{t-1}\|\delta x_s^l\rangle$$

$$= \xi_t\|U\rangle + \sum_{m=1}^{l}\frac{1}{\sqrt{L}}\|W_0^m x_t^{m-1}\widehat{\rangle} + \mathcal{O}\left(\frac{1}{L}\right)\sum_{m'=1}^{l}\sum_{s'=1}^{t-1}\|\delta x_{s'}^{m'}\rangle.$$

Then by unwinding $\|\delta x_{s'}^{m'}\rangle$ and noting that by induction, $\forall s < t$, $\frac{\partial\|\delta x_{s'}^{m'}\rangle}{\partial\|W_0^m x_s^{m-1}\widehat{\rangle}} = \mathcal{O}\left(\frac{1}{\sqrt{L}}\right)$, $\frac{\partial\|\delta x_{s'}^{m'}\rangle}{\partial\|W_0^{m\top}\delta x_s^m\widehat{\rangle}} = \mathcal{O}\left(\frac{1}{\sqrt{L}}\right)$, $\frac{\partial\|\delta x_{s'}^{m'}\rangle}{\partial\|U\rangle} = \mathcal{O}(1)$, $\frac{\partial\|\delta x_{s'}^{m'}\rangle}{\partial\|nV\rangle} = \mathcal{O}(1)$, we have

$$\frac{\partial\|x_t^l\rangle}{\partial\|W_0^m x_s^{m-1}\widehat{\rangle}} = \mathcal{O}\left(\frac{1}{\sqrt{L}}\right), \frac{\partial\|x_t^l\rangle}{\partial\|W_0^{m\top}\delta x_s^m\widehat{\rangle}} = \mathcal{O}\left(\frac{1}{\sqrt{L}}\right), \frac{\partial\|x_t^l\rangle}{\partial\|U\rangle} = \mathcal{O}(1), \frac{\partial\|x_t^l\rangle}{\partial\|nV\rangle} = \mathcal{O}(1).$$

Also by unwinding, $\forall\|y\rangle \in \{\|x_s^m\rangle, \|\delta x_s^m\rangle\}$,

$$\langle y\|x_t^l\rangle = \sum_{m'}\sum_{s'}\sum_{t'}\frac{\partial\|x_t^l\rangle}{\partial\|W_0^{m'}x_{t'}^{m'-1}\widehat{\rangle}} \cdot \frac{\partial\|y\rangle}{\partial\|W_0^{m'}x_{s'}^{m'-1}\widehat{\rangle}} \cdot \langle x_{t'}^{m'-1}\|x_{s'}^{m'-1}\rangle$$

$$+ \sum_{m'}\sum_{s'}\sum_{t'}\frac{\partial\|x_t^l\rangle}{\partial\|W_0^{m'\top}\delta x_{t'}^{m'}\widehat{\rangle}} \cdot \frac{\partial\|y\rangle}{\partial\|W_0^{m'\top}\delta x_{s'}^{m'}\widehat{\rangle}} \cdot \langle\delta x_{t'}^{m'}\|\delta x_{s'}^{m'}\rangle$$

$$+ \frac{\partial\|x_t^l\rangle}{\partial\|U\rangle} \cdot \frac{\partial\|y\rangle}{\partial\|U\rangle} + \frac{\partial\|x_t^l\rangle}{\partial\|nV\rangle}\frac{\partial\|y\rangle}{\partial\|nV\rangle}$$

$$= \mathcal{O}(1). \qquad\qquad \square$$

## D.2 FORMAL RECURSIVE FORMULA OF $\Gamma$ AND $C$

By the same way of expanding $\|x_t^l\rangle$ and $\langle y\|x_t^l\rangle$, we formally derive the recursive formula for $\Gamma$ and $C$ below.

**Lemma D.1** (Finite depth recursive formula for $\Gamma$ and $C$). *$\Gamma$ can be computed recursively as follows:*

*For $t = 0, \ldots, T-1$,*

- $\forall q \in (0, 1], \Gamma_{t,-1,0,q}(0, q) = \xi_t,$

- *For $l = 1, \ldots, L$, $\forall r \leq t$, $\forall p \in \left(\frac{l-1}{L}, \frac{l}{L}\right]$, $\forall q \in (0, 1]$, $\forall b \in \{0, 1\}$,*

$$C_{t,s,0}(p) = \sum_{t'=-1}^{t} \sum_{s'=-1}^{s} \sum_{b \in \{0,1\}} \int_0^1 \Gamma_{t,t',0,b}\left(\frac{l-1}{L}, q\right) C_{t',s',b}(q) \Gamma_{s,s',0,b}\left(\frac{l-1}{L}, q\right) dq;$$

$$\Gamma_{t,r,0,b}(p, q) = \Gamma_{t,r,0,b}\left(\frac{l-1}{L}, q\right) + \mathbb{I}_{[(t=r)\wedge(b=0)\wedge(l=\lceil Lq \rceil)]}$$

$$+ \frac{1}{L} \sum_{s=0}^{t-1} \Gamma_{s,r,1,b}\left(\frac{l}{L}, q\right) \left(\Gamma_{t,s,0,1}\left(\frac{l-1}{L}, \frac{l}{L}\right) - C_{t,s,0}\left(\frac{l}{L}\right)\right).$$

- $\mathring{f}_t = \Gamma_{t,-1,0,1}(1, 1)$,

- $\mathring{\chi}_t = \ell'_t(\mathring{f}_t)$,

- $\forall q \in (0, 1], \Gamma_{t,-1,1,1}(1, q) = \mathring{\chi}_t$,

- *For $l = L, \ldots, 1$, $\forall r \leq t$, $\forall p \in \left(\frac{l-2}{L}, \frac{l-1}{L}\right]$, $\forall q \in (0, 1]$, $\forall b \in \{0, 1\}$,*

$$C_{t,s,1}\left(p + \frac{1}{L}\right) = \sum_{t'=-1}^{t} \sum_{s'=-1}^{s} \sum_{b \in \{0,1\}} \int_0^1 \Gamma_{t,t',1,b}(l/L, q) C_{t',s',b}(q) \Gamma_{s,s',1,b}(l/L, q) dq;$$

$$\Gamma_{t,r,1,b}(p, q) = \Gamma_{t,r,1,b}\left(\frac{l}{L}, q\right) + \mathbb{I}_{[(t=r)\wedge(b=1)\wedge(l=\lceil Lq \rceil)]}$$

$$+ \frac{1}{L} \sum_{s=0}^{t-1} \Gamma_{s,r,0,b}\left(\frac{l-1}{L}, q\right) \left(\Gamma_{t,s,1,0}\left(\frac{l}{L}, \frac{l}{L}\right) - C_{t,s,1}\left(\frac{l}{L}\right)\right).$$

The proof is straightforward from Program 1. The recursive nature of $\Gamma$ and $C$ yields the following infinite-depth behavior.

**Proposition D.2** (Infinite depth limit of $\Gamma$ and $C$)**.** *In the limit $L \to \infty$, we have $\forall p \in [0, 1], q \in (0, 1], b \in \{0, 1\}$:*

$$\Gamma_{t,-1,0,0}(0, q) = \xi_t;$$

$$\Gamma_{t,r,0,b}(p, q) = \mathbb{I}_{[(t=r)\wedge(b=0)\wedge(p\geq q)]} + \int_0^p \sum_{s=0}^{t-1} \Gamma_{s,r,1,b}(p', q) \cdot (\Gamma_{t,s,0,1}(p', p') - C_{t,s,0}(p')) dp';$$

$$\mathring{f}_t = \Gamma_{t,-1,0,1}(1, 1);$$
$$\mathring{\chi}_t = \ell'_t(\mathring{f}_t);$$
$$\Gamma_{t,-1,1,1}(1, q) = \mathring{\chi}_t;$$

$$\Gamma_{t,r,1,b}(p, q) = \mathbb{I}_{[(t=r)\wedge(b=1)\wedge(p\leq q)]} + \int_p^1 \sum_{s=0}^{t-1} \Gamma_{s,r,0,b}(p', q) \cdot (\Gamma_{t,s,1,0}(p', p') - C_{t,s,1}(p')) dp';$$

$$C_{t,s,a}(p) = \sum_{t'=-1}^{t} \sum_{s'=-1}^{s} \sum_{b \in \{0,1\}} \int_0^1 \Gamma_{t,t',a,b}(p, q) C_{t',s',b}(q) \Gamma_{s,s',a,b}(p, q) dq.$$

## D.3 Convergence of $\Gamma$ and $C$ when $L = 2^k$

In this section, we prove $\Gamma$ and $C$ will converge when $L \to \infty$. For convenience, we will only consider the case when $L = 2^k$ for some integer $k$. To distinguish $\Gamma$ and $C$ corresponding to different $L$, we add the depth as the superscript, i.e., $\Gamma^L$ and $C^L$.

**Theorem D.3.** $\forall t \leq T, s < t, a \in \{0, 1\}, b \in \{0, 1\}, \forall p \in [0, 1], q \in (0, 1]$,

- $\{\Gamma_{t,s,a,b}^{2^k}(p, q)\}_{k \in \mathbb{N}}$ *is a Cauchy sequence,*

- $\{C^{2^k}_{t,s,a}(p)\}_{k\in\mathbb{N}}$ *is a Cauchy sequence.*

The proof is by induction on $t$. We will prove the following claims (A) (B) (C) (D) on $t > 0$ given they are satisfied for any $s < t$. For $t = 0$, (A) (B) (C) (D) are trivial.

**Assumption on $s < t$**   Assume $\exists c > 1$ such that $\forall L > L'$ and $L = 2^k$ for $k \in \mathbb{N}$, $\forall s < t$, $\forall r < s$,

(A) $\forall p \in \{0, \frac{1}{L}, \dots, 1\}, q \in (0, 1]$,

$$|\Gamma^{L/2}_{s,r,a,b}(p,q) - \Gamma^L_{s,r,a,b}(p,q)| \le c/L, \qquad |C^{L/2}_{s,r,a}(p,q) - C^L_{s,r,a}(p,q)| \le c/L.$$

(B) $|\Gamma^L_{s,r,a,b}(p,q)| \le c, |C^L_{s,r,a}(p)| \le c$.

(C) $C^L_{s,r,a}(p)$ is $c$-Lipschitz w.r.t. $p$, and $\Gamma^L_{s,r,a,b}(p,q)$ is $c$-Lipschitz w.r.t. $p$.

(D) $|\Gamma^L_{s,r,0,1}(p-\frac{1}{L},p+\frac{1}{L}) - \Gamma^L_{s,r,0,1}(p-\frac{1}{L},p)| \le c/L, |\Gamma^L_{s,r,1,0}(p,p) - \Gamma^L_{s,r,1,0}(p,p-\frac{1}{L})| \le c/L$.

**Remark**   (A) indicates that $\{\Gamma^{2^k}_{s,r,a,b}\}_k$ and $\{C^{2^k}_{s,r,a}\}_k$ converge. We only care about $r < s$ because $C^L_{s,s,a}$ will never be used, and $\Gamma^L_{s,s,a,b}$ is known: for $p \in \{0, \frac{1}{L}, \dots, 1\}$,

$$\Gamma^L_{s,s,a,b}(p,q) = \mathbb{I}[(a = 0) \wedge (b = 0) \wedge (p \ge q)] + \mathbb{I}[(a = 1) \wedge (b = 1) \wedge (p + 1/L \le q)].$$

**Proof for $t$-th step (the forward pass)**   In the following subsections, we will prove inductively on increasing order of all $L > L'$ and $L = 2^k$, and increasing order of $p \in \{0, 1/L, \dots, 1\}$ that $\forall s < t$,

(D0) $|\Gamma^L_{t,s,0,1}(p, p + \frac{2}{L}) - \Gamma^L_{t,s,0,1}(p, p + \frac{1}{L})| \le c_2 \exp(c_1 p)/L$;

(C0) For $s < t$, $|\Gamma^L_{t,s,0,b}(p,q) - \Gamma^L_{t,s,0,b}(p - \frac{1}{L}, q)| \le tcc_2 \exp(c_1(p - \frac{1}{L}))/L$;

(B0) $|\Gamma^L_{t,s,0,b}(p,q)| \le c_2 \exp(c_1(p - \frac{1}{2L}))$;

(A0) $|\Gamma^{L/2}_{t,s,0,b}(p,q) - \Gamma^L_{t,s,0,b}(p,q)| \le c_3 c_2 \exp(c_1(p - \frac{1}{2L}))/L$;

(C1) $|C^L_{t,s,0}(p + \frac{1}{L}) - C^L_{t,s,0}(p)| \le c_4 c_2 \exp(c_1(p - \frac{1}{L}))/L$;

(B1) $|C^L_{t,s,0}(p + \frac{1}{L})| \le c_2 \exp(c_1 p)$;

(A1) $|C^{L/2}_{t,s,0}(p + \frac{1}{L}) - C^L_{t,s,0}(p + \frac{1}{L})| \le c_5 c_2 \exp(c_1 p)/L$,

where $c_2 = \max\{\xi_t^2, |\xi_t|\}\exp(c_1/2L')$, $c_3 = 3ct$, $c_4 = 4t(t+1)c^2 + 2tc$, $c_5 = c_4 + 1$, $c_1 = c^3 t(4ct + 2c_4 + 29) + tc(3c_4 + 14) + c(2c_4 + 2c)$.

**Proof for $t$-th step (the backward pass)**   Similar bounds also apply to $\Gamma_{t,s,1,b}$ and $C_{t,s,1}$ by induction on decreasing order of $p$.

**Conclusion**   Combining both backward pass and forward pass at time $t$ shows (A)(B)(C)(D) also hold for $s = t$ with a larger (but constant) $c$. Thus, (A)(B)(C)(D) hold for any constant $s$ by induction on training steps.

### D.3.1   $\Gamma^L_{t,s,0,b}(p,q)$ IN FORWARD PASS (PROOF FOR D0, C0, B0, A0)

We first consider

$$\Gamma^L_{t,r,0,b}(p,q) = \Gamma^L_{t,r,0,b}\left(p - \frac{1}{L}, q\right) + \mathbb{I}[(t = r) \wedge (b = 0) \wedge (Lp = \lceil Lq \rceil)]$$

$$+ \frac{1}{L}\sum_{s=0}^{t-1}\Gamma^L_{s,r,1,b}(p,q)\left(\Gamma^L_{t,s,0,1}\left(p - \frac{1}{L}, p\right) - C^L_{t,s,0}(p)\right).$$

**(D0) Difference between** $\Gamma_{t,s,0,1}^L(p, p + \frac{2}{L})$ **and** $\Gamma_{t,s,0,1}^L(p, p + \frac{1}{L})$    Assume $p \geq 1/L$ ($p = 0$ is trivial), let $q = p + 1/L, q' = p + 2/L$, note that $\Gamma_{s,s,1,b}^L(p, q) = \Gamma_{s,s,1,b}^L(p, q')$ since $p + 1/L \leq q \leq q'$, so for $r < t$,

$$|\Gamma_{t,r,0,b}^L(p, q) - \Gamma_{t,r,0,b}^L(p, q')|$$

$$\leq |\Gamma_{t,r,0,b}^L\left(p - \frac{1}{L}, q\right) - \Gamma_{t,r,0,b}^L\left(p - \frac{1}{L}, q'\right)|$$

$$+ \frac{1}{L} \sum_{s=0}^{t-1} |\Gamma_{s,r,1,b}^L(p, q) - \Gamma_{s,r,1,b}^L(p, q')| \left| \Gamma_{t,s,0,1}^L\left(p - \frac{1}{L}, p\right) - C_{t,s,0}^L(p)\right|$$

$$\leq c_2 \exp(c_1(p - \frac{1}{L}))/L + \frac{1}{L} \cdot t \cdot c/L \cdot 2c_2 \exp(c_1(p - \frac{1}{L}))$$

$$= (1 + 2ct/L)c_2 \exp(c_1(p - \frac{1}{L}))/L \leq c_2 \exp(c_1 p)/L,$$

as $c_1 \geq 2ct$.

**(C0) Lipschitz w.r.t.** $p$    For $r < t$,

$$|\Gamma_{t,r,0,b}^L(p, q) - \Gamma_{t,r,0,b}^L(p - \frac{1}{L}, q)|$$

$$= |\frac{1}{L} \sum_{s=0}^{t-1} \Gamma_{s,r,1,b}^L(p, q) \left(\Gamma_{t,s,0,1}^L\left(p - \frac{1}{L}, p\right) - C_{t,s,0}^L(p)\right)|$$

$$\leq \frac{1}{L} \sum_{s=0}^{t-1} c \left(c_2 \exp(c_1(p - \frac{1}{L})) + c_2 \exp(c_1(p - \frac{1}{L}))\right)$$

$$= ctc_2 \exp(c_1(p - \frac{1}{L}))/L.$$

**(B0) Bounded**    Again assume $p \geq 1/L$ ($p = 0$ is trivial because $c_2 \geq |\xi_t| \exp(c_1/2L)$), since $|\Gamma_{t,r,0,b}^L(p - \frac{1}{L}, q)| \leq c_2 \exp(c_1(p - \frac{1}{L}))$, we can bound $|\Gamma_{t,r,0,b}^L(p, q)|$:

$$|\Gamma_{t,r,0,b}^L(p, q)| \leq c_2 \exp(c_1(p - \frac{1}{L})) + ctc_2 \exp(c_1(p - \frac{1}{L}))/L$$

$$= c_2 \exp(c_1(p - \frac{1}{L}))(1 + ct/L)$$

$$\leq c_2 \exp(c_1(p - \frac{1}{2L})),$$

as long as $c_1 \geq 2ct$.

**(A0) Difference between** $L$ **and** $L/2$ **bounded**    When $p = 0$, it is trivial. When $p = 1/L$, it is also trivial by Lipschitz w.r.t. $p$, which results

$$|\Gamma_{t,r,0,b}^{L/2}(p, q) - \Gamma_{t,r,0,b}^L(p, q)| \leq 3ctc_2/L \leq c_3 c_2 \exp(c_1/2L)/L.$$

When $p \geq 2/L$, since

$$\Gamma_{t,r,0,b}^{L/2}(p, q) = \Gamma_{t,r,0,b}^{L/2}\left(p - \frac{2}{L}, q\right) + \frac{2}{L} \sum_{s=0}^{t-1} \Gamma_{s,r,1,b}^{L/2}(p, q) \left(\Gamma_{t,s,0,1}^{L/2}\left(p - \frac{2}{L}, p\right) - C_{t,s,0}^{L/2}(p)\right),$$

we compare it with $\Gamma_{t,r,0,b}^L(p, q)$ expanded based on its previous two steps

$$\Gamma_{t,r,0,b}^L(p, q) = \Gamma_{t,r,0,b}^L\left(p - \frac{2}{L}, q\right) + \frac{1}{L} \sum_{s=0}^{t-1} \Gamma_{s,r,1,b}^L(p, q) \left(\Gamma_{t,s,0,1}^L\left(p - \frac{1}{L}, p\right) - C_{t,s,0}^L(p)\right)$$

$$+ \frac{1}{L} \sum_{s=0}^{t-1} \Gamma_{s,r,1,b}^L\left(p - \frac{1}{L}, q\right) \left(\Gamma_{t,s,0,1}^L\left(p - \frac{2}{L}, p - \frac{1}{L}\right) - C_{t,s,0}^L\left(p - \frac{1}{L}\right)\right).$$

In order to bridge the two above, namely matching the inputs for $\Gamma$ and $C$, we need a middle term

$$\widetilde{\Gamma}^L_{t,r,0,b}(p,q) = \Gamma^L_{t,r,0,b}\left(p - \frac{2}{L}, q\right) + \frac{2}{L}\sum_{s=0}^{t-1}\Gamma^L_{s,r,1,b}(p,q)\left(\Gamma^L_{t,s,0,1}\left(p - \frac{2}{L}, p\right) - C^L_{t,s,0}(p)\right).$$

Now we can bound $|\Gamma^L_{t,r,0,b}(p,q) - \widetilde{\Gamma}^L_{t,r,0,b}(p,q)|$, and $|\widetilde{\Gamma}^L_{t,r,0,b}(p,q) - \Gamma^{L/2}_{t,r,0,b}(p,q)|$ separately, which add up to be the bound for $|\Gamma^L_{t,r,0,b}(p,q) - \Gamma^{L/2}_{t,r,0,b}(p,q)|$.

$$|\Gamma^L_{t,r,0,b}(p,q) - \widetilde{\Gamma}^L_{t,r,0,b}(p,q)|$$

$$\leq \frac{1}{L}\sum_{s=0}^{t-1}|\Gamma^L_{s,r,1,b}(p,q)|\left|\Gamma^L_{t,s,0,1}\left(p - \frac{1}{L}, p\right) - \Gamma^L_{t,s,0,1}\left(p - \frac{2}{L}, p\right)\right|$$

$$+ \frac{1}{L}\sum_{s=0}^{t-1}\left|\Gamma^L_{s,r,1,b}\left(p - \frac{1}{L}, q\right)\left(\Gamma^L_{t,s,0,1}\left(p - \frac{2}{L}, p - \frac{1}{L}\right) - C^L_{t,s,0}\left(p - \frac{1}{L}\right)\right)\right.$$

$$\left. - \Gamma^L_{s,r,1,b}(p,q)\left(\Gamma^L_{t,s,0,1}\left(p - \frac{2}{L}, p\right) - C^L_{t,s,0}(p)\right)\right|$$

$$\leq \frac{1}{L}\cdot ct \cdot ctc_2\exp(c_1(p - \frac{2}{L}))/L + \frac{1}{L}\cdot 2t\cdot c/L\cdot c_2\exp(c_1(p - \frac{2}{L}))$$

$$+ \frac{1}{L}\cdot ct\cdot c_2\exp(c_1(p - \frac{2}{L}))/L + \frac{1}{L}\cdot ct\cdot c_4 c_2\exp(c_1(p - \frac{2}{L}))/L$$

$$= \frac{c^2t^2 + 3ct + c_4 ct}{L^2}\cdot c_2\exp(c_1(p - \frac{2}{L})).$$

and

$$|\Gamma^{L/2}_{t,r,0,b}(p,q) - \widetilde{\Gamma}^L_{t,r,0,b}(p,q)|$$

$$\leq |\Gamma^{L/2}_{t,r,0,b}\left(p - \frac{2}{L}, q\right) - \Gamma^L_{t,r,0,b}\left(p - \frac{2}{L}, q\right)|$$

$$+ \frac{1}{L}\sum_{s=0}^{t-1}c\left|\Gamma^{L/2}_{t,s,0,1}\left(p - \frac{2}{L}, p\right) - \Gamma^L_{t,s,0,1}\left(p - \frac{2}{L}, p\right) - C^{L/2}_{t,s,0}(p) + C^L_{t,s,0}(p)\right|$$

$$+ \frac{1}{L}\sum_{s=0}^{t-1}\frac{c}{L}\left(|\Gamma^L_{t,s,0,1}\left(p - \frac{2}{L}, p\right)| + |C^L_{t,s,0}(p)|\right)$$

$$\leq \frac{1}{L}\cdot (c_3 c_2\exp(c_1(p - \frac{1}{L})) + ct(c_3 + c_5)c_2\exp(c_1(p - \frac{1}{L}))/L + 2t\cdot\frac{c}{L}\cdot c_2\exp(c_1(p - \frac{1}{L})))$$

$$\leq \frac{c_3 + ct(c_3 + c_5 + 2)/L}{L}\cdot c_2\exp(c_1(p - \frac{1}{L})).$$

In sum, as $c_1 \geq \frac{2(c_3 + c_5 + ct + c_4 + 5)}{3}$,

$$|\Gamma^{L/2}_{t,r,0,b}(p,q) - \Gamma^L_{t,r,0,b}(p,q)| \leq \frac{c_3 + ct(c_3 + c_5 + ct + c_4 + 5)/L}{L}c_2\exp(c_1(p - \frac{1}{L}))$$

$$\leq c_2\exp(c_1(p - \frac{1}{2L}))/L.$$

### D.3.2 $C_{t,s,0}(p + \frac{1}{L})$ IN FORWARD PASS (PROOF FOR C1, B1, A1)

Now consider $C^L_{t,s,0}$. By expanding

$$C^L_{t,s,0}(p + \frac{1}{L}) = \sum_{t'=-1}^{t}\sum_{s'=-1}^{s}\sum_{b\in\{0,1\}}\int_0^1\Gamma^L_{t,t',0,b}(p,q)C^L_{t',s',b}(q)\Gamma^L_{s,s',0,b}(p,q)\mathrm{d}q,$$

we will have

$$C_{t,s,0}^L(p + \frac{1}{L}) = \sum_{t'=-1}^{t-1} \sum_{s'=-1}^{s} \sum_{b \in \{0,1\}} \int_0^1 \Gamma_{t,t',0,b}^L(p,q) C_{t',s',b}^L(q) \Gamma_{s,s',0,b}^L(p,q) dq$$
$$+ \sum_{s'=0}^{s} \int_0^p C_{t,s',0}^L(q) \Gamma_{s,s',0,0}^L(p,q) dq.$$

**(C1) Lipschitz**   Since $C_{t',s',b}^L$ and $\Gamma_{s,s',0,b}^L$ are bounded and Lipschitz,

$$|C_{t,s,0}^L(p + \frac{1}{L}) - C_{t,s,0}^L(p)|$$

$$\leq \sum_{t'=-1}^{t-1} \sum_{s'=-1}^{s} \sum_{b \in \{0,1\}} \int_0^1 |\Gamma_{t,t',0,b}^L(p,q) - \Gamma_{t,t',0,b}^L(p - \frac{1}{L},q)| \cdot c^2 dq$$

$$+ \sum_{t'=-1}^{t-1} \sum_{s'=-1}^{s} \sum_{b \in \{0,1\}} \int_0^1 |\Gamma_{t,t',0,b}^L(p - \frac{1}{L},q)| \cdot c \cdot \frac{c}{L} dq$$

$$+ \sum_{s'=0}^{s} \frac{1}{L} \cdot |C_{t,s',0}^L(p) \Gamma_{s,s',0,0}^L(p,p)|$$

$$+ \sum_{s'=0}^{s} \int_0^{p - \frac{1}{L}} |C_{t,s',0}^L(q)| \cdot \frac{c}{L} dq.$$

$$\leq 1/L \cdot (2t(s+1) \cdot ctc_2 \exp(c_1(p - \frac{1}{L})) \cdot c^2 + 2t(s+1)c_2 \exp(c_1(p - \frac{1}{L})) \cdot c^2$$

$$+ s \cdot c_2 \exp(c_1(p - \frac{1}{L})) \cdot c + s \cdot c_2 \exp(c_1(p - \frac{1}{L})) \cdot c)$$

$$= (4t(s+1)c^2 + 2sc)/L \cdot c_2 \exp(c_1(p - \frac{1}{L}))$$

$$\leq c_4 c_2 \exp(c_1(p - \frac{1}{L}))/L.$$

**(B1) Bounded**   Since $|C_{t,s,0}^L(p)| \leq c_2 \exp(c_1(p - \frac{1}{L}))$, we bound $C_{t,s,0}^L(p + \frac{1}{L})$ as:

$$|C_{t,s,0}^L(p + \frac{1}{L})| \leq c_2 \exp(c_1(p - \frac{1}{L})) \cdot (1 + c_4/L) \leq c_2 \exp(c_1 p),$$

as long as $c_1 \geq c_4$.

**(A1) Difference between $L$ and $L/2$ bounded**   It is easy to see that for $p = 0$,

$$C_{t,s,0}^L(p + \frac{1}{L}) - C_{t,s,0}^{L/2}(p + \frac{1}{L}) = 0,$$

we will prove that for $p \in \{2/L, 4/L, \ldots, 1\}$,

$$|C_{t,s,0}^L(p + \frac{1}{L}) - C_{t,s,0}^{L/2}(p + \frac{1}{L})| \leq c_2 \exp(c_1 p)/L.$$

Then by (C1), for $p \in \{1/L, 3/L, \ldots, 1 - 1/L\}$,

$$|C_{t,s,0}^L(p + \frac{1}{L}) - C_{t,s,0}^{L/2}(p + \frac{1}{L})| \leq c_2 \exp(c_1(p - \frac{1}{L}))/L + c_4 c_2 \exp(c_1(p - \frac{1}{L}))/L$$

$$\leq (c_4 + 1)c_2 \exp(c_1 p)/L$$

$$= c_5 c_2 \exp(c_1 p)/L.$$

Suppose $p \in \{2/L, 4/L, \ldots, 1\}$, we compare $C_{t,s,0}^L(p + \frac{1}{L}) - C_{t,s,0}^L(p - \frac{1}{L})$ and $C_{t,s,0}^{L/2}(p + \frac{1}{L}) - C_{t,s,0}^{L/2}(p - \frac{1}{L})$. Intuitively, both of them are $\mathcal{O}(1/L)$, and their difference is $\mathcal{O}(1/L^2)$. In particular,

both of them can be written into four parts:

$$C_{t,s,0}^L(p + \frac{1}{L}) - C_{t,s,0}^L(p - \frac{1}{L})$$

$$= \sum_{t'=-1}^{t-1} \sum_{s'=-1}^{s} \sum_{b\in\{0,1\}} \int_0^1 \left(\Gamma_{t,t',0,b}^L(p,q) - \Gamma_{t,t',0,b}^L(p - \frac{2}{L}, q)\right) C_{t',s',b}^L(q)\Gamma_{s,s',0,b}^L(p,q)\mathrm{d}q \qquad (\mathcal{E}_1^L)$$

$$+ \sum_{t'=-1}^{t-1} \sum_{s'=-1}^{s} \sum_{b\in\{0,1\}} \int_0^1 \Gamma_{t,t',0,b}^L(p - \frac{2}{L}, q)C_{t',s',b}^L(q)\left(\Gamma_{s,s',0,b}^L(p,q) - \Gamma_{s,s',0,b}^L(p - \frac{2}{L}, q)\right)\mathrm{d}q \quad (\mathcal{E}_2^L)$$

$$+ \sum_{s'=0}^{s} \int_{p-\frac{2}{L}}^{p} C_{t,s',0}^L(q)\Gamma_{s,s',0,0}^L(p,q)\mathrm{d}q \qquad (\mathcal{E}_3^L)$$

$$+ \sum_{s'=0}^{s} \int_0^{p-\frac{2}{L}} C_{t,s',0}^L(q)(\Gamma_{s,s',0,0}^L(p,q) - \Gamma_{s,s',0,0}^L(p - \frac{2}{L}, q))\mathrm{d}q \qquad (\mathcal{E}_4^L)$$

and $C_{t,s,0}^{L/2}(p + \frac{1}{L}) - C_{t,s,0}^{L/2}(p - \frac{1}{L}) = \mathcal{E}_1^{L/2} + \mathcal{E}_2^{L/2} + \mathcal{E}_3^{L/2} + \mathcal{E}_4^{L/2}$ where $\mathcal{E}_i^{L/2}$ is defined in the same way as $\mathcal{E}_i^L$ but with $C^{L/2}$ and $\Gamma^{L/2}$ instead of $C^L$ and $\Gamma^L$. Next we bound $|\mathcal{E}_i^L - \mathcal{E}_i^{L/2}|$ one by one:

1. The only hard part to bound in $|\mathcal{E}_i^L - \mathcal{E}_i^{L/2}|$ is

$$|\Gamma_{t,t',0,b}^L(p,q) - \Gamma_{t,t',0,b}^L(p - \frac{2}{L}, q) - (\Gamma_{t,t',0,b}^{L/2}(p,q) - \Gamma_{t,t',0,b}^{L/2}(p - \frac{2}{L}, q))|.$$

   By almost the same proof of (A0),

$$|\Gamma_{t,t',0,b}^L(p,q) - \Gamma_{t,t',0,b}^L(p - \frac{2}{L}, q) - (\Gamma_{t,t',0,b}^{L/2}(p,q) - \Gamma_{t,t',0,b}^{L/2}(p - \frac{2}{L}, q))|$$

$$\leq \frac{ct(c_3 + c_5 + ct + c_4 + 5)}{L^2} c_2 \exp(c_1(p - \frac{1}{L})).$$

   Then we have

$$|\mathcal{E}_1^L - \mathcal{E}_1^{L/2}|/(2t(s+1))$$

$$\leq \frac{ct(c_3 + c_5 + ct + c_4 + 5)}{L^2} c_2 \exp(c_1(p - \frac{1}{L})) \cdot c \cdot c$$

$$+ 4ctc_2 \exp(c_1(p - \frac{1}{L}))/L \cdot c/L \cdot c$$

$$+ 4ctc_2 \exp(c_1(p - \frac{1}{L}))/L \cdot c \cdot c/L$$

$$\leq \frac{c^3 t(c_3 + c_5 + ct + c_4 + 13)}{L^2} c_2 \exp(c_1(p - \frac{1}{L}))$$

2. Bounding $|\mathcal{E}_2^L - \mathcal{E}_2^{L/2}|$ is similar to $|\mathcal{E}_1^L - \mathcal{E}_1^{L/2}|$, where we first bound

$$|\Gamma_{s,s',0,b}^L(p,q) - \Gamma_{s,s',0,b}^L(p - \frac{2}{L}, q) - (\Gamma_{s,s',0,b}^{L/2}(p,q) - \Gamma_{s,s',0,b}^{L/2}(p - \frac{2}{L}, q))| \leq 9c^2 t/L^2.$$

   Then we have

$$|\mathcal{E}_2^L - \mathcal{E}_2^{L/2}|/(2t(s+1))$$

$$\leq c_3 c_2 \exp(c_1(p - \frac{2}{L}))/L \cdot c \cdot 2c/L$$

$$+ c_2 \exp(c_1(p - \frac{2}{L})) \cdot c/L \cdot 2c/L$$

$$+ c_2 \exp(c_1(p - \frac{2}{L})) \cdot c \cdot 9c^2 t/L^2$$

$$\leq \frac{c^2(2c_3 + 2 + 9ct)}{L^2} c_2 \exp(c_1(p - \frac{2}{L})).$$

3. For $|\mathcal{E}_3^L - \mathcal{E}_3^{L/2}|$, we first simplify

$$\mathcal{E}_3^{L/2} = \frac{2}{L} \sum_{s'=0}^{s} C_{t,s',0}^{L/2}(p) \Gamma_{s,s',0,0}^{L/2}(p,p),$$

and

$$\mathcal{E}_3^L = \frac{1}{L} \sum_{s'=0}^{s} C_{t,s',0}^L(p) \Gamma_{s,s',0,0}^L(p,p) + C_{t,s',0}^L(p - \frac{1}{L}) \Gamma_{s,s',0,0}^L(p, p - \frac{1}{L}).$$

Again, we introduce an intermediate term

$$\widetilde{\mathcal{E}}_3^L = \frac{2}{L} \sum_{s'=0}^{s} C_{t,s',0}^L(p) \Gamma_{s,s',0,0}^L(p,p).$$

Then we can bound

$$\begin{aligned}
&|\mathcal{E}_3^L - \mathcal{E}_3^{L/2}| \\
&\leq |\mathcal{E}_3^L - \widetilde{\mathcal{E}}_3^L| + |\widetilde{\mathcal{E}}_3^L - \mathcal{E}_3^{L/2}| \\
&\leq \frac{t}{L}(c_4 c_2 \exp(c_1(p - \frac{2}{L}))/L \cdot c + c_2 \exp(c_1(p - \frac{2}{L})) \cdot c/L) \\
&\quad + \frac{2t}{L}(c_5 c_2 \exp(c_1(p - \frac{1}{L}))/Lc + c_2 \exp(c_1(p - \frac{1}{L})) \cdot c/L) \\
&\leq \frac{tc(c_4 + 1 + 2c_5 + 2)}{L^2} c_2 \exp(c_1(p - \frac{1}{L})).
\end{aligned}$$

4. For $|\mathcal{E}_4^L - \mathcal{E}_4^{L/2}|$, we use

$$|\Gamma_{s,s',0,b}^L(p,q) - \Gamma_{s,s',0,b}^L(p - \frac{2}{L}, q) - (\Gamma_{s,s',0,b}^{L/2}(p,q) - \Gamma_{s,s',0,b}^{L/2}(p - \frac{2}{L}, q))| \leq 9c^2 t/L^2,$$

which is used in $|\mathcal{E}_2^L - \mathcal{E}_2^{L/2}|$. Finally,

$$\begin{aligned}
&|\mathcal{E}_4^L - \mathcal{E}_4^{L/2}|/t \\
&\leq c_4 c_2 \exp(c_1(p - \frac{2}{L}))/L \cdot 2c/L \\
&\quad + c_2 \exp(c_1(p - \frac{2}{L})) \cdot 9c^2 t/L^2 \\
&\leq \frac{c(2c_4 + 9ct)}{L^2} c_2 \exp(c_1(p - \frac{2}{L})).
\end{aligned}$$

In sum,

$$\begin{aligned}
&|C_{t,s,0}^L(p + \frac{1}{L}) - C_{t,s,0}^L(p - \frac{1}{L}) - C_{t,s,0}^{L/2}(p + \frac{1}{L}) + C_{t,s,0}^{L/2}(p - \frac{1}{L})| \\
&\leq \frac{c^3 t(4ct + 2c_4 + 14) + c^2(2 + 15ct) + tc(3c_4 + 5) + c(2c_4 + 9ct)}{L^2} \cdot c_2 \exp(c_1(p - \frac{1}{L})) \\
&= \frac{c^3 t(4ct + 2c_4 + 29) + tc(3c_4 + 14) + c(2c_4 + 2c)}{L^2} \cdot c_2 \exp(c_1(p - \frac{1}{L})).
\end{aligned}$$

Therefore, since $c_1 = c^3 t(4ct + 2c_4 + 29) + tc(3c_4 + 14) + c(2c_4 + 2c)$,

$$\begin{aligned}
&|C_{t,s,0}^L(p + \frac{1}{L}) - C_{t,s,0}^{L/2}(p + \frac{1}{L})| \\
&\leq |C_{t,s,0}^L(p - \frac{1}{L}) - C_{t,s,0}^{L/2}(p - \frac{1}{L})| + c_1/L^2 \cdot c_2 \exp(c_1(p - \frac{1}{L})) \\
&\leq (1 + c_1/L)c_2 \exp(c_1(p - \frac{1}{L}))/L \\
&\leq c_2 \exp(c_1 p)/L.
\end{aligned}$$

# E    NOTATIONS FOR THE GENERAL CASE

This section provides an introduction to the new TP notations from Yang & Littwin (2023). We only require the definition of the inner and outer products in this paper.

**Averaging over** $n$    When $x \in \mathbb{R}^n$, we always use greek subscript $\alpha, \beta, \ldots \in [n]$ to index its entries. Then $\langle x_\alpha \rangle_\alpha$ denotes its average entry. This notation will only be used to average over $n$-dimensions, but not over constant dimensions.

## E.1    THE TENSOR PROGRAM ANSATZ: REPRESENTING VECTORS VIA RANDOM VARIABLES

From the Tensor Programs framework (Yang et al., 2022), we know that as width becomes large, the entries of the (pre-)activation vectors and their gradients will become roughly iid, both at initialization and training. Hence any such vector's behavior can be tracked via a random variable that reflects the distribution of its entries. While we call this the "Tensor Program Ansatz", it is a completely rigorous calculus.

### E.1.1    KET NOTATION

Concretely, if $x \in \mathbb{R}^n$ is one such vector, then we write $\|x\rangle \in \mathbb{R}$ (called a *ket*) for such a random variable, such that $x$'s entries look like iid samples from $\|x\rangle$. For any two such vectors $x, y \in \mathbb{R}^n$, $(x_\alpha, y_\alpha) \in \mathbb{R}^2$ for each $\alpha$ will look like iid samples from the random vector $(\|x\rangle, \|y\rangle)$, such that, for example, $\lim_{n\to\infty} \frac{x^\top y}{n} = \mathbb{E}\,\|x\rangle \cdot \|y\rangle$, which we write succinctly as just $\langle x\|y\rangle$. Here $\langle x\|$ is called a *bra*, interpreted as a sort of "transpose" to $\|x\rangle$. In our convention, $\|x\rangle$ is always a random variable independent of $n$ and $x$ always has $\Theta(1)$ typical entry size.[17]

This notation can be generalized to the case where $\boldsymbol{x} \in \mathbb{R}^{n \times k}, \boldsymbol{y} \in \mathbb{R}^{n \times j}$. In this case, we can think of $\langle \boldsymbol{x}\|\boldsymbol{y}\rangle$ as the $k \times j$ matrix given by $(\langle x_a\|y_b\rangle)_{\substack{1 \le a \le k \\ 1 \le b \le j}}$.

Because we will often need to multiply a ket with a diagonal matrix, we introduce a shorthand:

$$\|\boldsymbol{x}\rangle_{\boldsymbol{\chi}} = \|\boldsymbol{x}\rangle \mathrm{Diag}(\boldsymbol{\chi}), \tag{4}$$

if $\boldsymbol{x}$ is $n \times k$ and $\boldsymbol{\chi}$ is a $k$-dimensional vector.

### E.1.2    OUTER PRODUCT

Likewise, if both $\boldsymbol{x}$ and $\boldsymbol{y}$ have shape $n \times k$, the expression

$$\|\boldsymbol{x}\rangle\langle\boldsymbol{y}\| \quad \text{represents the limit of } \boldsymbol{x}\boldsymbol{y}^\top \in \mathbb{R}^{n \times n}.$$

More formally, $\|\boldsymbol{x}\rangle\langle\boldsymbol{y}\|$ is defined as an operator that takes a ket $\|\boldsymbol{z}\rangle \in \mathbb{R}^j$ and return the ket

$$(\|\boldsymbol{x}\rangle\langle\boldsymbol{y}\|)\|\boldsymbol{z}\rangle = \|\boldsymbol{x}\rangle(\langle\boldsymbol{y}\|\boldsymbol{z}\rangle) \in \mathbb{R}^j$$

i.e., it returns the random vector $\|\boldsymbol{x}\rangle \in \mathbb{R}^k$ multiplied by the deterministic matrix $\langle\boldsymbol{y}\|\boldsymbol{z}\rangle \in \mathbb{R}^{k \times j}$ on the right. This corresponds to the limit of $\boldsymbol{x}\boldsymbol{y}^\top \boldsymbol{z}/n$. Likewise, $\|\boldsymbol{x}\rangle\langle\boldsymbol{y}\|$ acts on a bra $\langle\boldsymbol{w}\| \in \mathbb{R}^j$ by

$$\langle\boldsymbol{w}\|(\|\boldsymbol{x}\rangle\langle\boldsymbol{y}\|) = (\langle\boldsymbol{w}\|\boldsymbol{x}\rangle)\langle\boldsymbol{y}\| \in \mathbb{R}^j.$$

which corresponds to the limit of $\frac{1}{n}\boldsymbol{w}^\top \boldsymbol{x}\boldsymbol{y}^\top$. This definition of $\|\boldsymbol{x}\rangle\langle\boldsymbol{y}\|$ makes the expressions

$$\|\boldsymbol{x}\rangle\langle\boldsymbol{y}\|\boldsymbol{z}\rangle, \quad \langle\boldsymbol{w}\|\boldsymbol{x}\rangle\langle\boldsymbol{y}\|, \quad \langle\boldsymbol{w}\|\boldsymbol{x}\rangle\langle\boldsymbol{y}\|\boldsymbol{z}\rangle$$

unambiguous (since any way of ordering the operations give the same answer).

*Remark* E.1 (Potential Confusion).  One should *not* interpret $\|\boldsymbol{x}\rangle\langle\boldsymbol{y}\|$ as the scalar random variable $\|\boldsymbol{x}\rangle \cdot \|\boldsymbol{y}\rangle = \sum_{i=1}^k \|x^i\rangle\|y^i\rangle$, which would act on a ket $\|\boldsymbol{z}\rangle$ to produce $(\langle\boldsymbol{x}\| \cdot \|\boldsymbol{y}\rangle)\|\boldsymbol{z}\rangle = \mathbb{E}(\|\boldsymbol{x}\rangle \cdot \|\boldsymbol{y}\rangle)\|\boldsymbol{z}\rangle$, which is deterministic. On the other hand, $\|\boldsymbol{x}\rangle\langle\boldsymbol{y}\|\boldsymbol{z}\rangle$ is always a linear combination of $\|\boldsymbol{x}\rangle$, a nondeterministic random variable in general. In particular, any correlation between $\|\boldsymbol{x}\rangle$ and $\|\boldsymbol{y}\rangle$ does not directly play a role in their outer product $\|\boldsymbol{x}\rangle\langle\boldsymbol{y}\|$: we always have $\|\boldsymbol{x}\rangle\langle\boldsymbol{y}\|\boldsymbol{z}\rangle = \|\boldsymbol{x}\rangle\langle\boldsymbol{y}\|^{\boxed{1}}\|\boldsymbol{z}\rangle^{\boxed{1}}$, where $(\|\boldsymbol{y}\rangle^{\boxed{1}}, \|\boldsymbol{z}\rangle^{\boxed{1}})$ is an iid copy of $(\|\boldsymbol{y}\rangle, \|\boldsymbol{z}\rangle)$ independent from $\|\boldsymbol{x}\rangle$.

---

[17] i.e., $\|x\|^2/n = \Theta(1)$ as $n \to \infty$

**Outer Product with Diagonal Inserted** Finally, if $\chi \in \mathbb{R}^k$ is deterministic, then (consistent with eq. (4)) we define $\|x\rangle_{\chi}\langle y\|$ as the operator that acts on kets $\|z\rangle \in \mathbb{R}^j$ by

$$(\|x\rangle_{\chi}\langle y\|)\|z\rangle = \|x\rangle_{\chi}\langle y\|z\rangle = \|x\rangle\mathrm{Diag}(\chi)(\langle y\|z\rangle) \in \mathbb{R}^j.$$

Morally, $\|x\rangle_{\chi}\langle y\|$ is just a shorter way of writing $\|x\rangle\mathrm{Diag}(\chi)\langle y\|$ and represents the limit of $x\mathrm{Diag}(\chi)y^{\top}$. In particular, $\|x\rangle_{\mathbf{1}}\langle y\| = \|x\rangle\langle y\|$.

### E.1.3 Nonlinear Outer Product

If $xy^{\top} \in \mathbb{R}^{n \times n}$ is the (linear) outer product of two vectors $x \in \mathbb{R}^n$ and $y \in \mathbb{R}^n$, then $\phi(xy^{\top})$, the entrywise application of nonlinear $\phi : \mathbb{R} \to \mathbb{R}$ to $xy^{\top}$, is a kind of *nonlinear outer product*. Passing to the ket notation, in general we define $\phi(\|x\rangle\langle y\|)$ as the operator that acts on kets as

$$\phi(\|x\rangle\langle y\|)\|z\rangle \overset{\text{def}}{=} \underset{\boxed{1}}{\mathbb{E}}\, \phi\left(\sum_{i=1}^{k} \|x^i\rangle\|y^i\rangle^{\boxed{1}}\right)\|z\rangle^{\boxed{1}}$$

where $\left(\|y^1\rangle^{\boxed{1}}, \dots, \|y^k\rangle^{\boxed{1}}, \|z\rangle^{\boxed{1}}\right)$ is an iid copy of $\left(\|y^1\rangle, \dots, \|y^k\rangle, \|z\rangle\right)$ independent from $\|x\rangle$ and the expectation is taken only over the former. This is just like, in the finite $n$ case,

$$\phi\left(xy^{\top}\right)z/n = \phi\left(\sum_{i=1}^{k} x^i y^{i\top}\right)z/n.$$

Moreover, if $\|w\rangle \in \mathbb{R}^j, \|z\rangle \in \mathbb{R}^k$, then

$$\langle w\|\phi(\|x\rangle\langle y\|)\|z\rangle = \langle w\|\phi\left(\|x\rangle\langle y\|^{\boxed{1}}\right)\|z\rangle^{\boxed{1}} \in \mathbb{R}^{j \times k}$$

$$= \mathbb{E}\, \phi\left(\sum_{i=1}^{k} \|x^i\rangle\|y^i\rangle^{\boxed{1}}\right)\left(\|w\rangle \otimes \|z\rangle^{\boxed{1}}\right)$$

where $\otimes$ denotes outer product of vectors and expectation is taken over everything.

More generally, if $\phi : \mathbb{R}^t \to \mathbb{R}$, then $\phi\left(\|x_1\rangle\langle y_1\|, \dots, \|x_t\rangle\langle y_t\|\right)$ is an operator taking kets to kets, defined by

$$\phi\left(\|x_1\rangle\langle y_1\|, \dots, \|x_t\rangle\langle y_t\|\right)\|z\rangle \overset{\text{def}}{=} \underset{\boxed{1}}{\mathbb{E}}\, \phi\left(\sum_{i=1}^{k} \|x_1^i\rangle\|y_1^i\rangle^{\boxed{1}}, \dots, \sum_{i=1}^{k} \|x_t^i\rangle\|y_t^i\rangle^{\boxed{1}}\right)\|z\rangle^{\boxed{1}}$$

*Remark* E.2 (Potential Confusion). Note $\phi(\|x\rangle\langle y\|)$ is not the image of the operator $\|x\rangle\langle y\|$ under $\phi$ in the continuous function calculus of operators, but rather a "coordinatewise application" of $\phi$. For example, if $\phi(t) = t^2$, then $\phi(\|x\rangle\langle y\|)$ *is not* $\|x\rangle\langle y\|x\rangle\langle y\|$, the latter being what typically "squaring an operator" means, but rather $\|x\rangle^2\langle y\|^2 = \|x \odot x\rangle\langle y \odot y\|$.

### E.1.4 Comparison with Previous $Z^{\bullet}$ Notation

For readers familiar with the *Tensor Programs* papers, this new "bra-ket" notation (aka Dirac notation) relates to the old $Z^{\bullet}$ notation by

$$\|x\rangle = Z^x, \quad \langle x\|y\rangle = \mathbb{E}\, Z^x Z^y.$$

The new notation's succinctness of expectation inner product should already be apparent. Furthermore, the old notation is not very compatible with multi-vectors whereas $\|x\rangle$ makes it clear that $\rangle$ represents the constant dimension side. Consequently, (nonlinear) outer product is awkward to express in it, especially when its contraction with random variables requires an explicit expectation symbol $\mathbb{E}$.

## F Infinite-Width Limit with the Bra-ket notation

As before, when the width $n$ of the program goes to infinity, one can infer how the program behaves via a calculus of random variables. We define them below via the new ket notation instead of the earlier $Z$ notation.

**Ket Construction.** We recursively define the random variable $[\![x\rangle$ (called a *ket*) for each vector $x$ and deterministic number $\mathring{\theta}$ for each scalar $\theta$ in the program. For a vector $Wx$ in the program, we also define random variables $[\![W\widehat{x}\rangle$ and $[\![W\dot{x}\rangle$ (called *hat-ket* and *dot-ket* respectively) such that $[\![Wx\rangle = [\![W\widehat{x}\rangle + [\![W\dot{x}\rangle$. These are the same as $\widehat{Z}$ and $\dot{Z}$ in the old TP notation (Yang et al., 2022) and they satisfy

Hat  All hat-kets are jointly Gaussian with zero-mean and covariance[18]

$$\mathrm{Cov}([\![W\widehat{x}\rangle, [\![U\widehat{y}\rangle) = \mathbb{I}(W = U)\langle x[\![y\rangle \tag{5}$$

Dot  Every dot-ket is a linear combination of previous kets, expressed by the following equation

$$[\![W\dot{x}\rangle \overset{\text{def}}{=} \sum_{y \in \boldsymbol{x}} [\![y\rangle \, \mathbb{E} \, \frac{\partial [\![x\rangle}{\partial [\![W^\top \widehat{y}\rangle} \tag{6}$$

eq. (6) is the same equation as in (Yang et al., 2022, Zdot) but formulated much more succinctly in the bra-ket notation:

$$\text{(Yang et al., 2022, Zdot),} \quad \dot{Z}^{Wx} = \sum_{y \in \boldsymbol{x}} Z^y \, \mathbb{E} \, \frac{\partial Z^x}{\partial \widehat{Z}^{W^\top y}}.$$

There is an alternative notion for $[\![W\dot{x}\rangle$ in Yang & Littwin (2023) that write

$$[\![W\dot{x}\rangle = [\![\boldsymbol{x}\rangle\langle W^\top \boldsymbol{x}[\![x\rangle.$$

This is more convenient to write as we introduce the operator view.

We can see the ket $[\![Wx\rangle$ as the result of the action of an operator on the ket $[\![x\rangle$.

**Definition F.1.** Let $W$ be an initial matrix in a Tensor Program. We define $[\![W[\!], \widehat{[\![W[\!]}, \dot{[\![W[\!]}$ to be the linear operators on kets [19] that act by

$$\widehat{[\![W[\!]}[\![x\rangle \overset{\text{def}}{=} [\![W\widehat{x}\rangle$$
$$\dot{[\![W[\!]}[\![x\rangle \overset{\text{def}}{=} [\![W\dot{x}\rangle$$
$$[\![W[\!][\![x\rangle \overset{\text{def}}{=} \widehat{[\![W[\!]}[\![x\rangle + \dot{[\![W[\!]}[\![x\rangle.$$

Any linear operator that is equal to $[\![W[\!]$ for some initial matrix $W$ is called an *initial operator*.

We also define the adjoint relations between the operators:

$$\widehat{[\![W[\!]}^\dagger = \dot{[\![W^\top[\!]},$$
$$\dot{[\![W[\!]}^\dagger = \widehat{[\![W^\top[\!]},$$
$$[\![W[\!]^\dagger = [\![W^\top[\!].$$

**Parameter Update**  In the SGD case, the parameter update of $W^l$ is simple. With the operator notation and outer product notation, we can write

$$[\![W_{t+1}^l[\!] = [\![W_t^l[\!] - \eta [\![\delta h_t^l\rangle_{\chi_t}\langle x_t^{l-1}[\!].$$

---

[18] In eq. (5), $\mathbb{I}(W = U)$ is the deterministic number that is 1 iff $W$ and $U$ are the same matrix (as symbols in the program) and 0 otherwise. This should *not* be interpreted as a random variable that is 1 precisely when $W$ and $U$ take the same values.

[19]  To be rigorous, we need to specify the "Hilbert space" of kets. This is somewhat pedantic and not crucial to the key points of this paper, but the Hilbert space can be constructed as follows: Let $\sigma(\pi)$ be the $\sigma$-algebra generated by the kets of the program $\pi$. Let $\Sigma(\pi) \overset{\text{def}}{=} \bigcup_{\pi' \supseteq \pi} \sigma(\pi)$ be the union (more precisely, the direct limit) of $\sigma(\pi')$ over all programs $\pi'$ extending $\pi$. Then the Hilbert space in question is the $L^2$ space of random variables over the $\Sigma$ of our program.

In this work, $\Delta$ denotes change for one step, i.e.,

$$[\![ \Delta W_{t+1}^l ]\!] = -\eta [\![ \delta h_t^l \rangle_{\chi_t} \langle x_t^{l-1} ]\!];$$

$\bar{\Delta}$ denotes total change, i.e.,

$$[\![ \bar{\Delta} W_t^l ]\!] = -\sum_{\tau=0}^{t-1} \eta [\![ \delta h_\tau^l \rangle_{\chi_\tau} \langle x_\tau^{l-1} ]\!],$$

which we write succinctly $[\![ \bar{\Delta} W_t^l ]\!] = -\eta [\![ \delta \boldsymbol{h}_{<t}^l \rangle_{\boldsymbol{\chi}} \langle \boldsymbol{x}_{<t}^{l-1} ]\!]$. (Compared to Yang & Littwin (2023), $\Delta$ and $\bar{\Delta}$ are changed from $\delta$ and $\Delta$ because we want to use $\delta$ for gradients instead of $d$, which is now used for depth differentiation).

Note in the general case,

$$[\![ \Delta W_{t+1}^l ]\!] = -\eta \overline{[\![ \delta \boldsymbol{h}_{\leq t}^l \rangle_{\boldsymbol{\chi}_{\leq t}} \langle \boldsymbol{x}_{\leq t}^{l-1} ]\!]}$$

where

$$\overline{[\![ \delta \boldsymbol{h}_{\leq t}^l \rangle_{\boldsymbol{\chi}_{\leq t}} \langle \boldsymbol{x}_{\leq t}^{l-1} ]\!]} \overset{\text{def}}{=} Q_t^l([\![ \delta h_0^l \rangle_{\chi_0} \langle x_0^{l-1} ]\!], \dots, [\![ \delta h_t^l \rangle_{\chi_t} \langle x_t^{l-1} ]\!]).$$

So

$$[\![ \bar{\Delta} W_t^l ]\!] = -\eta \sum_{\tau=0}^{t-1} \overline{[\![ \delta \boldsymbol{h}_{\leq \tau}^l \rangle_{\boldsymbol{\chi}_{\leq \tau}} \langle \boldsymbol{x}_{\leq \tau}^{l-1} ]\!]}. \tag{7}$$

For the rest of the paper, we write $[\![ \bar{\Delta} W_t^l ]\!] = -\eta [\![ \delta \boldsymbol{h}_{<t}^l \rangle_{\boldsymbol{\chi}} \langle \boldsymbol{x}_{<t}^{l-1} ]\!]$ for convenience. The generalization to eq. (7) follows Yang & Littwin (2023).

## G  HEURISTICS FOR THE PROOFS IN THE GENERAL CASE

The notation in this section is mostly defined in appendix E. The complete notation is defined in Yang & Littwin (2023).

### G.1  DEPTH-$\mu$P

Let $\text{MS}(x) = x - \langle x, 1 \rangle / n = Gx$ where $G = I - 11^\top/n$, where $x \in \mathbb{R}^n$. Recall the definition of the network and the normalized gradients

$$
\begin{aligned}
x^1 &= U\xi \\
h^l &= W^l x^{l-1} \\
x^l &= x^{l-1} + \frac{1}{\sqrt{L}} G\phi(h^l) \\
f(\xi) &= V^\top x^L \\
\delta x^L &= nV \\
\delta h^l &= \phi'(h^l) \odot (G\delta x^l) \\
\delta x^{l-1} &= \delta x^l + \frac{1}{\sqrt{L}} W^{l\top} \delta h^l
\end{aligned}
$$

where $V = \Theta(1/n)$ coordinatewise, $\delta x^l = \Theta(1)$ coordinatewise and $W^l = \Theta(\frac{1}{\sqrt{n}})$ coordinatewise.

We also abuse the notation of $G$ and use it as an operator on kets: $G[\![ x \rangle \overset{\text{def}}{=} [\![ x \rangle - \mathbb{E}[\![ x \rangle$.

**Forward.** Similar to the linear case, one can show that under technical conditions (mostly on the activation function) that the infinite-depth limit of the TP follows the dynamics

$$d\llbracket x_t^\lambda\rangle = \sqrt{d\lambda}\, G\phi\left(\llbracket W_0^\lambda\rrbracket x_t^\lambda\rangle + \sqrt{d\lambda}\llbracket\widetilde{\overline{\Delta W_t^\lambda}}\rrbracket x_t^\lambda\rangle\right)$$

$$= \sqrt{d\lambda}\, G\phi\left(\llbracket W_0^\lambda\rrbracket x_t^\lambda\rangle\right) + d\lambda\, G\phi'\left(\llbracket W_0^\lambda\rrbracket x_t^\lambda\rangle\right)\llbracket\widetilde{\overline{\Delta W_t^\lambda}}\rrbracket x_t^\lambda\rangle$$

$$= \sqrt{d\lambda}\, G\phi\left(\llbracket\widehat{W_0^\lambda}\widehat{\rrbracket x_t^\lambda}\rangle + \llbracket \dot{W_0^\lambda}\rrbracket x_t^\lambda\rangle\right) + d\lambda\, G\phi'\left(\llbracket W_0^\lambda\rrbracket x_t^\lambda\rangle\right)\llbracket\widetilde{\overline{\Delta W_t^\lambda}}\rrbracket x_t^\lambda\rangle$$

$$= \sqrt{d\lambda}\, G\phi\left(\llbracket\widehat{W_0^\lambda}\widehat{\rrbracket x_t^\lambda}\rangle\right) + d\lambda\, G\phi'\left(\llbracket W_0^\lambda\rrbracket x_t^\lambda\rangle\right)\left(\llbracket\widetilde{\dot{W_0^\lambda}}\rrbracket x_t^\lambda\rangle + \llbracket\widetilde{\overline{\Delta W_t^\lambda}}\rrbracket x_t^\lambda\rangle\right)$$

where $\lambda \in [0,1]$ refers to the fractional layer index ($\lambda$ represents layer index $\lfloor \lambda L\rfloor$ as $L \to \infty$), $t$ refers to the training step, $\llbracket W_0^\lambda\rrbracket$ the matrix operator (defined in Appendix F), and the tilde symbol refers to the "normalized" version of the object, i.e., multiply the ket with $(d\lambda)^c$ for some $c$ such that the multiplication (normalized ket) is $\Theta(1)$ w.r.t. $L$, and same for the normalized operators. The first term represents a Gaussian noise.

In the linear case, we have

$$d\llbracket x_t^\lambda\rangle = \sqrt{d\lambda}\left(\llbracket\widehat{W_0^\lambda}\widehat{\rrbracket x_t^\lambda}\rangle\right) + d\lambda\left(\llbracket\widetilde{\dot{W_0^\lambda}}\rrbracket x_t^\lambda\rangle + \llbracket\widetilde{\overline{\Delta W_t^\lambda}}\rrbracket x_t^\lambda\rangle\right)$$

Note

$$\llbracket\dot{W_0^\lambda}\rrbracket \dot{x}_t^\lambda\rangle = \sqrt{d\lambda}\sum_{s=0}^{t-1}\llbracket\widetilde{\delta h_s^\lambda}\rangle\langle\nabla_{W_0^{\lambda\top}\delta h_s^\lambda}\llbracket x_t^\lambda\rangle = \sqrt{d\lambda}\llbracket\widetilde{\delta h_{<t}^\lambda}\rangle\check{\langle}W_0^{\lambda\top}\delta h_{<t}^\lambda\llbracket x_t^\lambda\rangle$$

Using multi-vector notation, we write

$$\llbracket\widetilde{\overline{\Delta W_t^\lambda}}\rrbracket x_t^\lambda\rangle = -\eta\llbracket\widetilde{\delta \boldsymbol{h}_{<t}^\lambda}\rangle_{\boldsymbol{\chi}}\langle\boldsymbol{x}_{<t}^\lambda\llbracket x_t^\lambda\rangle = -\eta\sum_{s<t}\llbracket\widetilde{\delta\boldsymbol{h}_s^\lambda}\rangle_{\chi_s}\langle x_s^\lambda\llbracket x_t^\lambda\rangle$$

$$\llbracket\overline{\Delta W_t^\lambda}\rrbracket x_t^\lambda\rangle = -\eta\llbracket\delta\boldsymbol{h}_{<t}^\lambda\rangle_{\boldsymbol{\chi}}\langle\boldsymbol{x}_{<t}^\lambda\llbracket x_t^\lambda\rangle = -\eta\sum_{s<t}\llbracket\delta h_s^\lambda\rangle_{\chi_s}\langle x_s^\lambda\llbracket x_t^\lambda\rangle$$

**Backward.** Similar to the forward prop, we obtain the following dynamics for the infinite-depth TP

$$-d\llbracket\delta x_\tau^\lambda\rangle = \sqrt{d\lambda}\llbracket W_\tau^{\lambda\top}\rrbracket\phi'(W_\tau^\lambda x_\tau^\lambda)\odot(G\delta x_\tau^\lambda)\rangle$$

$$= \sqrt{d\lambda}\left(\llbracket\widehat{W_0^{\lambda\top}}\widehat{\rrbracket} + \llbracket\dot{W_0^{\lambda\top}}\rrbracket + \sqrt{d\lambda}\llbracket\widetilde{\overline{\Delta W_\tau^\lambda}}\rrbracket^\dagger\right)\left[\phi'\left(\llbracket W_0^\lambda x_\tau^{\widehat\lambda}\rangle + \sqrt{d\lambda}\llbracket\widetilde{\dot{W_0^\lambda}x_\tau^\lambda}\rangle + \sqrt{d\lambda}\llbracket\widetilde{\overline{\Delta W_\tau^\lambda}}\rrbracket x_\tau^\lambda\rangle\right)\llbracket G\delta x_\tau^\lambda\rangle\right]$$

$$= \sqrt{d\lambda}\llbracket\widehat{W_0^{\lambda\top}}\widehat{\rrbracket}\left[\phi'(\llbracket W_0^\lambda x_\tau^{\widehat\lambda}\rangle)\llbracket G\delta x_\tau^\lambda\rangle\right] + \sqrt{d\lambda}\llbracket W_0^{\lambda\top}\dot\rrbracket\left[\phi'(\llbracket W_0^\lambda x_\tau^{\widehat\lambda}\rangle)\llbracket G\delta x_\tau^\lambda\rangle\right] + d\lambda\llbracket\widetilde{\overline{\Delta W_\tau^\lambda}}\rrbracket^\dagger\left[\phi'(\llbracket W_0^\lambda x_\tau^{\widehat\lambda}\rangle)\llbracket G\delta x_\tau^\lambda\rangle\right]$$

$$+ d\lambda\llbracket\dot{W_0^{\lambda\top}}\rrbracket\left[\phi''\left(\llbracket W_0^\lambda x_\tau^{\widehat\lambda}\rangle\right)\left\{\llbracket\widetilde{\dot{W_0^\lambda}x_\tau^\lambda}\rangle + \llbracket\widetilde{\overline{\Delta W_\tau^\lambda}}\rrbracket x_\tau^\lambda\rangle\right\}\llbracket G\delta x_\tau^\lambda\rangle\right]$$

Here the $(d\lambda)^{3/2}$ term got dropped. The individual terms can be simplified as follows

$$\llbracket\widetilde{\overline{\Delta W_\tau^\lambda}}\rrbracket^\dagger\left[\phi'(\llbracket W_0^\lambda x_\tau^{\widehat\lambda}\rangle)\llbracket G\delta x_\tau^\lambda\rangle\right] = -\eta\llbracket x_{<\tau}^\lambda\rangle_{\boldsymbol\chi}\langle\widetilde{\delta\boldsymbol{h}_{<\tau}^\lambda}\llbracket\left[\phi'(\llbracket W_0^\lambda x_\tau^{\widehat\lambda}\rangle)\llbracket G\delta x_\tau^\lambda\rangle\right] \approx -\eta\llbracket x_{<\tau}^\lambda\rangle_{\boldsymbol\chi}\langle\widetilde{\delta\boldsymbol{h}_{<\tau}^\lambda}\llbracket\widetilde{\delta h_\tau^\lambda}\rangle$$

$$\llbracket\dot{W_0^{\lambda\top}}\rrbracket\left[\phi'(\llbracket W_0^\lambda x_\tau^{\widehat\lambda}\rangle)\llbracket G\delta x_\tau^\lambda\rangle\right] = \left[\llbracket\boldsymbol{x}_{<\tau}^\lambda\rangle\check{\langle}W_0^\lambda\boldsymbol{x}_{<\tau}^\lambda\llbracket + \llbracket\boldsymbol{x}_\tau^\lambda\rangle\check{\langle}W_0^\lambda\boldsymbol{x}_\tau^\lambda\llbracket\right]\left[\phi'(\llbracket W_0^\lambda x_\tau^{\widehat\lambda}\rangle)\llbracket G\delta x_\tau^\lambda\rangle\right]$$

$$= \llbracket\boldsymbol{x}_{<\tau}^\lambda\rangle\,\mathbb{E}\left[\phi'(\llbracket W_0^\lambda x_\tau^{\widehat\lambda}\rangle)\frac{\partial\llbracket G\delta x_\tau^\lambda\rangle}{\partial\llbracket W_0^\lambda\boldsymbol{x}_{<\tau}^{\widehat\lambda}\rangle}\right] + \llbracket\boldsymbol{x}_\tau^\lambda\rangle\,\mathbb{E}\left[\phi''(\llbracket W_0^\lambda x_\tau^{\widehat\lambda}\rangle)\llbracket G\delta x_\tau^\lambda\rangle\right]$$

$$= \Theta(\sqrt{d\lambda})$$

where the other terms from the product rule drops out because

$$\frac{\partial\phi'(\llbracket W_0^\lambda x_\tau^{\widehat\lambda}\rangle)}{\partial\llbracket W_0^\lambda\boldsymbol{x}_{<\tau}^{\widehat\lambda}\rangle} = \frac{\partial\llbracket G\delta x_\tau^\lambda\rangle}{\partial\llbracket W_0^\lambda x_\tau^{\widehat\lambda}\rangle} = 0$$

### G.2 $1/L$ BRANCHES

#### G.2.1 FORWARD:

$$d\llbracket x_t^\lambda \rangle = d\lambda G \, \mathbb{E} \left[ \phi \left( \llbracket W_0^\lambda \rrbracket x_t^\lambda \rangle + \llbracket \bar{\Delta} W_t^\lambda \rrbracket x_t^\lambda \rangle \right) \mid \llbracket U_0, V_0 \rangle \right]$$

$$= d\lambda G \, \mathbb{E} \left[ \phi \left( \widehat{\llbracket W_0^\lambda \rrbracket} x_t^\lambda \rangle + \llbracket \bar{\Delta} W_t^\lambda \rrbracket x_t^\lambda \rangle \right) \mid \llbracket U_0, V_0 \rangle \right]$$

where the equality follows because $\llbracket x_t^\lambda \rangle$ is contained the $\sigma$-algebra of $\llbracket U_0, V_0 \rangle$, so $\mathring{\llbracket} W_0^\lambda \mathring{\rrbracket} x_t^\lambda \rangle = 0$. Since $\llbracket \bar{\Delta} W_t^\lambda \rrbracket \in \sigma(\llbracket U_0, V_0 \rangle) \otimes \sigma(\llbracket U_0, V_0 \rangle)$, $\llbracket \bar{\Delta} W_t^\lambda \rrbracket x_t^\lambda \rangle \in \sigma(\llbracket U_0, V_0 \rangle)$, and the expectation is really just over $\widehat{\llbracket W_0^\lambda \rrbracket} x_t^\lambda \rangle$.

#### G.2.2 BACKWARD

$$-d\llbracket \delta x_\tau^\lambda \rangle = d\lambda \, \mathbb{E} \left[ \llbracket W_\tau^{\lambda\top} \rrbracket \phi'(W_\tau^\lambda x_\tau^\lambda) \odot (G \delta x_\tau^\lambda) \rangle \mid \llbracket U_0, V_0 \rangle \right]$$

$$= d\lambda \, \mathbb{E} \left[ \llbracket \bar{\Delta} W_\tau^{\lambda\top} \rrbracket \phi'(W_\tau^\lambda x_\tau^\lambda) \odot (G \delta x_\tau^\lambda) \rangle \mid \llbracket U_0, V_0 \rangle \right]$$

Here the $\widehat{\llbracket W_0^{\lambda\top} \rrbracket}$ and $\mathring{\llbracket} W_0^{\lambda\top} \mathring{\rrbracket}$ drop out because the former is zero-mean and indepenent from $\llbracket U_0, V_0 \rangle$ and the latter drops out because $\llbracket x_t^\lambda \rangle$ is contained the $\sigma$-algebra of $\llbracket U_0, V_0 \rangle$.

### G.3 $1/L^\alpha$ BRANCHES, $\alpha \in (1/2, 1]$

#### G.3.1 FORWARD

$$d\llbracket x_t^\lambda \rangle = (d\lambda)^\alpha G \, \mathbb{E} \left[ \phi \left( \llbracket W_0^\lambda \rrbracket x_t^\lambda \rangle + (d\lambda)^{1-\alpha} \widetilde{\llbracket \bar{\Delta} W_t^\lambda \rrbracket} x_t^\lambda \rangle \right) \mid \llbracket U_0, V_0 \rangle \right]$$

$$= d\lambda \, \mathbb{E} \left[ \phi' \left( \widehat{\llbracket W_0^\lambda \rrbracket} x_t^\lambda \rangle \right) \right] G \llbracket \bar{\Delta} W_t^\lambda \rrbracket x_t^\lambda \rangle$$

because the same reason as above.

#### G.3.2 BACKWARD

$$-d\llbracket \delta x_\tau^\lambda \rangle = d\lambda \, \mathbb{E} \left[ \llbracket W_\tau^{\lambda\top} \rrbracket \phi'(W_\tau^\lambda x_\tau^\lambda) \odot (G \delta x_\tau^\lambda) \rangle \mid \llbracket U_0, V_0 \rangle \right]$$

$$= d\lambda \, \mathbb{E} \left[ \llbracket \bar{\Delta} W_\tau^{\lambda\top} \rrbracket \phi'(W_\tau^\lambda x_\tau^\lambda) \odot (G \delta x_\tau^\lambda) \rangle \mid \llbracket U_0, V_0 \rangle \right]$$

$$= d\lambda \, \mathbb{E} \left[ \phi'(\llbracket W_0^\lambda x_\tau^\lambda \widehat{\rangle}) \right] \llbracket \bar{\Delta} W_\tau^{\lambda\top} \rrbracket G \delta x_\tau^\lambda \rangle$$

Here,

$$\llbracket \phi'(W_\tau^\lambda x_\tau^\lambda) \odot (G \delta x_\tau^\lambda) \rangle \equiv \mathbb{E} \left[ \phi'(\llbracket W_0^\lambda x_\tau^\lambda \widehat{\rangle}) \right] G \delta x_\tau^\lambda \rangle + (d\lambda)^{1-\alpha} \mathbb{E} \left[ \phi''(\llbracket W_0^\lambda x_\tau^\lambda \widehat{\rangle}) \right] G \llbracket \bar{\Delta} W_\tau^{\lambda\top} \rrbracket \delta x_\tau^\lambda \rangle$$

### G.4 JUSTIFICATIONS OF THE CLAIMS

**Claim 5.2.** Stability during training when $\alpha + \gamma \geq 1$ is straightforward (some technical conditions on the activation function are required). This is because the weight updates are of order $L^{-\alpha-\gamma}$ and feature updates involve no more than $L$ terms of size $L^{-\alpha-\gamma}$ (plus higher order terms that do not contribute to the update in the large depth limit). When $\alpha + \gamma > 1$, the contribution of a sum of at most $L$ terms of order $L^{-\alpha-\gamma}$ will decrease to zero, and the network output $f_t$ will converge to $f_0$ in this case, yielding a trivial limit. However, when $\alpha + \gamma = 1$, the updates remain important in the infinite depth limit, yielding a non-trivial limit.

**Claim 5.3.** Consider a stable and nontrivial parametrization (i.e. $\alpha + \gamma = 1$). Faithfullness at initialization is achieved only when $\alpha \geq 1/2$. This was proven in Hayou et al. (2021) in a more general setup. Faithfullness during training is ensured as long as $\alpha \leq 1$ because feature updates are always $\Theta(1)$ in depth. With $\alpha > 1$, $\gamma < 0$ and the weight updates explode with depth in this case,

which yield exploding behaviour for $\boldsymbol{h}$.

**Claim 5.4** When $\alpha \in (1/2, 1]$, we obtain smooth limiting dynamics when $L \to \infty$ as demonstrated in Appendix G.3. This limiting process is a smooth process (no Brownian jumps) that satisfies the required definition of redundancy.

**Claim 5.5.** It remains to prove that Depth-$\mu$P is non-redundant. This is a result of the limiting dynamics in this case (Appendix G.1) . With Depth-$\mu$P, the randomness of the initialization in the hidden layer remains present throughout training, inducing a Brownian-like term that breaks redundancy.

### G.5 SUBLETY: LAYERWISE (LOCAL) LINEARIZATION BUT NOT GLOBAL LINEARIZATION

**Definition G.1.** We say a parametrization induces layerwise linearization iff each layer can be linearized without changing the network output when $L \to \infty$, that is, $\forall l \in [L]$,

$$L^{-\alpha} G \left( \phi(W_t^l \boldsymbol{x}_t^{l-1}) - \phi(W_0^l \boldsymbol{x}_t^{l-1}) - \phi'(W_0^l \boldsymbol{x}_t^{l-1}) \odot ((W_t^l - W_0^l)\boldsymbol{x}_t^{l-1}) \right) = o(L^{-1})$$

**Claim G.1.** *A stable and nontrivial parametrization induces layerwise linearization iff $\alpha \in [1/2, 1)$.*

In Depth-$\mu$P, $W_t^l - W_0^l$ is $\Theta(1/\sqrt{L})$ which is much smaller than $W_0^l$. Therefore, $\phi(W_t^l \boldsymbol{x}_t^{l-1}) - \phi(W_0^l \boldsymbol{x}_t^{l-1}) - \phi'(W_0^l \boldsymbol{x}_t^{l-1}) \odot ((W_t^l - W_0^l)\boldsymbol{x}_t^{l-1}) = o(1/\sqrt{L})$, thus satisfies Definition G.1. Similar to the depth-$\mu$P case, for $\alpha \in [1/2, 1)$, the activation in the forward pass can be linearized which indicates layerwise linearization when $\alpha + \gamma = 1$.

However, note that this does not imply the entire network is linearized (w.r.t. all the parameters in the sense of Neural Tangent Kernel). In our setup, where the widthwise scaling follows $\mu$P and the input/output layers are initialized at a constant scale (w.r.t. $L$), it is actually not possible to have a kernel limit. Even in our linear case in Section 3.2, one can see the learned model is not linear w.r.t. the parameters.

If the initialization of the output layer is $L$ times larger than our setup (assuming $L \ll n$ so the widthwise scaling still follows $\mu$P), it may induce a parametrization that can linearize the entire network. In that situation, the learning rate has to be $L$ times smaller than Depth-$\mu$P to obtain stability during training, so the change of parameters is also $L$ times smaller, which can lead to the linearization of the entire network. Since we focus on maximal feature learning, the rigorous argument is beyond the scope of this paper.

# H CLASSIFICATION OF DEPTHWISE PARAMETRIZATIONS IN LINEAR CASE

We discuss the classification results on the linear residual networks with SGD training and give rigorous proofs for the claims in this simplified setting. Recall the linear residual networks:

$$\forall l \in [L], x^l = x^{l-1} + aL^{-\alpha}h^l,$$

where $h^l = W^l x^l$, and the effective learning rate of $W^l$ is $\eta n^{-1} L^{-\gamma}$. Without loss of generality, we assume $\eta = a = 1$.

## H.1 INITIALIZATION

At initialization, we have

$$[\![x_0^l\rangle = [\![x_0^{l-1}\rangle + L^{-\alpha}[\![h_0^l\rangle,$$

where

$$[\![h_0^l\rangle = [\![W_0^l x_0^{l-1}\rangle = [\![W_0^l \widehat{x_0^{l-1}}\rangle.$$

Since $[\![x_0^{l-1}\rangle$ is independent from $[\![W_0^l \widehat{x_0^{l-1}}\rangle$, we have

$$\langle x_0^l [\![ x_0^l \rangle = \langle x_0^{l-1}[\![x_0^{l-1}\rangle + L^{-2\alpha}\langle h_0^l[\![h_0^l\rangle = \langle x_0^{l-1}[\![x_0^{l-1}\rangle + L^{-2\alpha}\langle x_0^{l-1}[\![x_0^{l-1}\rangle = (1+L^{-2\alpha})\langle x_0^{l-1}[\![x_0^{l-1}\rangle.$$

Using this recursion, we can write

$$\langle x_0^l [\![ x_0^l \rangle = (1 + L^{-2\alpha})^l \langle x_0^0 [\![ x_0^0 \rangle.$$

Therefore, $\langle x_0^L [\![ x_0^L \rangle = \Theta(1)$ iff $\alpha \geq 1/2$, otherwise $(1 + L^{-2\alpha})^L \approx e^{L^{-2\alpha+1}}$ explodes with large $L$.

A similar argument stands for $h_0^l$ and $f_0$. Therefore, we have proved Claim 5.1.

Similarly, we can get the stability of the first backward pass, i.e., $\widetilde{\delta x}_0^l = \Theta(1)$ for $\alpha \geq 1/2$. Given $\alpha \geq 1/2$, we can also settle the size of $\widetilde{\delta h}_0$ that

$$\widetilde{\delta h}_0^l = \Theta(L^{-\alpha}),$$

which implies

$$\Delta W_1^l = L^{-\gamma+\alpha} \cdot \widetilde{\delta h}_0^l \otimes x_0^{l-1}.$$

## H.2 AFTER THE FIRST STEP OF GRADIENT UPDATE

Now we look at the second forward pass, and assume the input is the same, i.e., $[\![x_1^0\rangle = [\![x_0^0\rangle$, we have

$$[\![x_1^l\rangle = [\![x_1^{l-1}\rangle + L^{-\alpha}([\![W_0^l \widehat{x_1^{l-1}}\rangle + [\![W_0^l x_1^{l-1}\rangle + [\![\Delta W_1^l [\![x_1^{l-1}\rangle))$$

where $[\![\Delta W_1^l[\![ = -L^{-\gamma}[\![\widetilde{\delta h}_0^l\rangle \langle x_0^{l-1}[\![ = -L^{-\gamma}[\![\widetilde{\delta x}_0^l\rangle \langle x_0^{l-1}[\![$, and $[\![\widetilde{\delta h}_0^l\rangle \stackrel{\text{def}}{=} L^\alpha [\![\widetilde{\delta h}_0^l\rangle$ is the normalized version of $[\![\widetilde{\delta h}_0^l\rangle$, which happens to equal to $[\![\widetilde{\delta x}_0^l\rangle$. By the definition of $[\![W_0^l \widehat{x_1^{l-1}}\rangle$ and $[\![W_0^l x_1^{l-1}\rangle$, we get a similar formula to the Depth-$\mu$P case:

$$[\![x_1^l\rangle = [\![x_1^{l-1}\rangle + L^{-\alpha}[\![W_0^l \widehat{x_1^{l-1}}\rangle + L^{-\alpha}[\![\widetilde{\delta x}_0^l\rangle \left( \frac{\partial [\![x_1^{l-1}\rangle}{\partial [\![W_0^{l\top}\widetilde{\delta x}_0^l\rangle} - L^{-\gamma}\langle x_0^{l-1}[\![x_1^{l-1}\rangle \right).$$

Now we write $b^l = L^\gamma \frac{\partial [\![x_1^{l-1}\rangle}{\partial [\![W_0^{l\top}\widetilde{\delta x}_0^l\rangle}$ and $c^l = -\langle x_0^{l-1}[\![x_1^{l-1}\rangle$, then

$$[\![x_1^l\rangle = [\![x_1^{l-1}\rangle + L^{-\alpha}[\![W_0^l \widehat{x_1^{l-1}}\rangle + L^{-\alpha-\gamma}(b^l + c^l)[\![\widetilde{\delta x}_0^l\rangle.$$

By expanding $[\![\widetilde{\delta x}_0^{l-1}\rangle = [\![\widetilde{\delta x}_0^l\rangle + L^{-\alpha}[\![W_0^{l\top}\widehat{\widetilde{\delta x}_0^l}\rangle = [\![\widetilde{\delta x}_0^L\rangle + \sum_{m=l}^L L^{-\alpha}[\![W_0^{m\top}\widehat{\widetilde{\delta x}_0^m}\rangle$, we have

$$[\![x_1^l\rangle = [\![x_1^{l-1}\rangle + L^{-\alpha}[\![W_0^l \widehat{x_1^{l-1}}\rangle + L^{-\alpha-\gamma}(b^l + c^l)\left( [\![\widetilde{\delta x}_0^L\rangle + \sum_{m=l+1}^L L^{-\alpha}[\![W_0^{m\top}\widehat{\widetilde{\delta x}_0^m}\rangle \right)$$

$$= [\![x_1^0\rangle + \sum_{m=1}^l L^{-\alpha}[\![W_0^m \widehat{x_1^{m-1}}\rangle + \sum_{m=1}^l L^{-\alpha-\gamma}(b^m + c^m)[\![\widetilde{\delta x}_0^L\rangle$$

$$+ \sum_{m=2}^L L^{-\alpha-\gamma} \sum_{l'=1}^{\min\{m-1,l\}} (b^{l'} + c^{l'})L^{-\alpha}[\![W_0^{m\top}\widehat{\widetilde{\delta x}_0^m}\rangle. \tag{8}$$

Note the four terms in eq. (8) are independent of each other.

Now it is easy to compute $c^l$ because only the first two terms in eq. (8) have correlation with $x_0^l$:

$$c^l = c^{l-1}(1 + L^{-2\alpha}) = \Theta(1)$$

with $\alpha \geq 1/2$. For $b^l$, we have the following recursive formula:

$$b^{l+1} = L^{-2\alpha} \sum_{m=1}^{l} (b^l + c^l) = \Theta(l \cdot L^{-2\alpha}).$$

**Stable during training and nontrivial.** Finally, we can reason about the $\mathring{f}_1$ (note $\mathring{f}_0 = 0$, so $\Delta \mathring{f}_1 = \mathring{f}_1$), which indicates whether the parametrization is stable during the first step[20], and whether the parametrization is nontrivial for the first step:

$$\mathring{f}_1 = \langle nV \| x_1^L \rangle = \sum_{m=1}^{L} L^{-\alpha-\gamma}(b^m + c^m)\chi_0 = \Theta(L^{1-\alpha-\gamma}).$$

Therefore, we have proved Claim 5.2 that the parametrization is stable during training iff $\alpha + \gamma \geq 1$, and is nontrivial iff $\alpha + \gamma \leq 1$.

**Faithfulness.** Although there is no activation in the linear case, we still prove Claim 5.3 to enlighten the proof of the general case.

At the initialization, $h_0^l$ and $x_0^{l-1}$ have the same size, therefore, faithfulness is equivalent to stability, which means it happens iff $\alpha \geq 1/2$.

During training, we can expand $\| h_1^l \rangle$ in a similar way to eq. (8) as

$$\| h_1^l \rangle = \| W_0^l x_1^{l-1} \widehat{\rangle} + L^{-\gamma}(b^l + c^l)\left(\| \widetilde{\delta} x_0^L \rangle + \sum_{m=l+1}^{L} L^{-\alpha}\| W_0^{m\top} \widetilde{\delta} x_0^m \widehat{\rangle}\right) = \Theta(1 + L^{-\gamma}).$$

Therefore, it is faithful iff $\gamma \geq 0$. It is equivalent to $\alpha \leq 1$ because we have $\alpha + \gamma = 1$.

**Feature diversity exponent.** To simplify the analysis, we assume that $\epsilon L$ is always an integer. We first expand $x_1^{l+\epsilon L} - x_1^l$

$$\| x_1^{l+\epsilon L} \rangle - \| x_1^l \rangle = \sum_{m=l+1}^{l+\epsilon L} L^{-\alpha}\| W_0^m x_1^{m-1} \widehat{\rangle} + \sum_{m=l+1}^{l+\epsilon L} L^{-\alpha-\gamma}(b^m + c^m)\| \widetilde{\delta} x_0^L \rangle$$
$$+ \sum_{m=2}^{L} L^{-\alpha-\gamma} \sum_{l'=\min\{m-1,l\}+1}^{\min\{m-1,l+\epsilon L\}} (b^{l'} + c^{l'})L^{-\alpha}\| W_0^{m\top} \widetilde{\delta} x_0^m \widehat{\rangle}.$$

With $\alpha + \gamma = 1$, it is clear that the first term is $\Theta(L^{-\alpha}\sqrt{\epsilon L}) = \Theta(\epsilon^{1/2}L^{-\alpha+1/2})$, the second term has size $\Theta(\epsilon)$, and the third term has size $\Theta(\sqrt{L} \cdot \epsilon L^{-\alpha}) = \Theta(\epsilon L^{-\alpha+1/2})$. Therefore, there are only two cases here: if $\alpha = 1/2$, the overall size is $\Theta(\epsilon^{1/2} + \epsilon) = \Theta(\epsilon^{1/2})$; if $\alpha > 1/2$, the first and the third term vanish as $L \to \infty$, so the overall size is $\Theta(\epsilon)$. In sum, we have proved Claims 5.4 and 5.5.

**Layerwise linearization.** Claim G.1 is trivial in this simplified setting because layerwise linearization is always true for linear nets. To enlighten the proof of the general case, we recap that $\| \Delta W_1^l \| x_1^{l-1} \rangle = L^{-\gamma}c^l\| \widetilde{\delta} x_0^l \rangle = \Theta(L^{-\gamma})$, which is much smaller than $\| W_0^l x_1^{l-1} \rangle = \Theta(1)$ when $\gamma > 0$. If there were an activation function, the linearization would bring an error of $o(L^{-\gamma})$ in $h_1^l$, which means an error of $o(L^{-\gamma-\alpha}) = o(L^{-1})$ to $x_1^l$.

---

[20]We need $\Delta x$ and $\Delta h$ for stability, but they are similar to $\Delta \mathring{f}_1$.

## H.3 Beyond one step

The argument above is in general tracking the derivatives and covariance, in other words, $\Gamma$ and $C$ in the Depth-$\mu$P case.

Now we generalize Lemma C.3, and obtain the following recursion for $\Gamma$ and $C$

$$\Gamma_{t,r,0,b}\left(\frac{l}{L},q\right) = \Gamma_{t,r,0,b}\left(\frac{l-1}{L},q\right) + L^{1/2-\alpha}\mathbb{I}_{[(t=r)\wedge(b=0)\wedge(l=\lceil Lq\rceil)]}$$

$$+ L^{-\alpha-\gamma}\sum_{s=0}^{t-1}\Gamma_{s,r,1,b}\left(\frac{l}{L},q\right)\left(L^{\gamma-1/2}\Gamma_{t,s,0,1}\left(\frac{l-1}{L},\frac{l}{L}\right) - C_{t,s,0}\left(\frac{l}{L}\right)\right).$$

$$\Gamma_{t,r,1,b}\left(\frac{l-1}{L},q\right) = \Gamma_{t,r,1,b}\left(\frac{l}{L},q\right) + L^{1/2-\alpha}\mathbb{I}_{[(t=r)\wedge(b=1)\wedge(l=\lceil Lq\rceil)]}$$

$$+ L^{-\alpha-\gamma}\sum_{s=0}^{t-1}\Gamma_{s,r,0,b}\left(\frac{l-1}{L},q\right)\left(L^{\gamma-1/2}\Gamma_{t,s,1,0}\left(\frac{l}{L},\frac{l}{L}\right) - C_{t,s,1}\left(\frac{l}{L}\right)\right).$$

$$C_{t,s,a}(p) = \sum_{t'=-1}^{t}\sum_{s'=-1}^{s}\sum_{b\in\{0,1\}}\int_0^1 \Gamma_{t,t',a,b}(l/L,q)C_{t',s',b}(q)\Gamma_{s,s',a,b}(l/L,q)\mathrm{d}q,$$

where $l = \lceil Lp\rceil - 1$ if $a = 0$, and $l = \lceil Lp\rceil$ if $a = 1$.

Then all the claims can be reasoned by tracking the order of $\Gamma$ and $C$.

**Distinguish parametrizations with $\alpha + \gamma = 1$ and $1/2 \leq \alpha \leq 1$.** The parametrizations with $\alpha + \gamma = 1$ and $1/2 \leq \alpha \leq 1$ are all nontrivial, stable, and faithful. However, there is a large gap between $\alpha = 1/2$ (Depth-$\mu$P) and $\alpha > 1/2$ in terms of the difficulty of tracking $\Gamma$ and $C$. For $\alpha > 1/2$, we can see that $C_{t,s,a} = \Theta(1)$, $\Gamma_{t,-1,a,b} = \Theta(1)$ and $\Gamma_{t,s,a,b} = o(1)$ for $s \geq 0$. In this case, we can simplify the recursion by ignoring $\Gamma_{t,s,a,b}$ with $s \geq 0$:

$$\Gamma_{t,-1,0,b}\left(\frac{l}{L}\right) \approx \Gamma_{t,-1,0,b}\left(\frac{l-1}{L}\right) - \frac{1}{L}\sum_{s=0}^{t-1}\Gamma_{s,-1,1,b}\left(\frac{l}{L}\right)C_{t,s,0}\left(\frac{l}{L}\right).$$

$$\Gamma_{t,-1,1,b}\left(\frac{l-1}{L}\right) \approx \Gamma_{t,-1,1,b}\left(\frac{l}{L}\right) - \frac{1}{L}\sum_{s=0}^{t-1}\Gamma_{s,-1,0,b}\left(\frac{l-1}{L}\right)C_{t,s,1}\left(\frac{l}{L}\right).$$

$$C_{t,s,a}(p) \approx \sum_{b\in\{0,1\}}\Gamma_{t,-1,a,b}(l/L)\Gamma_{s,-1,a,b}(l/L),$$

where $l = \lceil Lp\rceil - 1$ if $a = 0$, and $l = \lceil Lp\rceil$ if $a = 1$. Note $\Gamma_{t,-1,a,b}(p,q)$ is simplified to a function that only depends on $p$ because $\Gamma_{t,-1,a,b}(p,q)$ is constant when fixing $p$.

This simplification means the randomness in any $W_0^l$ does not have an effect on the dynamics in the infinite depth limit — the complicated functional integrals for $\alpha = 1/2$ in Proposition C.4 are simplified to be ODEs when $\alpha > 1/2$. This ODE dynamic also directly implies that the feature diversity exponent is 0 for $\alpha > 1/2$.

## I What Causes Hyperparameter Transfer?

In a popular misconception, hyperparameter transfer is implied by the existence of a limit. For example, the fact that $\mu$P transfers hyperparameters, in this misconception, is because of the existence of the feature learning limit (aka the $\mu$ limit), the limit of $\mu$P as width goes to infinity. However, this is not the case. Indeed, there are a plethora of infinite-width limits, such as the NTK limit, but there can only be one way how the optimal hyperparameters scale, so existence cannot imply transfer. In a stronger version of this misconception, transfer is implied by the existence of a

"feature learning" limit. But again, this is False, because there are infinite number of feature learning limits (where the $\mu$ limit is the unique maximal one).

Instead, what is true is that the *optimal* limit implies the transfer of *optimal* hyperparameters. For example, in the width limit case, $\mu$P is the unique parametrization that yields a maximal feature learning limit. Compared to all other limits, this is obviously the optimal one. Hence $\mu$P can transfer hyperparameters across width.

So far, there is no *a priori* definition for the "optimality" of a limit: One can only tell by *classifying* all possible limits; it turns out only a small number of different behavior can occur in the limit, and thus one can manually inspect for which limit is the optimal one.

Similarly, in this work, to *derive* a depthwise scaling that allows transfer, we need to *classify* all possible infinite depth limits — and Depth-$\mu$P will turn out to be optimal in a sense that we define later in the paper.[21] More interestingly than the width case, here we have multiple modes of feature learning when taking the depth limit and it is important to discern which mode of feature learning is optimal. Thus, again, it is *insufficient* to derive any one limit, even with feature learning, and be able to infer it yields HP transfer.

In appendix L, we provide experiments with $1/L$ block scaling $(\alpha, \gamma) = (1, 0)$, aka ODE scaling, which provably induces feature learning in the infinite-depth limit, but is sub-optimal. Our results show a significant shift in the optimal learning rate with this parametrization.

## J  FEATURE DIVERSITY

In this section, we show that the choice of nonlinearity and placement of nonlinearities can affect feature diversity greatly.

### J.1  GRADIENT DIVERSITY

*Gradient diversity* is an important factor toward feature diversity. Observe that the gradient $\delta x^l$ at $x^l$ is continuous in $l$ in the limit $L \to \infty$. In a linear model (or the pre-nonlin model, where nonlinearity is put before the weights), this causes $\delta h^l = L^{-\alpha} \delta x^l$ to be very similar between neighboring blocks. As a result (because the weights $W^l$ receives an update proportional to $\delta h^l \otimes x^{l-1}$), in the next forward pass, neighboring blocks contribute very similarly to the main branch $x^l$. This leads to a waste of model capacity.

### J.2  PRE-NONLIN LEADS TO POOR PERFORMANCE

For example, in Figure 3, for a relu pre-nonlin resnet (i.e. blocks are given by $W^l \phi(x^{l-1})$ instead of $\phi(W^l x^{l-1})$), we see that although Depth-$\mu$P indeed transfers hyperparameters (as predicted by our theory), the performance is dramatically worse than the post-nonlin resnet in Figure 11, and depth gives no performance gains beyond 8 layers. Specifically, it is because $\delta h^l = L^{-\alpha} \delta x^l$ like the linear case, and $\phi(x^{l-1})$ is also similar between neighboring blocks. As a result, the gradient of the weights $W^l$, proportional to $\delta h^l \otimes \phi(x^{l-1})$, has little diversity compared to nearby blocks.

### J.3  MAXIMIZING FEATURE DIVERSITY WITH ABSOLUTE VALUE NONLINEARITY

In a nonlinear model, we have $\delta h^l = \delta x^l \odot \phi'(h^l)$. Because $h^l$ is almost independent from all other $h^m, m \neq l$ in the Depth-$\mu$P limit, $\phi'(h^l)$ can serve to decorrelate the $\delta h^l$, depending on what $\phi$ is. For example, if $\phi$ is relu, then $\phi'$ is the step function. $h^l$ is approximately a zero-mean Gaussian in the Depth $\mu$P limit, so that $\phi'(h^l)$ is approximately 0 or 1 with half probability each. This decorrelates $\delta h^l$ much better than the linear case. But of course, this line of reasoning naturally leads to the conclusion that $\phi' = \text{sign}$ would be the best decorrelator of $\delta h^l$ and the maximizer of feature diversity (with $\phi$ among the class of positively 1-homogeneous functions) — then $\delta h^l$ and $\delta h^m$ are completely decorrelated for $l \neq m$.

---

[21]There are important nuances here that will be spelled out in an upcoming paper. For example, if the space of hyperparameters is not chosen correctly, then it could appear that no limit is *optimal* in any manner. For example, if one in (widthwise) SP, one only thinks about the 1D space of the global learning rate, then all infinite-width limits are defective — and indeed there is no hyperparameter transfer where the bigger always does better.

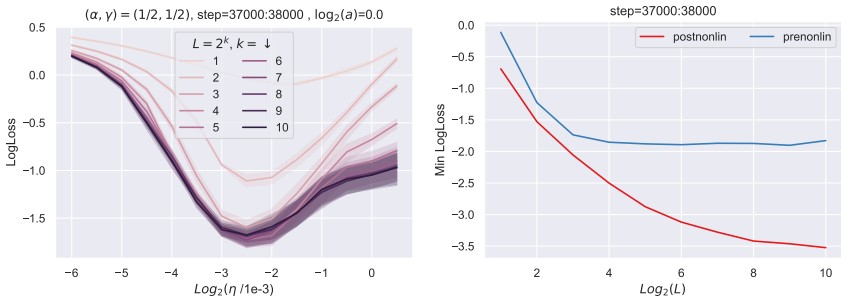

Figure 3: **Pre-Nonlin Leads to Poor Performance** Although Depth-$\mu$P for prenonlin resnet indeed transfers hyperparameters (Left), depth gives no performance gains beyond 8 layers and the performance is dramatically worse than the post-nonlin resnet (Right). In right plot, the "Min LogLoss" is minimal log loss over all block multiplier and learning rate. Networks are trained on CIFAR-10 with Adam. See Figure 11 for more details about the setup.

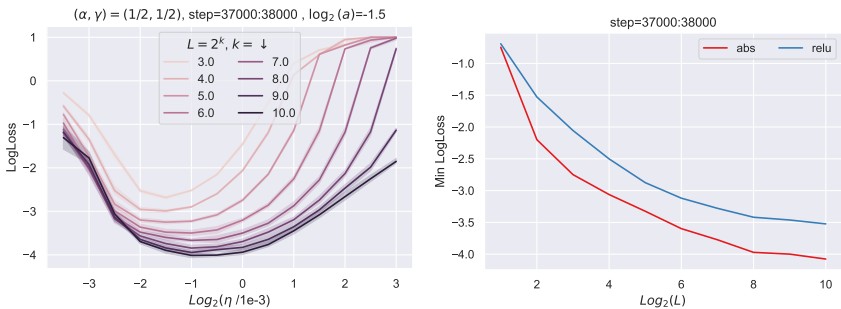

Figure 4: **Improving performance with absolute value non-linearity**, which maximizes feature diversity. (Networks are trained on CIFAR-10 with Adam.). See Figure 11 for more details about the setup.

Indeed, as shown in Figure 4, swapping in absolute value for $\phi$ dramatically improves the training performance of deep (block depth 1) resnets.

In general, in lieu of absolute value, any even nonlinearity would suffice.

### J.4 FEATURE DIVERSITY IS IN TENSION WITH LAYERWISE LINEARIZATION

The reason that $\phi'(h^l)$ can decorrelate $\delta h^l$ is very much related to layerwise linearization. Recall that in Depth-$\mu$P, $h^l$ can be decomposed to a zero-mean Gaussian part $\widehat{h}^l$ of size $\Theta(1)$ and a correction term $\dot{h}^l$ of size $\Theta(L^{-1/2})$ (corresponding to the decomposition $[\![h^l\rangle = [\![\widehat{h^l}\rangle + [\![\dot{h^l}\rangle)$. $\widehat{h}^l$ is independent from $\widehat{h}^m$ for $m \neq l$ but $\dot{h}^l$ can be very strongly correlated to all other $\dot{h}^m$. Thus, $\phi'(h^l)$ can decorrelate $\delta h^l$ precisely because $\widehat{h}^l$ dominates $\dot{h}^l$, and this is also precisely the reason we have layerwise linearization.

In the $1/L$ scaling $(\alpha, \gamma) = (1, 0)$, $\widehat{h}^l$ is on the same order as $\dot{h}^l$ and layerwise linearization does not occur, but also $\phi'(h^l)$ can no longer effectively decorrelated $\delta h^l$.

Once again, we remind the reader that layerwise linearization in this case is not detrimental (in this block depth 1 case) because $\widehat{h}^l$ in fact accumulate contributions from the learned features of all previous blocks and thus strongly depends on the learning trajectory (in contrast to the (widthwise) NTK case where $\widehat{h}^l$ is already determined at initialization).

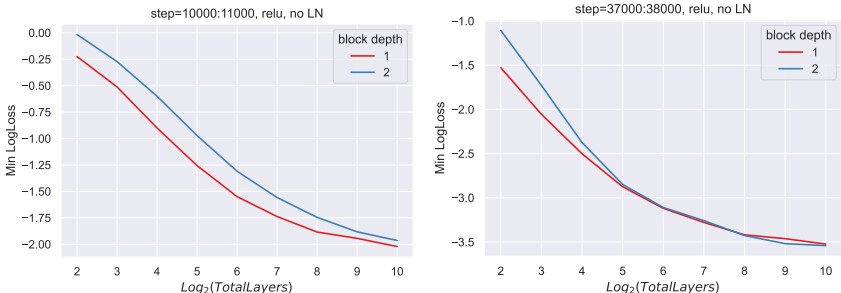

Figure 5: **Block Depth 2 performs worse than Block Depth 1, Relu**. In relu resnet with no LN, block depth 2 does worse than block depth 1 when matching total number of layers (and thus parameter count). However, training longer (38000 steps, Right) helps it catch up (compared to 11000 steps, Left). The y-axis is minimal log loss over all block multiplier and learning rate

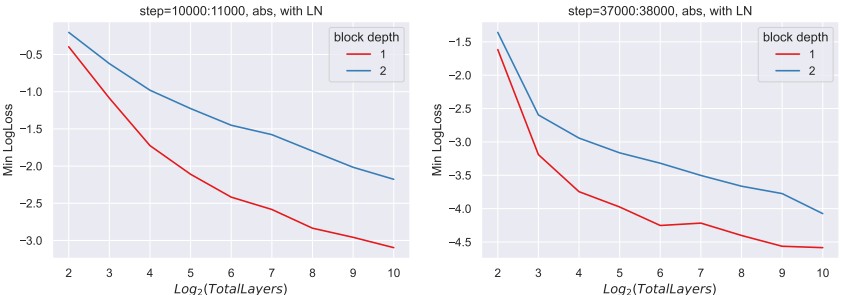

Figure 6: **Block Depth 2 performs worse than Block Depth 1, Abs**. In abs resnet with LN, block depth 2 does significantly worse than block depth 1 when matching total number of layers (and thus parameter count). Training longer (38000 steps, Right) does not close the performance gap (compared to 11000 steps, Left). The y-axis is minimal log loss over all block multiplier and learning rate

## K    BLOCK DEPTH 2 AND ABOVE

*Remark on notation:* Here and in the next section, all big-O notation is in $L$ only; the scaling in width is assumed to be in $\mu$P.

In most of this work, we have considered depth-1 MLP for $g^l$ in eq. (1), it's straightforward to derive and classify the infinite-width-then-infinite-depth limits for larger depths in each block. In particular, the following $1/\sqrt{L}$ scaling still makes sense in this more general setting with block depth $k$ and leads to a well defined limit:

$$x^l = x^{l-1} + \frac{a}{\sqrt{L}} \cdot g^l(x^{l-1}; W^{l1}, \ldots, W^{lk}), \quad \Theta(1) \text{ initialization scale}, \quad \Theta(1/\sqrt{L}) \text{ learning rate} \tag{9}$$

This is what we call Depth-$\mu$P in the block depth 1 case, but we shall not use this name in the general block depth case because *this parametrization is no longer optimal*.[22]

### K.1    BLOCK DEPTH $\geq$ 2 IS DEFECTIVE

A very clear symptom of this is that the *performance of block-depth-2 resnets is worse than that of block-depth-1 networks*, when matching parameter count, although they can (but not always) catch up after training for a long time (figs. 5 and 6). Simultaneously, we are seeing nontrivial or even significant hyperparameter shifts as the total number of blocks increases (fig. 7).

---

[22]What we exactly mean by *optimal* will be explained below.

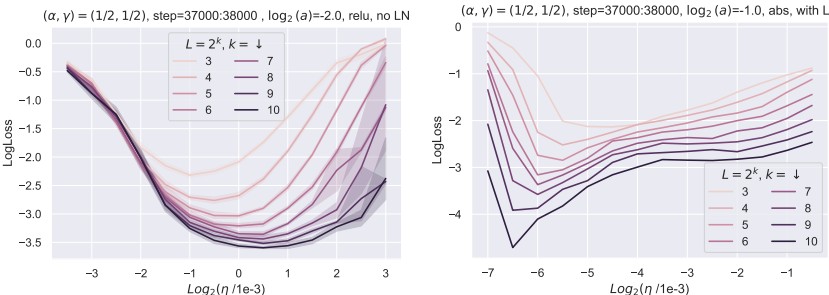

Figure 7: **Block Depth 2 Hyperparameter Shift** in relu resnet with no LN (Left) and abs resnet with LN (Right).

## K.2 DEFECT OF $1/\sqrt{L}$ SCALING IN BLOCK DEPTH 2

The reason that the $1/\sqrt{L}$ scaling is no longer fine in the block depth $\geq 2$ case is the *linearization of the multiplicative interaction* between the layers in the block. Indeed, just like the block depth 1 case, the $1/\sqrt{L}$ scaling forces the weight updates $\Delta W$ of each weight matrix to be $\Theta(\sqrt{L})$ smaller than the initialization $W_0$. Thus, within the block, the training dynamics when depth $L$ is large is in the kernel regime, where the contribution to the block output $g(x; W^\bullet)$ is only a *summation*, instead of *product*, of individual contributions from each layer's weights updates.

When aggregated over all $L$ blocks, the result is that there is only multiplicative interaction of $\Delta W$ across blocks but not within layers. In other words, the network output is dominated, for example in the linear case, by the contributions of the form $M^L \cdots M^1$ where each $M^l$ can be one of $I, W_0^{l2}W_0^{l1}, W_0^{l2}\Delta W^{l1}$, or $\Delta W^{l2}W_0^{l1}$, but NOT $\Delta W^{l2}\Delta W^{l1}$. All other contributions (which all involve within-block interactions like $\Delta W^{l2}\Delta W^{l1}$) are subleading. In the general nonlinear case, replacing the block

$$\phi(W^{l2}\phi(W^{l1}x^{l-1}))$$

with the linearized version

$$\phi(h_\wedge^l) + \phi'(h_\wedge^l) \odot [\Delta W^{l2}\phi(h_\vee^l)] + \phi'(h_\wedge^l) \odot [W_0^{l2}(\phi'(h_\vee^l) \odot [\Delta W^{l1}x^{l-1}])]$$

will achieve the same performance as depth $L \to \infty$, where $h_\wedge^l = W_0^{l2}\phi(h_\vee^l)$ and $h_\vee^l = W_0^{l1}x^{l-1}$.

When block depth $k = 1$ (our main subject of study in this work), *all* interactions are included but this is no longer true when $k > 1$.

In fig. 8, the heatmap of loss as a function of block multiplier and learning rate demonstrates this vividly for block depth 2.

**Small depth**    The optimal sublevel set of (learning rate, block multiplier) has slope $\approx -2$ when the number of blocks is $2^1$. In other words, around the optimum, double the learning rate while dividing the block multiplier by 4 has similar performance. This is because $\Delta W^{l1}$ and $\Delta W^{l2}$ interact *multiplicatively*, so that doubling their sizes leads to quadrupling their contribution to the block output. The simultaneous decrease of block multiplier by 4 then roughly keep their contribution invariant in size.

**Large depth**    On the other hand, the optimal sublevel set has slope $\approx -1$ when the depth is $2^{10}$: Doubling the learning rate while halving the block multiplier has similar performance. This reflects the fact that $\Delta W^{l1}$ and $\Delta W^{l2}$ now interact *additively*.

Intermediate depths interpolate this phenomenon, as seen in the plot for depth $2^5$.

In the same heatmaps, one can see the optimal (learning rate, block multiplier) (in the $1/\sqrt{L}$ parametrization) shifts from the middle of the grid to the upper left as depth goes from $2^5$ to $2^{10}$, demonstrating the lack of hyperparameter transfer.

This change in slope is seen in relu networks as well, with or without layernorm.

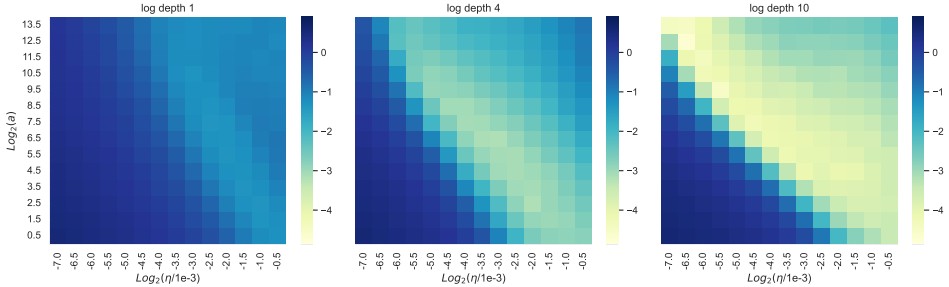

Figure 8: The "slope" of the optimal sublevel set in the (learning rate, block multiplier) space changes from $-2$ to $-1$ as depth goes from $2^1$ to $2^{10}$. Here we use absolute value nonlinearity with layer normalization, block depth 2, and networks are trained for 50 epochs with Adam on CIFAR-10.

Finally, we note that the $1/\sqrt{L}$ scaling still yields a $L \to \infty$ limit where the network still learns features as a whole, even though within each block this is no longer true. Thus, this is another reminder that mere "feature learning" does not imply "hyperparameter transfer"!

### K.3 CLASSIFICATION OF PARAMETRIZATIONS

These heatmaps already demonstrate that no parametrization of (global learning rate[23], block multiplier) can transfer hyperparameters robustly, because any such parametrization can only *shift* the heatmaps but not *stretch* them, so one cannot "transfer" a sublevel set of one slope into a sublevel set of another slope.

But even if we allow learning rate to vary between layers in a block, no stable, faithful, nontrivial parametrization can avoid the linearization problem described above.

For simplicity, fix a positive-homogeneous nonlinearity and block depth 2.[24] We consider the space of hyperparameters consisting of the learning rate for each of the layers in a block, as well as the block multiplier (one for each block); WLOG all weights are initialized $\Theta(1)$.[25] This yields a space of dimension $\text{blockdepth} + 1 = 3$.

Indeed, for this to happen, the weight update $\Delta W^{li}$ must be at least of order $\Omega(1)$ (size of initialization) for some $i$. But this would contribute a drift term to the block output $g^l = g^l(x^{l-1}; W^\bullet)$ that is as large as the noise term. This then implies that either the parametrization is unstable (if the block multiplier $L^{-\alpha}$ is $\Omega(1/L)$) or lacks feature diversity (if the block multiplier $L^{-\alpha}$ is $O(1/L)$).

For example, in a linear model,

$$L^\alpha \llbracket g^l \rangle = \llbracket W^{l2}W^{l1}x^{l-1} \rangle = \llbracket W_0^{l2}W^{l1}x^{l-1}\widehat{\rangle} + \llbracket W_0^{l2}W^{l1}x^{l-1}\rangle + \llbracket \Delta W^{l2}W^{l1}x^{l-1}\dot\rangle.$$

$\llbracket W_0^{l2}W^{l1}x^{l-1}\widehat{\rangle}$ is independent and zero-mean across $l$ (the noise term), while $\llbracket W_0^{l2}W^{l1}x^{l-1}\rangle + \llbracket \Delta W^{l2}W^{l1}x^{l-1}\rangle$ is correlated across $l$ (the drift term). $\llbracket W_0^{l2}W^{l1}x^{l-1}\widehat{\rangle}$ is always $\Theta(1)$ because the $W_0^{l2}, W_0^{l1}$ are. If $\Delta W^{l2}$ is $\Omega(1)$, then $\llbracket \Delta W^{l2}W^{l1}x^{l-1}\rangle = \Omega(1)$ as well, making the drift term as large as the noise term. If $\Delta W^{l1}$ is $\Omega(1)$, then $\llbracket W_0^{l2}\Delta W^{l1}x^{l-1}\dot\rangle = \Omega(1)$, causing $\llbracket W_0^{l2}W^{l1}x^{l-1}\dot\rangle = \llbracket W_0^{l2}W_0^{l1}x^{l-1}\rangle + \llbracket W_0^{l2}\Delta W^{l1}x^{l-1}\dot\rangle$ to be $\Omega(1)$.[26]

The same argument can be straightforwardly adapted to nonlinear MLPs (with mean subtraction) and arbitrary block depth $\geq 2$, and as well to general nonlinearities that are not necessarily positive-homogeneous, with hyperparameter space enlarged to include initialization.

---

[23] meaning, the learning tied across all layers in a block

[24] but our arguments generalize trivially to arbitrary block depth $\geq 2$

[25] This is WLOG because the nonlinearities are homogeneous

[26] One can also observe that if $\Delta W^{l1} = \Omega(1)$, then by symmetry the backward pass suffers the same problem. But for general block depth, this argument does not say anything about the middle layers, while the argument presented above implies that $\Delta W^{li}$ cannot be $\Omega(1)$ for any $i$.

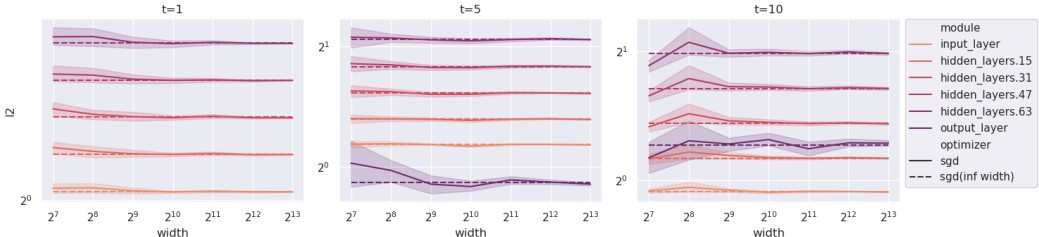

Figure 9: Trained linear network converges to its infinite width limit which is computed recursively based on $\Gamma$ and $C$. Depth is fixed at 64, width varies between $2^7, 2^8, \ldots, 2^{13}$. Networks are trained with SGD for 10 steps. The root mean square statistics ($y$-axis) at 1st, 5th and 10th steps are plotted using solid lines where the $x$-axis is the width. The root mean square values are computed on the outputs of some of the layers (including the input layer, output layer, and hidden layers at each quarter). The corresponding value for the infinite width is indicated with dashed lines.

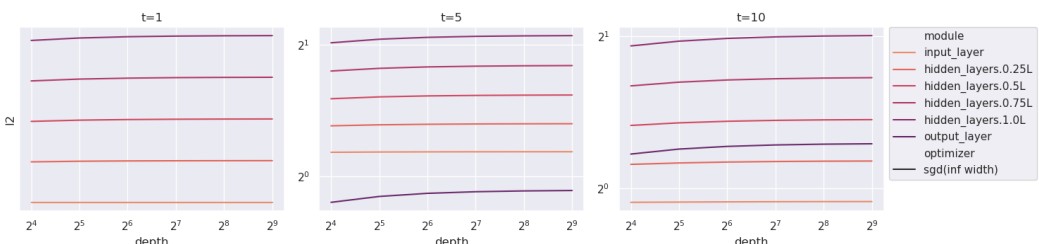

Figure 10: Under Depth-$\mu$P, infinite wide linear network training converges when increasing the depth. Infinite wide linear networks of depth $2^4, 2^5, \ldots, 2^9$ are computed recursively based on $\Gamma$ and $C$. The root mean square statistics ($y$-axis) at 1st, 5th and 10th steps are plotted across the depth ($x$-axis).

### K.4  SO WHAT IS THE OPTIMAL PARAMETRIZATION?

All of the above considerations suggest that *we are missing crucial hyperparameters in our consideration* when increasing the complexity of each block. Our study right now is akin to the naive study of the 1-dimensional hyperparameter space of the global learning rate in SP. Discovering these missing hyperparameters will be an important question for future work.

## L  FULL EXPERIMENTS

### L.1  VERIFYING THE THEORY IN THE LINEAR CASE

In Section 3.2, we showed that a complete description of the training dynamics of linear networks can be formulated in terms of $\Gamma$ and $C$. In this section, we provide empirical results supporting our theoretical findings. We first verify the finite-depth recursive formula for $\Gamma$ in Lemma C.3 is the correct limit when the width goes to infinity, then proceed to show that the infinite-depth limit is the correct one.

**Infinite-width limit.** In Figure 9, we train a series of 64-layer linear networks of width $2^7, 2^8, \ldots, 2^{13}$ with $1, 5, 10$ steps on MNIST, and plot the root mean square[27] of the layer outputs using solid lines. We also compute the infinite width limit of the corresponding statistics using the recursive formula for $\Gamma$ and plot them as dashed horizontal lines. For clarity of the figure, we only

---

[27]The root mean square of a vector $x = (x_1, \ldots, x_n)$ is $\sqrt{\frac{\sum_{i=1}^{n} x_i^2}{n}}$, which is denoted as "l2" in Figures 9 and 10.

plot the statistics of the input layer, output layer, and hidden layers of index 16, 32, 48, and 64. It is clear that as the width grows, the solid lines converge to the dashed lines consistently across the training steps. It indicates that our computation of the infinite width limit is correct.

**Infinite-depth limit.** We verify that the infinite *width* limit above converges when the *depth* grows. We consider linear networks of the same architecture but vary the depth from $2^4$ to $2^9$. We again compute the root mean square values of the layer outputs using the recursive formula for $\Gamma$, and plot them in Figure 10 with depth being $x$-axis. For clarity of the figure, we only plot the statistics of the input layer, output layer, and hidden layers of index $L/4$, $L/2$, $3L/4$, and $L$. One can observe that the statistics of the layer outputs converge quickly when the depth grows from $2^4$ to $2^9$, which verifies our convergence result.

## L.2 HYPERPARAMETER TRANSFER

In this section, we provide empirical evidence to show the optimality of Depth-$\mu$P scaling and the transferability of some quantities across depth. We train vanilla residual network with block depth 1 (1 MLP layer in each residual block) on CIFAR-10 dataset using Adam optimizer, batch size $64$, for 50 epochs (input and output layers are fixed). The network is parameterized as follows

$$x^l = x^{l-1} + a \times L^{-\alpha}\text{MS}(\phi(W^l x^{l-1})),$$

and the weights are trained with the rule

$$W^l \leftarrow W^l - \eta \times n^{-1}L^{-\gamma}Q_t^l(nL^\delta g_0, \ldots, nL^\delta g_t),$$

where the learning rate $\eta$ and the block multiplier $a$ are the *hyperparameters*.[28] The values of $\alpha, \gamma$ depend on the parametrization of choice. For Depth-$\mu$P, we have $\alpha = \gamma = 1/2$, and for standard parametrization, we have $\alpha = 0, \gamma = 1$.[29] In our experiments, we assume base depth 8, meaning that we replace $L$ by $L/8$ in the parametrization above.

**Learning rate transfer ($\eta$).** In Figure 11, we show the training loss versus learning rate for depths $2^k$, for $k \in \{3, 4 \ldots, 10\}$. For Depth-$\mu$P, a convergence pattern can be observed for the optimal learning rate as depth grows. Optimal learning rates for small depths (e.g. $L = 2^3$) exhibit a mild shift which should be expected, as our theory shows convergence in the large depth limit. However, starting from depth $L = 2^6$, the optimal learning rate is concentrated around $10^{-3}$. For parametrization that only scales the multiplier but not LR ($\alpha = 1/2$, $\gamma = 0$), we observe the optimal learning rate shifts significantly. For standard parametrization without any depth scaling ($\alpha = \gamma = 0$), the optimal learning rate exhibits a more significant shift as depth grows. Moreover, even if one picks the optimal learning rate for each depth, the performance still degrades when the depth is very large, suggesting that standard parametrization is not suitable for depth scaling. Additional figures with multiple time slices are provided in Appendix M.

**Is feature learning sufficient for HP transfer?** In Appendix I, we explained when and why hyperparameter transfer occurs. Precisely, to obtain HP transfer, one needs to classify all feature learning limits and choose the optimal one. We introduced the notion of feature diversity and showed that Depth-$\mu$P is optimal in the sense that it maximizes feature diversity. To show that optimality is needed for HP transfer, we train a resnet with $(\alpha, \gamma) = (1, 0)$ which is also a feature learning limit. Figure 12 shows that in this case the learning rate exhibits a significant shift with depth. Interestingly, the constant $\eta$ in this case seems to increase with depth, suggesting that the network is trying to break from the *ODE* limit, which is sub-optimal. Note that in Figure 11, with Depth-$\mu$P we obtain better training loss compared to the ODE parametrization in Figure 12.

**Do we still have transfer with LayerNorm (LN)?** Our theory considers only Mean Substraction (MS), and Figure 11 shows the results with MS. To see wether LN affects HP transfer, we train resnets with the same setup as Figure 11 with absolute value non-linearity and LN applied to $x^{l-1}$

---

[28]Note that $\eta$ here is the constant, and the effective learning rate is given by $\eta n^{-1}L^{-\gamma}$.

[29]In standard parametrization, there is generally no rule to scale the learning rate with depth, and the optimal learning rate is typically found by grid search. Here, we assume that in standard parametrization, the learning rate is scaled by $L^{-1}$ to preserve faithfulness.

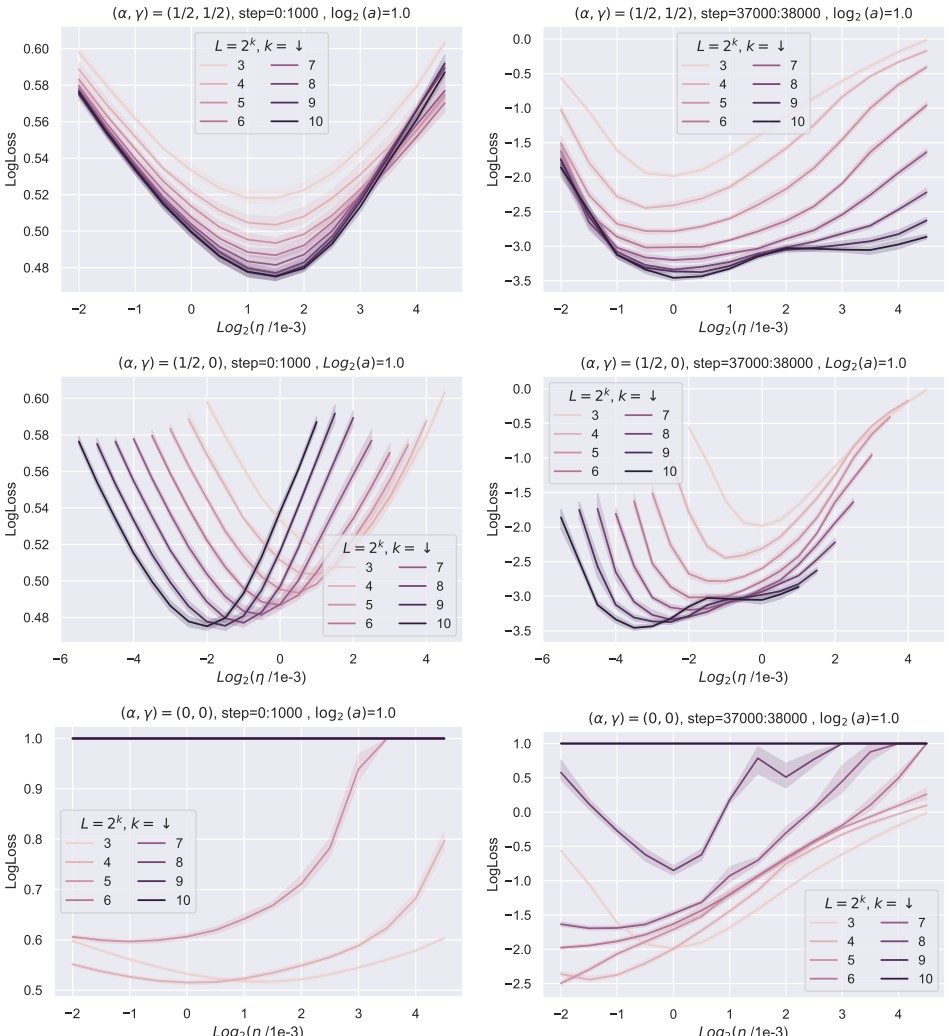

Figure 11: Train logloss versus learning rate for width $n = 256$ and varying depths. The network consists of MLP blocks (with block depth 1), trained for 50 epochs on CIFAR10 dataset using Adam. The batch size is fixed to 64. We tune the depth $2^3$ network to obtain the optimal $(\log_2(a), \log_2(\eta/1e - 3)) = (1, 0)$, and scale all deeper networks using $2^3$ as base depth. The reader can check that the $L = 2^3$ curves in each columns are the same. We show the logloss versus the learning rate of the hidden layers (input/output layers fixed) for three parametrizations: Depth-$\mu$P (**Top**), Scaling only the blocks (no LR scaling), i.e. $\gamma = 0$ (**Middle**), and Standard Parametrization without any scaling ($\alpha = \gamma = 0$) (**Bottom**). Each curve represents the average training loss over a time slice of 1000 steps for depths $2^k$ for $k \in \{1, 2, \ldots, 10\}$. Confidence intervals are based on 5 seeds. The results show that Depth-$\mu$P preserves the optimal learning rate while consistently improving the training loss as depth increases. If we only scale the blocks without scaling the LR ($\alpha = 1/2, \gamma = 0$) when training with Adam, the optimal learning rate shifts significantly with depth. With standard parametrization without any depth scaling (common practice), the results show a significant shift in the optimal learning rate as well. For SP, we cap the log loss at 1, which is why for depth $2^9, 2^{10}$, we have a black horizontal line at $LogLoss = 1$.

before matrix multiplication with $W^l$ (preLN). We keep MS after non-linearity although it can be removed since LN is applied in the next layer. Our results, reported in Figure 13 suggest that Depth-$\mu$P guarantees learning rate transfer with LN as well.

**Block multiplier transfer** ($a$). In Figure 14, we investigate the stability of the hyperparameter $a$ in Depth-$\mu$P as depth increases. The results suggest that the optimal value of this constant converges

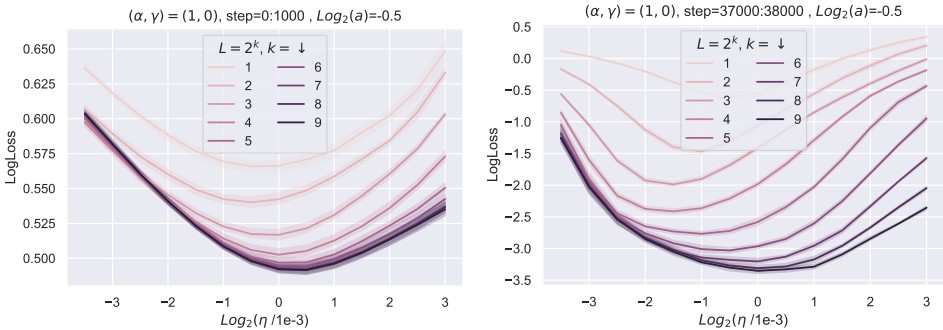

Figure 12: Same setup as fig. 11 for the parametrization $(\alpha, \gamma) = (1, 0)$ (the ODE limit).

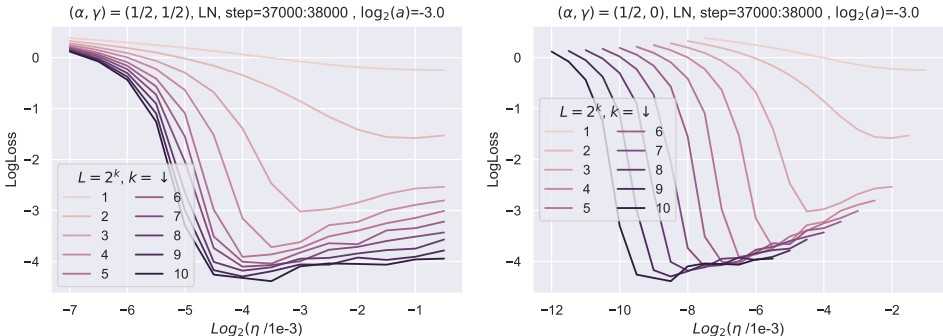

Figure 13: Same setup as Figure 11 with Abs non-linearity instead of ReLU and LayerNorm applied to $x^{l-1}$ before matrix multiplication with $W^l$. We show the logloss versus the learning rate of the hidden layers (input/output layers fixed) for two parametrizations: Depth-$\mu$P (**Left**) and scaling only the blocks without LR scaling ($(\alpha, \gamma) = (1/2, 0)$) (**Right**). The results show that Depth-$\mu$P preserves the optimal learning rate while consistently improving the training loss as depth increases. If we only scale the blocks without scaling the LR ($\alpha = 1/2, \gamma = 0$) when training with Adam, the optimal learning rate shifts significantly with depth.

as depth grows, which suggest transferability. Additional experiments with multiple time slices are provided in Appendix M.

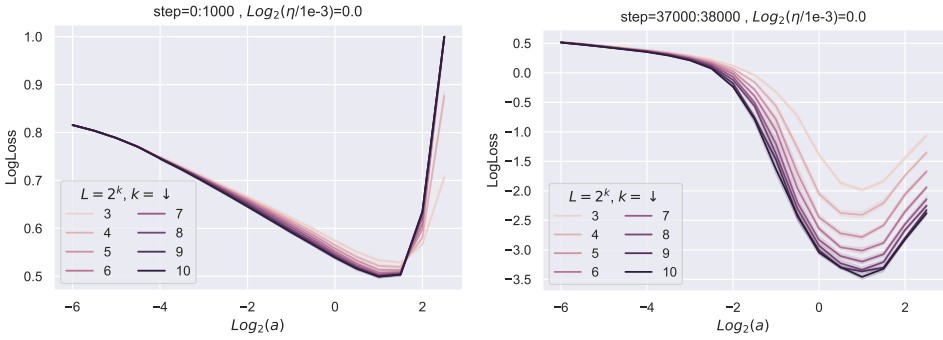

Figure 14: Train logloss versus block multiplier $a$ for varying depths. Same training setup as in fig. 11. The results suggest that Depth-$\mu$P stabilizes the hyperparameter $a$ as depth increases.

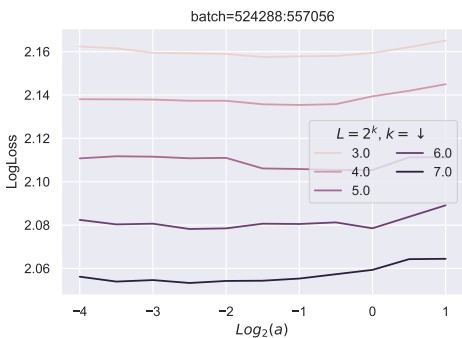

Figure 15: Modern transformers are insensitive to block multiplier $a$.

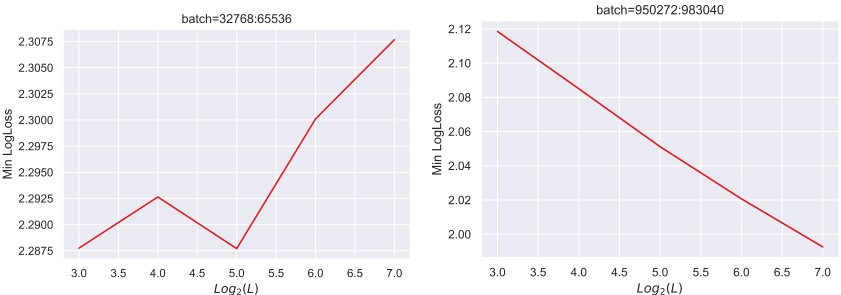

Figure 16: In (Megatron) Transformer trained on Common Crawl, deeper does worse initially (Left) but eventually does better (Right).

### L.3 What Happens in a Transformer?

Because transformers have block depth 2, as discussed in appendix J.4, we have plenty of reasons to suspect that no parametrization of (learning rate, block multiplier) will be able to robustly transfer hyperparameters across depth for transformers.

Here we do a large scale experiment using Megatron trained on Common Crawl and catalogue our observations.[30] In summary, in our particular setup (which should be close to most large language model pretraining), we see that the $1/\sqrt{L}$ scaling seems to transfer hyperparameters at the end of training (Figure 17(Right)). However, we also see that 1) deeper does worse in initial training (Figure 16(Left)), and 2) optimal hyperparameters scale like $\Theta(1)$ in the middle of training (Figure 17(Left)). Combined with the theoretical insights of Appendix J.4, this leads us to conclude that while the $1/\sqrt{L}$ scaling can potentially be practically useful in transformer training, it is likely to be brittle to architectural and algorithmic changes, or even simple things like training time.

In fact, we observe that transformers are insensitive to the block multiplier $a$ (Figure 15), so that the only relevant hyperparameter is really just learning rate. Thus, empirically measuring the scaling trend of the optimal learning rate, as done in modern large scale pretraining, can be a practically more robust way to transfer hyperparameters.

Here $L$ is the number of transformer layers, each of which consists of an attention layer and an MLP layer (each of which has depth 2).

### L.4 Feature Diversity

In this section, we empirically verify our claims about feature diversity exponent (Claims 5.4 and 5.5). We use the same setup as in Appendix L.2, i.e., we train deep residual networks of width $n = 256$ on CIFAR-10 dataset with Adam and batch size $64$. In Figure 18, we compare

---

[30]We train the models for 3900 steps, using cosine decay schedule with 500 warmup steps. We use a sequence length of 4096, batch size 256, resulting in approximately 4B tokens per training run.

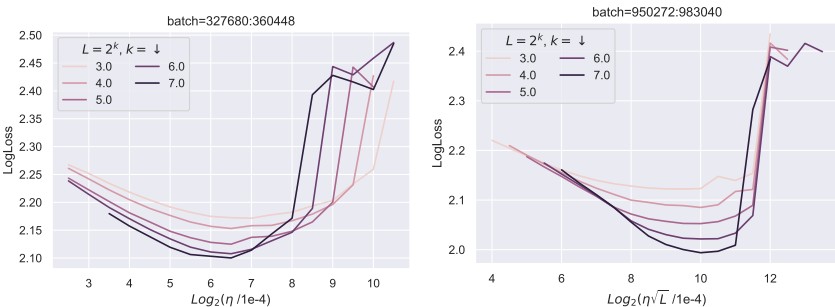

Figure 17: In the middle of (Megatron) transformer training, optimal learning rate is approximately invariant (Left), while at the end of training, it approximately scales like $1/\sqrt{L}$. However, the $1/\sqrt{L}$ scaling transfers the maximum viable learning rate better in either case.

two parametrizations, Depth-$\mu$P ($\alpha = \gamma = 1/2$) and the ODE parametrization $(\alpha, \gamma) = (1, 0)$. We measure $\left\| \boldsymbol{x}_t^{\lfloor (\lambda + \epsilon) L \rfloor} - \boldsymbol{x}_t^{\lfloor \lambda L \rfloor} \right\| \stackrel{\text{def}}{=} d(\epsilon)$ at $t = 1000$ for the two parametrizations and varying depth. For each parametrization and depth $L$, we rescale function $d$ by multiplying a constant $c$ such that $c \cdot d(1/256) = 1$, and then plot the rescaled function $c \cdot d$ for a clean presentation. One can observe clearly that Depth-$\mu$P has feature diversity exponent (almost) $1/2$ for any $L$, while the curves for ODE parametrization move from $\epsilon^{1/2}$ to $\epsilon$ when $L$ grows. This exactly fits our theory that Depth-$\mu$P maximizes the feature diversity, while other parametrizations (even with feature learning) have smaller feature diversity exponents that should go to $0$ in the infinite depth limit.

**Growth along with $L$ and $t$.** In Figure 19, we measure $d(\epsilon)$ at $t = 100, 500, 1000$, and rescale it by *dividing additional* $\epsilon^{0.5}$ and a constant $c$ such that $\frac{d(1/256)}{c \cdot \epsilon^{0.5}} = 1$, and then plot the rescaled function $d/(c \cdot \epsilon^{0.5})$ for a clean comparison between $d$ and $\epsilon^{0.5}$. We observe that for both Depth-$\mu$P and ODE parametrization, the slopes of the curves grow along with $L$ and $t$. The growth along $t$ can be explained by the cumulative correlation between layers. The growth along $L$ for ODE parametrization is because the independent components between nearby layers decrease when $L$ grows. We do not have a clear understanding for the growth along $L$ for Depth-$\mu$P and we leave it as a future work.

**Absolute value activation increases feature diversity.** In Figure 20, we plot the same curves as in Figure 19 but comparing ReLU activation and absolute value activation under Depth-$\mu$P. We observe that the slope of the curves for absolute value activation is smaller than ReLU activation. It matches our theory in Appendix J that absolute value activation increases feature diversity.

## M    ADDITIONAL FIGURES FOR HYPERPARAMETER TRANSFER EXPERIMENTS

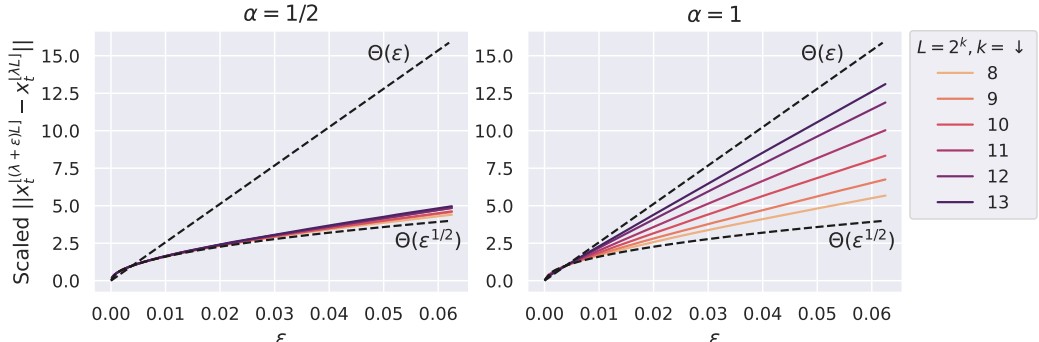

Figure 18: Difference between feature at layer $\lfloor \lambda L \rfloor$ and feature at layer $\lfloor (\lambda + \epsilon) L \rfloor$ as a curve of $\epsilon$ for width $n = 256$ and varying depths. For a clean presentation, each curve is scaled by a constant so it always passes $(1/256, 1)$. The feature diversity exponent $\kappa$ depends on the growth of the curve when $L \to \infty$. For Depth-$\mu$P (left), the curve is always close to $\epsilon^{1/2}$, meaning $\kappa = 1/2$. For ODE parametrization (right), the curve shifts from $\epsilon^{1/2}$ to $\epsilon$ when $L$ grows, indicating its $\kappa$ goes to $0$ in the infinite depth limit.

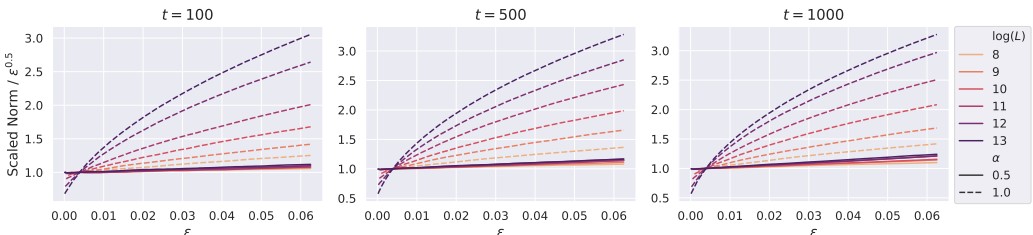

Figure 19: Same setup as Figure 18 but at step $t = 100, 500, 1000$, and each curve is scaled by dividing a constant and *additional* $\epsilon^{1/2}$ so it always passes $(1/256, 1)$. The curve indicating feature diversity exponent $\kappa$ exactly $1/2$ should be a horizontal line at $1$. For Depth-$\mu$P ($\alpha = 0.5$), the curves are almost horizontal. For ODE parametrization ($\alpha = 1$), slopes of the curves are larger with larger $L$ and larger $t$.

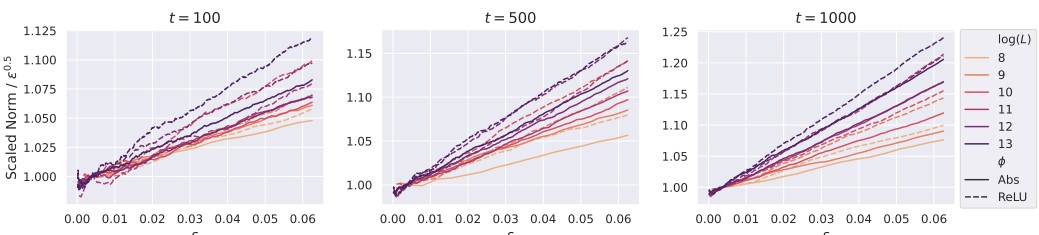

Figure 20: Same setup as Figure 19, but comparing Depth-$\mu$P with ReLU activation and absolute value activation. Each curve is scaled by dividing a constant and $\epsilon^{1/2}$ so it always passes $(1/256, 1)$. The curve indicating feature diversity exponent $\kappa$ exactly $1/2$ should be a horizontal line at $1$. For both activations, slopes of curves are small, but growing along with $L$ and $t$. The slopes with absolute value activation ($\phi = \text{Abs}$) are slower than the slopes with ReLU activation ($\phi = \text{ReLU}$), indicating feature diversity is higher with absolute value activation.

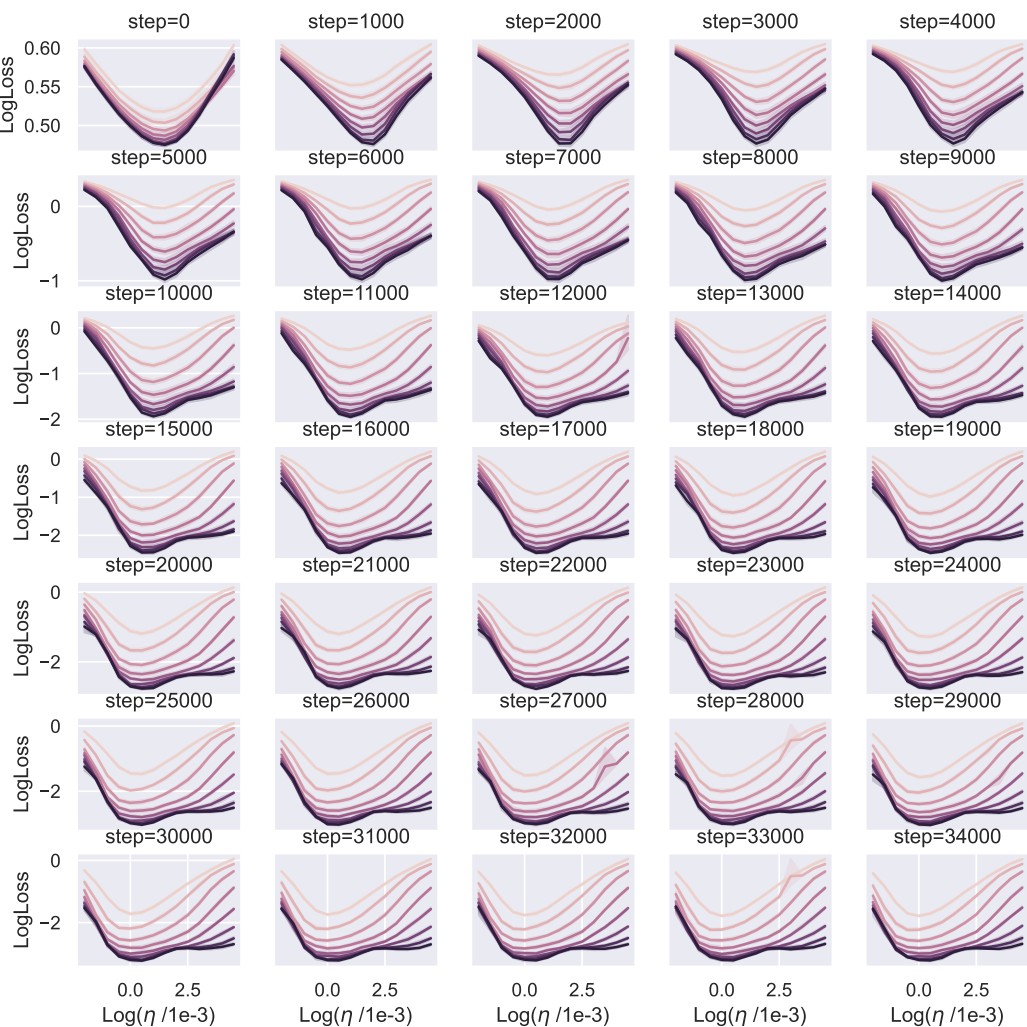

Figure 21: Same as fig. 11 (**Up**, Depth-$\mu$P) with multiple time slices.

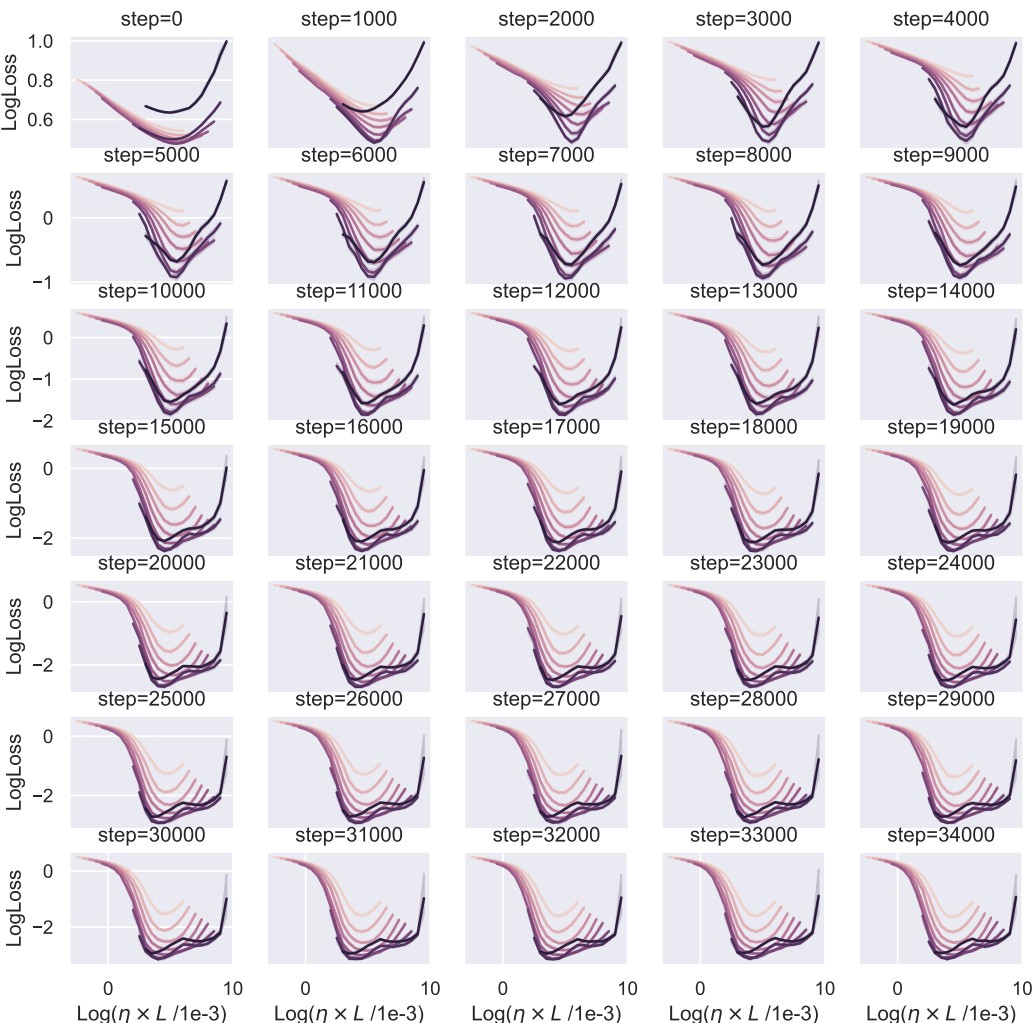

Figure 22: Same as fig. 11 (**Middle**, Standard Parametrization with $\gamma = 1$) with multiple time slices.

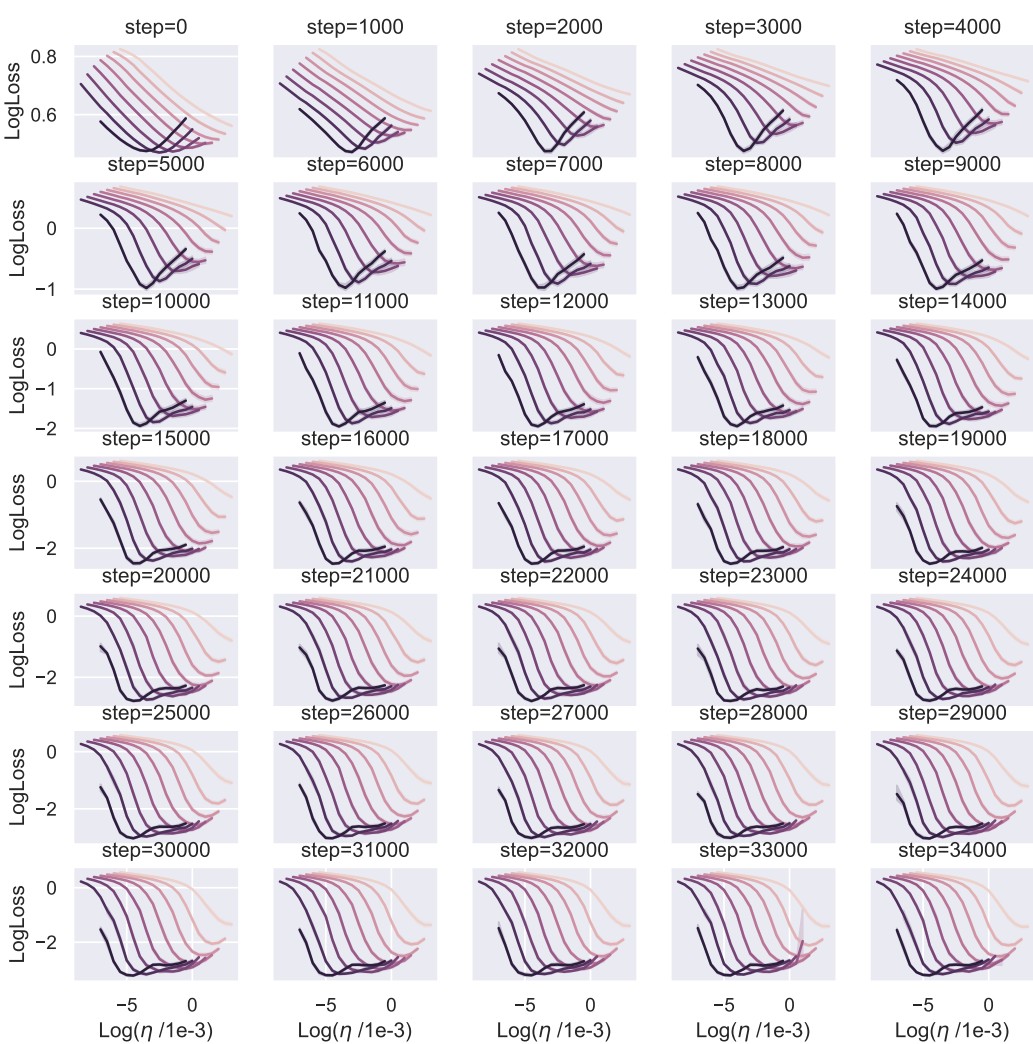

Figure 23: Same as fig. 11 (**Bottom**, Standard Prameterization with no scaling, $\alpha = 0, \gamma = 0$) with multiple time slices.

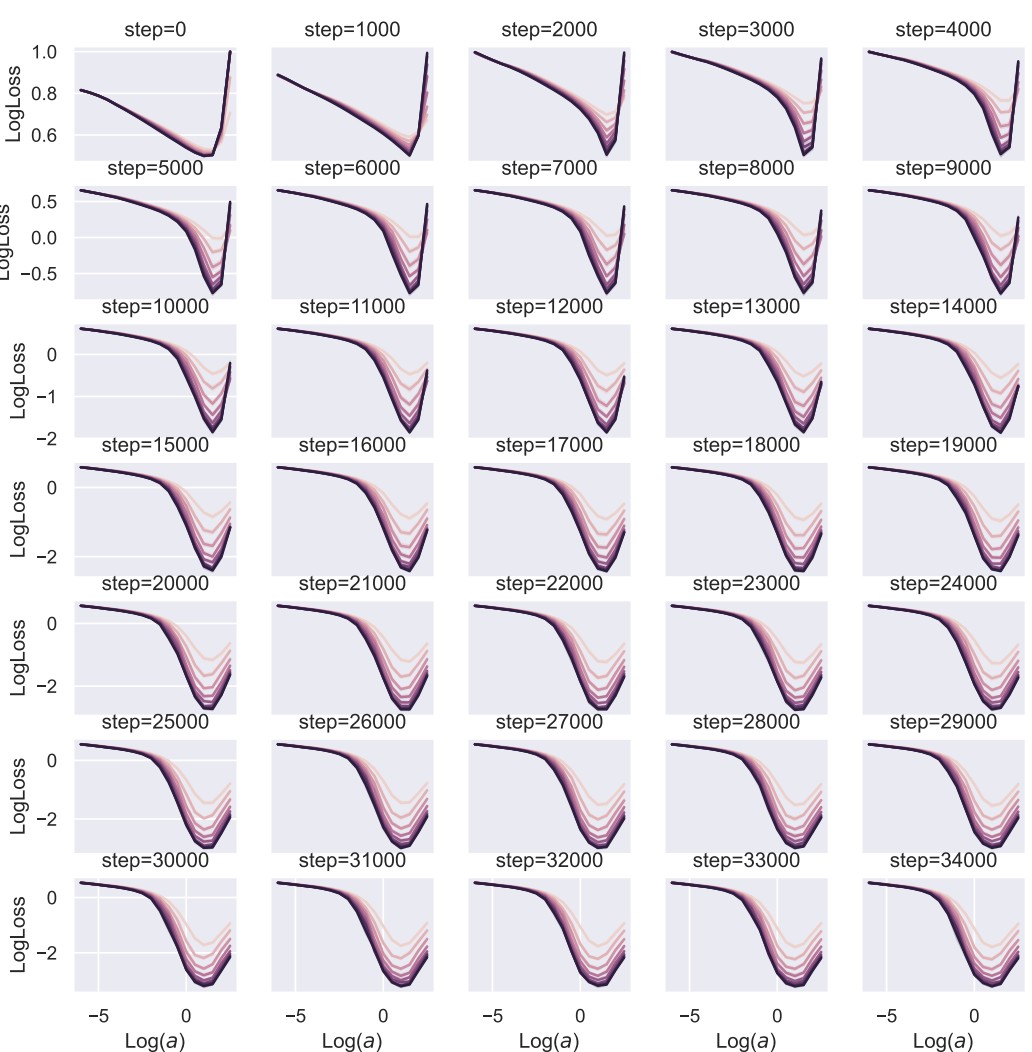

Figure 24: Same as fig. 14 with multiple time slices.

