# OpenReview forum: "Tensor Programs VI: Feature Learning in Infinite Depth Neural Networks"
_ICLR.cc/2024/Conference — ICLR 2024 poster_

### Official Review · Reviewer_5jxB · 2023-10-30

**Soundness:** 2 fair
**Presentation:** 3 good
**Contribution:** 4 excellent
**Rating:** 8
**Confidence:** 4

**Summary:**

The paper studies the scaling limits of residual models in the infinite depth limit $L \to \infty$ and their training dynamics. In particular, they study a parametrization where the residual branches are scaled by $1/\sqrt{L}$ (previously studied at initialization by several prior works). In order to achieve a feature learning limit as both the width and depth tend (sequentially) to infinity, the authors adopt the $\mu P$ parametrization that guarantees maximal feature learning in the infinite width limit ($N \to \infty$). For the proposed model, the authors devise equations describing the training dynamics using the Tensor Programs framework. The authors also introduce the concept of feature diversity, defined as the norm of the difference of the features of nearby layers. They show that among the class of admissible parameterizations in the limit $N,L \to \infty$, the proposed parametrization maximizes feature diversity. The authors apply their theory to hyperparameter transfer, showing that the optimal hyperparameters transfer as the model size increases (in terms of depth).

**Strengths:**

The main strengths of the paper lie in the fact of contributing to multiple research areas, and in striking a combination of important theoretical results and practical consequences. In particular:
1. The authors devise a parametrization that guarantees feature learning in the infinite width-and-depth limit, nicely extending existing works on feature learning in the mean-field (infinite width) limit.
2. The authors have extensive theory describing the training dynamics in the limit (Section 4, Appendix).
3. The concept of feature diversity is novel, and allows the authors to nicely classify the infinite depth parameterizations according to this measure.
4. The problem of hyperparameter transfer is of utmost importance for practitioners. To my knowledge, this problem was previously addressed when scaling the width to infinity, but not the depth. Hence, this work allows the practitioners to tune the architecture at relatively small widths and depths, and then scale up the architecture with a rough guarantee that the hyperparameters also transfer.
5. Experiments, although they do not seem to test more widely used architectures such as convolutions and attention models, test very deep networks, which makes me confident about the validity of most of the proposed claims.

**Weaknesses:**

The main weaknesses lie around a few conceptual leap of faiths that in my view are not entirely justified either by theory or experimentally.
1. **How does maximizing feature diversity result in the "optimal" parametrization?**: The authors identify a class of parameterizations that satisfies a number of desiderata (stable, nontrivial, faithful, feature learning). Among these, the authors argue at multiple points in the paper that Depth-$\mu$P is the one maximizing feature diversity, and hence "optimal" (E.g. at the end of the introduction, the beginning of Experiment Section). In which sense is maximizing feature diversity optimal? Suppose it is in terms of convergence to low training loss. In that case, the authors should have included more experiments testing more models in the class of parameterizations that satisfy stability, nontriviality, faithfulness, and feature learning, but are not maximally diversifying features. However, all the baselines are for $\alpha=0$, which are not in the regime mentioned above ($\alpha \geq 1/2$).
2. **Why does hyperparameter transfer happen?**. While it is understandable that hyperparameters should not transfer under the SP baselines (initialization is unstable under $\alpha=0$), the authors make little effort to justify (with a heuristic argument or experiments) why hyperparameters should transfer under the proposed feature learning regime. I guess in principle, the constant learning rate $\eta$ could still shift constantly with increasing model size. Hence even an experiment testing (e.g. showing that other admissible parameterizations do not exhibit transfer) would have made the claims in the paper more supported by evidence.
3. **Missing citations**. There are a number of works studying feature learning in infinite-width networks. I think some of these works should have been cited. Also, most of the literature on Gaussian process behavior in neural networks is not cited (e.g. see below).

Minor:
4. "In Appendix G.4, we provide “heuristic” proofs that can be made rigorous under non-trivial technical conditions". It is slightly confusing. I thought these claims were the result of the theory developed in Section 5. While I believe it is perfectly fine for a subset of the proofs to be non-rigorous, I am confused as to what are the additional assumptions beyond the ones in the rigorous theory of Section 5.

Song Mei, Theodor Misiakiewicz, and Andrea Montanari. "Mean-field theory of two-layers neural networks: dimension-free bounds and kernel limit."

B Bordelon, C Pehlevan, "Self-consistent dynamical field theory of kernel evolution in wide neural networks"

Lenaic Chizat and Francis Bach. "Implicit bias of gradient descent for wide two-layer neural networks
trained with the logistic loss"

Lee at al. "Deep Neural Networks as Gaussian Processes"

Lee et al. "Wide neural networks of any depth evolve as linear models under gradient descent"

**Questions:**

1. I would like the authors to comment on why they think hyperparameters should transfer in the proposed feature learning limit (as opposed to the NTK limit for instance).
2. Training time. How much do the optimal hyperparameters (e.g. learning rate) shift as a function of training time t? It would be nice to comment on this in the main paper, as practitioners might be interested in tuning the architecture with minimal amount of training steps.
3. Why use the mean-centered version of Resnet? For instance, in https://arxiv.org/abs/2302.00453 the weights are applied after the nonlinearity, thus achieving centering.
3. Validity of infinite depth limit under the tensor program. It is my understanding of tensor programs that a finite number of tensor variables (i.e. finite number of layers) are required to apply the tensor program framework. How is the infinite depth limit handled? At initialization, the commutativity of the limits was proven in https://arxiv.org/abs/2302.00453. Is the order of the limits still commutative during training?

Overall, I am in favor of acceptance of this paper, although I have some reserves that I hope will be addressed during the rebuttal period.

---

> ### Author Response · Authors · 2023-11-19
>
> We thank the reviewer for the thoughtful and insightful review! We would like to first address the major concern about the cause of hyperparameter transfer.
>
> **What causes hyperparameter transfer in general?** We have added a new section (Appendix I), particularly for discussing what leads to hyperparameter transfer. The fundamental idea is that the *optimal* limit implies the transfer of *optimal* hyperparameters. In the scenario of widthwise scaling (discussed in Tensor Program IV and Tensor Program V paper), $\mu$P has the optimal limit. In other words, all other widthwise scalings have obvious defects when compared with $\mu$P, so they must perform worse than $\mu$P in the limit. Thus, optimal hyperparameters transfer under $\mu$P.
>
> **What causes hyperparameter transfer across depth?** Back to our scenario where we consider the depthwise scaling, the question is whether there exists a parameterization (represented by $\alpha$ and $\gamma$) that is always better than others. Without the concept of feature diversity, we are unable to identify one because a parametrization is stable, nontrivial, faithful, and induces feature learning if $\alpha+\gamma=1$ and $\alpha\in [1/2, 1]$. The good news is that there is one unique parametrization (i.e., Depth-$\mu$P, or $\alpha=\gamma=1/2$) that maximizes feature diversity. The feature diversity exponent for Depth-$\mu$P is $1/2$, while the feature diversity exponent for $\alpha\in (1/2,1]$ is always $0$.
>
> **Why is feature diversity important?** Conceptually, low feature diversity prevents the intermediate features from changing quickly across layers, and hence the network is wasting the capacity. We have added another section (Appendix J) to explore more details and applications of feature diversity. We have also added experiments in Appendix L.2 showing that maximal feature diversity is necessary for hyperparameter transfer, as the so-called ODE parametrization ($\alpha=1,\gamma=0$) induces feature learning but fails to transfer hyperparameters. In Appendix L.4, we also empirically verify our claims about feature diversity exponents for different parametrizations and different nonlinearities.
>
> ---
>
> Below we address other comments one by one.
>
> **Missing citations.**
>
> We will provide a more comprehensive reference section in the final version.
>
> **How much do the optimal hyperparameters (e.g. learning rate) shift as a function of training time t? It would be nice to comment on this in the main paper, as practitioners might be interested in tuning the architecture with minimal amount of training steps.**
>
> In general, the shift across time depends on the task, model, and training recipe. For example, the optimal learning rate becomes smaller when trained longer with a constant learning rate schedule. However, one would expect using a proper learning rate decaying strategy can reduce the shift. Note that with the same reasoning, we expect the whole learning rate schedule to transfer with depth.
>
> **Why use the mean-centered version of Resnet? For instance, in https://arxiv.org/abs/2302.00453 the weights are applied after the nonlinearity, thus achieving centering.**
>
> In our revised version, we provide the reason during our discussion about feature diversity in Appendix J. The short answer is that with nonlinearity posed before the weights, the gradients of nearby layers have little diversity, hence effectively wasting capacity.
>
> **… How is the infinite depth limit handled? … Is the order of the limits still commutative during training?**
>
> This work focuses on the infinite-width-then-depth limit. In other words, we consider the case where width >> depth, which is exactly why we can apply the Tensor Program framework while the depth goes to infinity.
>
> However, we still expect that the infinite width and depth limits *commute* during training, especially because the width is sometimes smaller than the depth in our experiments. Nevertheless, it is difficult to prove without more advanced theoretical tools, and we are currently investigating this for future work.

---

> > ### Comment · Reviewer_5jxB · 2023-11-22
> >
> > I appreciate and thank the authors for their response. I also appreciate the additional experiments that the authors run in Appendix J, K, and L.
> >
> > ## On why hyperparameter transfer and maximizing feature diversity.
> > Indeed maximizing feature diversity seems to correlate with better transfer (e.g. Figure 11, 12). I still slightly disagree with the phrasing regarding the optimality of a limit and how they imply hyperparameter transfer. I.e. how does maximizing feature diversity imply that the optimal learning rate converges to an optimal in $L, N$ and it does so at a faster rate than for other values of $\alpha$ (e.g. $\alpha=1$)? Also, intuitively I imagine that having a better usage of the weights (i.e. model capacity) should lead to a better performance in terms of better training loss at comparable width/depth. Then why in Figure 11, 12 does the ODE limit have a similar training loss to the ($\alpha=1/2$) model (despite not having hyperparameter transfer?).

---

> ### Author Response · Authors · 2023-11-22
>
> Thanks for your valuable time and further comments! We provide our response below:
>
> > Then why in Figure 11, 12 does the ODE limit have a similar training loss to the ($\alpha=1/2$) model (despite not having hyperparameter transfer?).
>
> Recall the "real" block multiplier in the experiment is $aL^{-\alpha}$ and the "real" learning rate is $\eta L^{-\gamma}$, so no matter what $\alpha$ and $\gamma$ are chosen, for fixed $L$, one can always sweep $a,\eta$ extensively such that the training loss is minimal over all possible "real" block multiplier and "real" learning rate choice for this particular $L$. So directly comparing the minimal training loss for $L=2^{10}$ is not meaningful. Instead, we should fix $a,\eta$ for all $L$ by sweeping them extensively at some small $L$. In our experiments, we choose to find the optimal $a,\eta$ when $L=2^3$ (you can see that the $L=2^3$ curves in Fig. 11 and 12 have the same shape). Now back to Fig. 12, if we fix $\eta$ to be the optimal one at $L=2^3$, then the log of training loss at $L=2^{10}$ is around -3.1; while for $\alpha=\gamma=1/2$ in Fig. 11, the number is around -3.5. I hope this also convinces you that having a better usage of the weights (i.e. model capacity) should lead to better performance at least when $n\to\infty, L\to\infty$.
>
> > ... how does maximizing feature diversity imply that the optimal learning rate converges to an optimal in $L, N$ and it does so at a faster rate than for other values of $\alpha$ (e.g. $\alpha=1$)?
>
> For better illustration, we introduce function $f^L(\bar a, \bar\eta)$ that represents the training loss after $T$ steps of training with "real" block multiplier $\bar a$ and "real" learning rate $\bar\eta$ with depth $L$ and infinite width. Let $\bar a^{*L}, \bar\eta^{*L}\triangleq\arg\min_{\bar a, \bar\eta} f^L(\bar a,\bar\eta)$. (For simplicity, let us assume the minimum is unique.) Then we are essentially looking for how $\bar a^{*L}, \bar\eta^{*L}$ changes when $L$ grows, i.e., if $\bar a^{*L}=\Theta(L^{-\alpha}), \bar\eta^{*L}=\Theta(L^{-\gamma})$, then optimal hyperparameters converge (only) under $(\alpha,\gamma)$ parametrization.
>
> It is clear that $\bar a^{*L}$ must not be $\Theta(L^{-0.49})$. $\alpha=0.49$ makes the network unstable when $L\to\infty$, so $\lim_{L\to\infty} f^L(aL^{-0.49},\bar\eta)$ cannot be small for any $\bar\eta$.
>
> Similarly, if $\lim_{L\to\infty} f^L(aL^{-1/2}, \eta L^{-1/2}) <  \lim_{L\to\infty} f^L(a'L^{-1}, \eta')$ is always true, then $\bar a^{*L}, \bar\eta^{*L}$ cannot be $\Theta(L^{-1})$ and $\Theta(1)$.
>
> If the reviewer agrees that having a better usage of the weights (i.e. model capacity) should lead to better performance when $L\to\infty$, then we have $\lim_{L\to\infty} f^L(aL^{-1/2}, \eta L^{-1/2}) <  \lim_{L\to\infty} f^L(a'L^{-\alpha}, \eta'L^{-\gamma})$ holds true for any $(\alpha,\gamma)\neq (1/2,1/2)$. Thus, if $\bar a^{*L}=\Theta(L^{-\alpha}), \bar\eta^{*L}=\Theta(L^{-\gamma})$ for some $\alpha, \gamma$, then $(\alpha,\gamma)$ must be $(1/2,1/2)$. Of course, one can argue that $\bar a^{*L}$ and $\bar\eta^{*L}$ may not scale like real powers of $L$, but they can be rejected by a straightforward generalization of our claims in the paper.

---

> > ### Comment · Reviewer_5jxB · 2023-11-23
> > **Final Comment**
> >
> > I thank the authors for their time and insights. I agree with the experimental design of Figures 11 and 12. I am still not entirely convinced by the optimality of the proposed scaling due to the fact that there should have been a more thorough empirical evaluation (e.g. more datasets and tasks) of the fact that the training is always faster under the proposed optimal parametrization. This seems to be a crucial point for the argument: optimal scaling --> better loss --> faster convergence of optimal hyperparameters, and I would have appreciated a more convincing suite of experiments beyond Figures 11 and 12.  However, the major revision provides a significant improvement to the manuscript. Hence, I increased my score to 8, trusting that the authors will include the missing background citations (which are still missing in this revised version of the paper).

---

### Official Review · Reviewer_TGji · 2023-11-01

**Soundness:** 4 excellent
**Presentation:** 4 excellent
**Contribution:** 3 good
**Rating:** 8
**Confidence:** 4

**Summary:**

This paper investigates how to transfer optimal hyperparameters for a network with different depths. The authors use the tensor program framework to analyze the covariance of a linear resnet. The authors verified their theory on toy resnets.

**Strengths:**

The paper presents a solid theory for an important problem. Although from the reviewer's viewpoint, the question is far from being answered this paper makes a solid concrete first-step contribution.

**Weaknesses:**

- The resnet archtiecture is too simplifeid. The muP initialization can't work when the residual block is two-layer. How about the theory for linear resnet when the residual block is two-layer?
- the tensor program framework can deal with the activation function and normalization layer but this paper dropped the part there. Can even just analyze one step of gd？ (or linear net+noramlization layer)
- the convergence in the tensor program is just loss function convergence and can't obtain the convergence of optimal hyperparameters. Can the author show that the optimal hyperparameter have faster convergence speed?

**Questions:**

See above

---

> ### Author Response · Authors · 2023-11-19
>
> We would like to thank the reviewer for encouraging review and thoughtful questions! Please see our response below.
>
> **The resnet archtiecture is too simplifeid. The muP initialization can't work when the residual block is two-layer. How about the theory for linear resnet when the residual block is two-layer?**
>
> We thank the reviewer for pointing this out! This is indeed an important question. We have added discussions about using two-layer residual blocks in Appendix K, where we provide both theoretical and empirical arguments as to why the optimal parametrization of the general architecture is impossible with the current setup. Our argument in Appendix K.3 indeed uses the linear resnet as an example.
>
> **the tensor program framework can deal with the activation function and normalization layer but this paper dropped the part there. Can even just analyze one step of gd？ (or linear net+noramlization layer)**
>
> In our general case (see Sec. 5 and 6), the activation function and mean subtraction are included. Replacing mean subtraction with the normalization layer will not change our results because normalization does not change the scaling of the activation for any stable parametrization. To avoid unwanted complexity, we decided not to include it in the paper for better presentation. Empirically, we verify that adding normalization layers still exhibits hyperparameter transfer under Depth-$\mu$P (see Appendix L.2).
>
> **the convergence in the tensor program is just loss function convergence and can't obtain the convergence of optimal hyperparameters. Can the author show that the optimal hyperparameter have faster convergence speed?**
>
> We have included a section on “What causes hyperparameter transfer?” in Appendix I. By classifying all the parametrizations, we found Depth-$\mu$P to be the unique optimal parametrization in the sense that it guarantees both feature learning and maximal feature diversity, and that is why we claim the convergence of optimal hyperparameters.
>
> The convergence speed of the optimal hyperparameter indeed decides whether hyperparameter transfer is useful in practice, but is almost impossible to prove or estimate without empirical evaluation.

---

### Official Review · Reviewer_f9Ax · 2023-11-02

**Soundness:** 2 fair
**Presentation:** 3 good
**Contribution:** 2 fair
**Rating:** 5
**Confidence:** 3

**Summary:**

The paper addresses the problem of scaling the depth of neural networks and its impact on model performance especially the norm of the feature will blow up as depth grows. The authors introduce a principled approach called Depth-µP, which allows for training of arbitrarily deep networks while maximizing feature learning and diversity among layers. They propose dividing the contribution of each residual block and parameter update by the square root of the depth, ensuring more stable training and hyperparameter transferability.

**Strengths:**

1. The paper introduces a novel depth scaling strategy called Depth-µP, which addresses the challenges of increasing the depth of neural networks. This method is designed for a specific network architecture where the feature output of each residual block is divided by a hyperparameter L to control the norm, not blowing up.
2. The authors establish a theoretical foundation for Depth-µP by using Tensor Programs to rigorously analyze the behavior of infinitely deep neural networks under the proposed scaling scheme.

**Weaknesses:**

1. The motivation described in the introduction `The stacking of many residual blocks causes an obvious issue even at the initialization — the norm of $x^l$ grows with $l$,...` does not match the real design of the neural network. As there will be a normalization layer after or before the activation layers in common resnet or transformers.

2. The Depth-$\mu$P algorithm is designed on a modified structure of the residual network, but there is not any performance comparison of this structure and commonly used structure.
Even though with such a modified structure we can train networks with an arbitrary depth, it does not indicate the performance is better.

3. It is not so easy for me to follow the theoretical analysis. The introduction of the $\Gamma$ and $C$ functions is not smooth. And why the analysis needs the width of the network to go to infinity is still not clear.

**Questions:**

1. It would be better if the author could provide some experimental results on the performance of the modified structure.
Otherwise, the analysis will be too specific but not related to any real applications.

2. When analyzing the infinite-depth effect, I think it will be better the separate the infinite-width effect.
However the current analysis seems to combine these two cases, and difficult to tell the contribution of the analysis of the infinite depth.

---

> ### Author Response · Authors · 2023-11-19
>
> We thank the reviewer for the detailed review. We provide our responses to the comments and questions below.
>
> **“The stacking of many residual blocks causes an obvious issue even at the initialization — the norm of $x^l$ grows with $l$,...” does not match the real design of the neural network. As there will be a normalization layer …**
>
> We thank the reviewer for pointing this out. We will include the following discussion in the final paper.
>
> Firstly, this motivation still stands for the pre-LN architecture, where the normalization is applied *inside* the residual branch. In particular, if we consider the following residual network $x^l=x^{l-1}+W^{l2}ReLU(W^{l1} LN(x^{l-1}))$, then when the width $n$ goes to infinity, we have $||x^l||_2^2=||x^{l-1}||_2^2 + 1/2$ at initialization since inner product between $x^{l-1}$ and $W^{l2}ReLU(W^{l1} LN(x^{l-1}))$ is zero and $||W^{l2}ReLU(W^{l1} x)||_2^2 =1/2$ for any $x$ with $||x||_2=1$. So with pre-LN, the norm of $x^l$ still grows with $l$, but with a slower speed.
>
> Then one might ask the follow-up question: since people apply LN on $x^L$ in practice, would that solve the problem? The answer is still “no”. In particular, at the initialization of the above residual network, the last-layer features of any two different inputs are *always independent* (i.e., the inner product of the last-layer features is always zero) when $n, L\to \infty$. It means the network is not aware of any correlation between the inputs! This observation is followed by a simple calculation: let $x_1^{l}$ and $x_2^l$ be the outputs of $l$-th layer for two inputs respectively, then $\langle x_1^{l}, x_2^l\rangle=\langle x_1^{l-1}, x_2^{l-1}\rangle+\rho^{l-1}(\pi-arccos(\rho^{l-1}))/2\pi$ where $\rho^{l-1}=\frac{\langle x_1^{l-1}, x_2^{l-1}\rangle}{||x_1^{l-1}||_2||x_2^{l-1}||_2}$. One can easily see that $\rho^{l}$ decreases when $l$ grows, and $\rho$ converges to $0$ in the end.
>
> A similar calculation can be applied to post-LN architectures as well, and lead to the same observation on the independence between last-layer features.
>
> **No performance comparison between this structure and commonly used structure.**
>
> We are not sure what the reviewer means by “modified structure”. Does it refer to adding the branch multiplier? Or $g$ contains only one layer?
>
> We aim to deliver the theory of depthwise scaling with block depth 1 in this paper, so we care more about the performance comparison between different scaling strategies. Our experiments serve as a verification of the theory and we do not expect it to perform exactly the same on convnets or transformers where the block depth is not 1. (We have included a discussion about using deeper blocks in Appendix K, and provided experiments on Megatron trained on Common Crawl in Appendix L.3.)
>
> **It is not so easy for me to follow the theoretical analysis…why the analysis needs the width of the network to go to infinity is still not clear.**
>
> We have added more intuition to $\Gamma$ and $C$ in Sec. 4.
>
> Our analysis relies on Tensor Programs, which helps us better characterize the training dynamics, but it requires taking the width to infinity first. It is believable that our results hold even when the width is not very large compared to the depth, especially because the width is sometimes smaller than the depth in our experiments. However, it is difficult to prove without better theoretical tools.

---

> ### Author Response · Authors · 2023-11-21
> **Further comments before the end of author response period?**
>
> Thanks again for your valuable time in reviewing our paper!
>
> Since the discussion period is coming to an end, we would like to know if our response addresses your main concerns. We are happy to address any further comments!

---

### Official Review · Reviewer_Mvpr · 2023-11-02

**Soundness:** 4 excellent
**Presentation:** 3 good
**Contribution:** 3 good
**Rating:** 6
**Confidence:** 4

**Summary:**

This work proposes a desirable scaling of hyper-parameters to achieve feature learning in networks with skip connections, infinite width, and infinite depth. In other words, it is about $\mu$P for extremely deep networks with skip connections. First, the authors introduce two scaling factors depending on the depth — $\alpha$ for the forward propagation and $\gamma$ for the gradient. These factors are key to enabling feature learning in such deep networks. Second, the authors focus on a deep linear network and obtain an analytical expression for feature learning under the proposed scaling factors. Finally, the effectiveness of these scaling factors is empirically tested through some experiments,  particularly those examining the transferability of learning rates.

**Strengths:**

-  While some previous work proposed to use $\alpha=1/2$ for the skip connections,  the scale of gradient updates for such networks has not been investigated. This study addresses this gap by demonstrating that $\gamma=1/2$ is appropriate for feature learning in networks with infinite depth.

- This study keeps up with the latest research on muP, particularly highlighting Yang & Littwin 2023 for entry-wise adaptive gradients, and further increases the usefulness of muP.

**Weaknesses:**

-**Not a few ambiguous points**

As I have noted in questions (i)-(v), there are numerous unclear aspects in this paper that are critical to understanding its main claim. In particular, the dependencies among theoretical claims are ambiguous, making it uncertain how Main Claim 6.5 is derived, as highlighted in my Questions (i) and (ii).

-**Empirical verification is limited**

Figure 2 is the only evidence that the proposed Depth-$\mu$P works for non-linear networks because Figures 3 & 4 suppose linear networks and Figures 5-9 are part of Figure 2. However, the activation function used in Figure 2 is not mentioned.  In addition, it is also unclear whether the training is finished (Question (iii)). Thus, it is hard to judge whether the Depth-$\mu$P works in realistic situations. Because a large part of this work focuses on the linear network, there is a concern that it may only function effectively in models close to the linear network.

One more concern is that the width is not so large (n=256) compared to the depth.  The theoretical framework of this study considers a scenario where depth << width. However, the experimental conditions seem closer to a proportional limit where depth ~ width. It is surprising that the theory and experiments align despite this discrepancy. It would be advisable to validate this with several types of widths, activation functions and datasets (or tasks).

**Questions:**

(i) Where in the paper do the authors utilize the tensor program formulation to derive the equation $\alpha+\gamma=1$? It appears that the dynamics of linear networks discussed in Sections C & D presuppose $\alpha=\gamma=1/2$. Furthermore, the general case presented in Section E merely offers a tensor program representation of the training process without shedding light on how we should determine ($\alpha$, $\gamma$).

I hypothesize that the assessment of $A_l^2$ in Section 3 leads to the determination of $\alpha = \gamma = 1/2$. However, I am unable to find this evaluation in Sections C, D and E.  Could you please elucidate precisely where in these sections the authors derive both $\alpha + \gamma = 1$ and $\alpha = \gamma = 1/2$, and explain how the tensor program formulation is applied to achieve these conclusions?

(ii) Feature learning is defined as $\Delta \boldsymbol{h}_t^{\lfloor\lambda L\rfloor}=\Theta(1)$ (Definition 6.5). Contrarily, the authors derive the unique parameterization for feature learning (Claim 6.5) from the non-redundant exponent $\kappa=1/2$. However, the manuscript does not clearly show the connection between definition 6.5 and $\kappa$. Could you write down explicitly how these two quantities are equivalent to each other? Furthermore, why and in what manner is the case of $\kappa=1/2$ *unique* for enabling feature learning outlined in Definition 6.5?

(iii) What is the activation function utilized in Figure 2? Additionally, what logarithm base is employed for the log loss depicted on the vertical axis? These may appear as minor details, but I believe they are crucial.  Since the analysis in Section 4 and Figures 3 & 4 focus on the linear network, it seems no guarantee that the Depth-$\mu$P works for general nonlinear activation functions.  For instance, could you empirically verify whether the results obtained are applicable to Tanh and ReLU activation functions?

Regarding the scale of the vertical axis, the issue lies with the right side of Figure 2.  If the base is $e$,  the model exhibits Loss ~ exp(-3.5) ~ 0.03. This implies that the training is not finished. Consequently, it becomes difficult to judge whether the learning rate transfer is successful in the models that have been completely trained.

(iv) Difference from Jelassi et al. (2023)

> Jelassi et al. (2023) showed that a learning rate scaling of depth−3/2 guarantees stability after the initial gradient step

The distinction between the current work and Jelassi et al. (2023) should be more explicitly stated.  I think that $\mu$P in the current work is also consistent with the initial gradient step. I mean, the $\mu$P derived from the initial gradient step would remain consistent across general t steps. Are you suggesting that your Depth-$\mu$P differs between the initial step and subsequent steps where $t > 1$?

(v) In section G.5., the authors say
>In our setup, where the input and output layers are initialized at a constant scale (w.r.t. L), it is actually not possible to have a kernel limit. Even in our linear case in Section 4, one can see the learned model is not linear.

I am confused about this explanation and believe a more detailed and comprehensive elucidation is necessary.  Firstly, are you implying that the Depth-$\mu$P is the unique parameterization that is stable, nontrivial, and faithful? Specifically, in very deep networks with skip connections, are you suggesting that the kernel regime is absent?  As far as I understand, the original work on $\mu$P [Yang & Hu, 2021] characterizes feature learning as the boundaries of the kernel regime. Thus, it would be surprising (and interesting, if it is true) that the kernel regime disappears, leaving only feature learning as a stable state of training.
Secondly, in what sense, is the learned linear network not linear? By definition, the linear network is a linear mapping of the input x.  Could you clarify in what aspect the learned linear network deviates from this linear characteristic?

Due to these ambiguous points, I have currently assigned a lower score. However, I am open to increasing this score if they are clarified or if any misunderstandings on my part are resolved.

---

> ### Author Response · Authors · 2023-11-19
> **Major Comment by Authors**
>
> We thank the reviewer for the detailed review. We believe there are some misunderstandings that we would like to clarify first:
> - Depth-$\mu$P is unique because it not only exhibits feature learning but also maximizes feature diversity. All parametrizations with $\alpha+\gamma=1$ and $\alpha> 1/2$ also enable feature learning, but they lack feature diversity (Claim 6.4). We have emphasized this in the paper and added more discussion about feature diversity in Appendix J.
> - The purpose of this work is mostly theoretical and focuses on the optimization side. The experiments on CIFAR-10 serve as a verification of the theory and we do not expect one can find perfect hyperparameter transfer for transformers where the block depth is not 1. (We have included a discussion about using deeper blocks in Appendix K, and provided experiments on Megatron trained on Common Crawl in Appendix L.3.) We neither expect hyperparameter transfer for the late stage of training on image datasets where the noise caused by SGD is prominent. (This is also true for the original $\mu$P.)
> - Sections 3, 4, C, and D in the paper serve as a warmup for readers to understand why Depth-$\mu$P is a good scaling strategy, i.e., why it is stable, why it exhibits feature learning, and how to derive its training dynamics in the limit. We compare Depth-$\mu$P with other parametrizations in Section 6.

---

> > ### Author Response · Authors · 2023-11-19
> >
> > We thank the reviewer for the detailed comments again and hope our responses below clear all the confusion.
> >
> > **(i) Could you please elucidate precisely where in these sections the authors derive both $\alpha+\gamma=1$ and $\alpha=\gamma=1/2$, and explain how the tensor program formulation is applied to achieve these conclusions?**
> >
> > In Section G.1 to G.3, we derive the limiting dynamics of Tensor Programs for $\alpha=1/2, 1$ or in between. Based on these, we provide our justifications for the claims in Section G.4. We have also included proof of all claims in the linear case (see Appendix H of the revised version), where the notation is simpler and easy to follow without too much knowledge of Tensor Programs.
> >
> > **(ii) … However, the manuscript does not clearly show the connection between definition 6.5 and $\kappa$. Could you write down explicitly how these two quantities are equivalent to each other?**
> >
> > As we mentioned in our major comments, these two are not equivalent. Feature learning is easier to achieve, and Depth-$\mu$P ($\alpha=\gamma=1/2$) is optimal in the sense that it is the only one that achieves both feature learning and maximal feature diversity. We hope this clarifies the misunderstanding.
> >
> > **(iii) What is the activation function utilized in Figure 2? Additionally, what logarithm base is employed for the log loss depicted on the vertical axis? … This implies that the training is not finished.**
> >
> > We apologize for the missing specification of the activation function. The activation is indeed ReLU. In our discussion about feature diversity (Appendix J in the revision), we predicted absolute value activation helps feature diversity, and verified it empirically as well.
> >
> > The base of the log is $e$ (natural logarithm). As we mentioned in the major comments, the CIFAR-10 experiments serve as a testbed for optimization speed. The ideal application of our theory (and the original $mu$P) is training language models where the primary goal is minimizing the training loss as quickly as possible. We have included training Megatron transformer on Common Crawl in Appendix L.3.
> >
> > **(iv) Difference from Jelassi et al. (2023)**
> >
> > We believe there is a misunderstanding on this point. Jelassi et al. (2023) derive scaling rules for *MLPs* with no skip connections. We consider residual networks. Therefore, the models and the scalings are fundamentally different.
> >
> > **(v) Explain “In our setup, …, it is actually not possible to have a kernel limit. … the learned model is not linear”.**
> >
> > As we clearly mentioned in Sec 5.2, we assume the widthwise scaling follows $\mu$P in our setup. Our claim here is still under this assumption. We have emphasized this in our revision.
> >
> > If the widthwise scaling is in the kernel regime, although it is not the interest of this paper, one can imagine the infinite-width-then-depth limit being characterized by the evolution of NTK when depth grows.
> >
> > For the linear case, we would like to emphasize that in the context of the kernel regime, linearization is w.r.t. parameters instead of the input. So the linear residual net is not linear. We have also emphasized this in our revision.
> >
> > **Concerns that width is not so large in experiments.**
> >
> > Although our theory focuses on the infinite-width-then-depth limit, it is reasonable to believe that all the limits (including the proportional limit) are the same. Indeed, we expect that the infinite width and depth limits *commute*, in the sense that no matter how you take that limit, the result is the same. For instance, you get the same limiting dynamics if you take the infinite-width-then-depth limit or the proportional limit. This was proven rigorously *at initialization*  in Hayou and Yang (2023).  However, proving the same result during training is challenging and we are currently investigating this for future work.
> >
> >
> >
> > Due to a limited computation budget, we are not able to train with a larger width. However, based on the experimental results, hyperparameter transfers without a very large width, which supports the above hypothesis. One should expect a better result with a larger width.

---

> ### Author Response · Authors · 2023-11-21
> **Further comments before the end of author response period?**
>
> Thanks again for your valuable time in reviewing our paper!
>
> Since the discussion period is coming to an end, we would like to know if our response addresses your main concerns. We are happy to address any further comments!

---

> ### Comment · Reviewer_Mvpr · 2023-11-22
> **Reply to the authors' comment**
>
> Thank you for your thorough reply. It has largely cleared up my concerns and confusion, and I believe that the readability has greatly improved. I have raised my score to an acceptance side. Since the options are only 6 and 8, I chose 6, but if allowed, I would prefer a middle evaluation 7.
>
> There are still some parts that concern me slightly, although none are critical flaws. They just seem preferable to furhter enhance the significance of this study.
>
> **On question (i)**
>
> My understanding is that the derivation of Depth-$\mu$P is based on the linear activation case.  As the authors remarked, for non-linear cases (the general case), the derivation remains heuristic. I mean, it seems quite hard to derive the Depth-$\mu$P (shown in Section G.4) from only watching the tensor program lines of the general case (shown in Section G.1-G3). Certainly, in the revised version, the authors have added Section H and clarified that the tensor program lines certainly give the correct Depth-$\mu$P for the linear case. However, without such an example, it seems not easy to evaluate the concrete values of ($\alpha, \gamma$). From a theoretical standpoint, readers may wonder about any potential discrepancy arising from non-linearity.
>
> **On concerns that width is not so large in experiments**
>
> Initially, I was concerned about the lack of experimental verification for Depth-$\mu$P, but the revised manuscript now includes both relu nets and transformers, significantly enhancing the theory's validity. My only minor dissatisfaction concerns how closely a finite width (=256) approximates the infinite limit. While the argument about *limits commute* is persuasive, there's still some uncertainty about whether the success of Depth-$\mu$P in such a finite setting is coincidental or inherent. Therefore, a more explicit demonstration of width dependence would be advantageous. I look forward to possibly seeing it after the decision.

---

> > ### Author Response · Authors · 2023-11-22
> >
> > We would like to thank the reviewer for reading our revision and raising the score!
> >
> > We are sure that Sec. G.1-G.3 provides enough information to derive Depth-$\mu$P, although we admit that the current presentation for the general case is not very friendly for readers who are not familiar with Tensor Programs. We did the best we could to give a brief overview of the Tensor Program framework. However, understanding the framework may require a lot more (potentially reading the Tensor Programs IVb paper). We will add another section about any potential discrepancy arising from non-linearity.
> >
> > For the finite width (=256) in experiments, we kindly argue that a small width will only introduce noise that may potentially kill our theory, so experiments showing hyperparameters still transfer with a small width means the noise introduced by the small width does not affect the results, which actually strengthens our theory. It is hard to imagine the other scenario where our theory is only true when the width is small. Nevertheless, we will try our best to include experiments with larger widths in the final paper if the computation budget allows.

---

### Official Review · Reviewer_AGJB · 2023-11-04

**Soundness:** 3 good
**Presentation:** 3 good
**Contribution:** 4 excellent
**Rating:** 8
**Confidence:** 4

**Summary:**

The paper studies feature learning parameterizations of deep residual networks when depth goes to infinity (after taking infinite width in a muP regime), and studies the impact of two scaling parameters: a multiplier $L^{-\alpha}$ of each block before adding on the residual stream, and a multiplier $L^{-\gamma}$ on the learning rate, where $L$ is the depth.

The authors find that the best regime, termed Depth-muP, is for $\alpha = \gamma = 1/2$, leading to various desirable properties. Of particular interest and novelty is the "feature diversity" property, which leads to diverse features across neighboring layers and is only achieved for $\alpha = 1/2$.

**Strengths:**

The findings in the paper are of significant interest for scaling of neural networks to large depths, both for theoreticians and practitioners.
While the multiplier scalings had been studied before, especially in kernel regimes, its extension to feature learning is crucially important, and provides a meaningful criterion for the benefits of $\alpha = 1/2$, namely feature diversity (while higher $\alpha$ leads to redundancy across layers).

**Weaknesses:**

see questions

**Questions:**

Here are a few points that should be addressed in order to strengthen the paper.

* feature diversity: it'd be helpful to provide additional experiments on this point, especially for $\alpha > 1/2$, something which seems to be missing in the current draft. It'd be helpful to provide more intuition about the benefits of feature diversity (which seem closely related to benefits of depth), perhaps with concrete examples. Can the quantity in Def 6.6 be plotted during training to see the effect of $\alpha$?

* presentation: while the warmup section 3 is quite insightful, I found section 4 to be much more obscure, and perhaps not essential for the later discussions. I encourage the authors to either provide some more intuition on the "physical" meaning of the different equations, or to shorten the section and defer details to the appendix. Perhaps an informal version of the actual tensor program would be more insightful? Also, how does this related to the AMP and DMFT literature? some more comparison to related literature would be helpful (in particular, this concurrent [paper](https://arxiv.org/abs/2309.16620) seems highly related and should be cited and compared to in the final version)

* mean subtraction: could you clarify a bit more the role of this? Is MS specifically needed because of the activation? Would adding an MLP output layer as in transformers drop this requirement? Note that in practice LLM practitioners often drop the mean subtraction part of layer-norm, using RMSnorm instead (e.g. in LLaMa, likely for computational reasons, but it seems to convey that MS isn't really needed?)

* scaling: do you expect any different behaviors in different limits, e.g. in the proportional deep-wide limit studied e.g. by Hanin, or what do you expect would change if the number of GD steps can grow with the number of layers? any comparison or intuition here would be useful.

Other comments:
- on feature diversity: how does the quantity in Def 6.6 behave at initialization vs later in training? what do you mean by "maximal" in claim 6.5? any more intuition beyond the footnote on how to get smaller exponents than 1/2?
- "analogous to width situation where deep mean field collapses to a single neuron": can you elaborate? what is deep mean field, and how are these related?
- Figures: any observations on what happens when changing width and depth together, as opposed to fixing the width?

---

> ### Author Response · Authors · 2023-11-19
>
> We thank the reviewer for the thoughtful and encouraging review! We provide our response below.
>
> **Additional feature diversity experiments?**
>
> We have added more discussions on feature diversity in Appendix J, and corresponding experiments in Appendix L.4. Comparison between Fig. 11 and 12 demonstrates the empirical advantage of Depth-$\mu$P ($\alpha=\gamma=1/2$) over the so-called ODE parametrization ($\alpha=1, \gamma=0$). Fig. 18 and 19 show their feature diversity difference. Depth-$\mu$P with higher feature diversity performs better.
>
> **… section 4 … I encourage the authors to either provide some more intuition on the "physical" meaning of the different equations, or to shorten the section and defer details to the appendix. …**
>
> Thanks for the suggestion. We have shortened Sec. 4 and added more details of the technical road map.
>
> **Also, how does this related to the AMP and DMFT literature? some more comparison to related literature would be helpful (in particular, this concurrent paper seems highly related and should be cited and compared to in the final version)**
>
> Thank you for pointing out this paper, which was also recently submitted to ICLR. We have indeed been in touch with the authors of that paper, and we intend to include a comparison with that work in the final version. In short, our work introduces the notion of feature diversity and we argue that it is essential in obtaining hyperparameter transfer (optimal limit), this was not discussed in the reference mentioned above. Secondly, regarding optimal parameterization with Adam, it seems that our conclusions are in contradiction with the conclusions of that work: Depth-muP suggests that both block multiplier and learning rate should scale as $1/\sqrt{depth}$, while in that work, they suggest only scaling the blocks. In our experiments with depths up to 2^10, we showed that Depth-muP yields hyperparameter transfer with Adam.
>
> **mean subtraction: could you clarify a bit more the role of this? Is MS specifically needed because of the activation? Would adding an MLP output layer as in transformers drop this requirement? Note that in practice LLM practitioners often drop the mean subtraction part of layer-norm, using RMSnorm instead**
>
> Yes, as we mentioned in the last paragraph of Section 5, MS is needed due to the activation, which leads to the exploding behavior of intermediate features because of the non-zero mean of the block outputs at the initialization. Therefore, adding an MLP output layer in the residual blocks can solve the problem as it makes the features zero-mean, so it is not surprising that dropping MS in transformers works in practice. However, the analysis in this paper does not trivially apply to 2-layer blocks. We have included a discussion of 2-layer blocks in Appendix K and experiments in Appendix L.3.
>
> **scaling: do you expect any different behaviors in different limits, e.g. in the proportional deep-wide limit studied e.g. by Hanin, or what do you expect would change if the number of GD steps can grow with the number of layers? any comparison or intuition here would be useful.**
>
> This is an interesting question. Actually, we expect (and can give heuristic arguments) that the infinite width and depth limits *commute*, in the sense that no matter how you take that limit, the result is the same. For instance, you get the same limiting dynamics if you take infinite-width-then-depth limit or the proportional limit. This was proven rigorously *at initialization*  in Hayou and Yang (2023).  However, proving the same result during training is challenging and we are currently investigating this for future work.
>
> **on feature diversity: how does the quantity in Def 6.6 behave at initialization vs later in training? what do you mean by "maximal" in claim 6.5? any more intuition beyond the footnote on how to get smaller exponents than 1/2?**
>
> The behavior does not change during training. $\kappa=1/2$ is maximal for all stable and non-trivial parametrization. Specifically, combining Claim 6.4 and Claim 6.5, there are only two choices of $\kappa$: $0$ for $\alpha\in(1/2,1]$, and $1/2$ for $\alpha=1/2$. Regarding exponents smaller than 1/2, this is closely related to rough path theory in stochastic calculus. Consider Depth-muP parameterization: the weights are generally initialized to be iid, and therefore when this noise adds up (because of the skip connection), it yields something similar to a Brownian motion, which has a continuity exponent of 1/2. This exponent can be further reduced by *correlating the weights of different layers* in which case we obtain something similar to a *fractional Brownian motion* which can have a general continuity exponent $H \in (0,1)$, depending on the correlation level.

---

### Author Response · Authors · 2023-11-19
**Major revision of the paper**

We thank all reviewers for their thoughtful reviews! We have made revisions to our paper according to the reviewers’ comments. We mark the major changes in red, and list the newly added contents below:
- Appendix H gives proof of our major claim in the linear case. We hope it provides more intuition of our theory.
- Appendix I gives a thorough discussion of what causes hyperparameter transfer. It illustrates the necessity of identifying the unique optimal scaling strategy, which is accomplished by the concept of feature diversity in this paper.
- Appendix J gives extended applications of feature diversity, which again sheds light on the importance of feature diversity. We show that the choice of nonlinearity and placement of nonlinearity can affect feature diversity, and indeed the best architecture that maximizes feature diversity also performs better in the experiments.
- Appendix K gives a comprehensive discussion on the optimal scaling of residual networks containing the block depth 2, and provides an impossibility result.
- Appendix L.2 further includes experiments
    (a) showing the necessity of maximal feature diversity to hyperparameter transfer, and
    (b) verifying hyperparameter transfer under Depth-$\mu$P generalizes to residual networks with layer normalization (LN).
- Appendix L.3 provides experiments of Megatron transformer trained on Common Crawl. The results show that Depth-$\mu$P improves hyperparameter transfer but is not the ultimate solution for scaling the depth of transformers.
- Appendix L.4 empirically verifies our claims about feature diversity exponent.

---

### Meta-Review · Area_Chair_8ASU · 2023-12-08

**Metareview:**

This paper introduces a novel approach for scaling the depth of neural networks, called Depth-P, which allows for the training of arbitrarily deep architectures while maximizing feature learning and diversity among nearby layers. The proposed method involves adjusting the contribution of each residual block and the parameter update by the square root of the depth, a strategy that ensures more stable training for deep neural networks. A significant theoretical finding in this paper is the establishment, via Tensor Programs, of a limit for infinitely deep and wide neural networks under this proposed scaling scheme. Importantly, this scaling strategy guarantees the transferability of hyperparameters from shallow to deep models.

**Justification For Why Not Higher Score:**

The main concerns from reviewers are about oversimplified architecture of resnet in analysis compared with practice, missing citations on mean-field literature of infinite width networks, and discussions supporting their claims.

**Justification For Why Not Lower Score:**

The novelty and significance of these findings offer a principled approach to scaling depth, mitigating issues associated with model performance degradation in excessively deep networks.  The paper could be accepted based on the condition that authors addressed those points raised by reviewers in discussions, including a more comprehensive background citations.

---

### Decision · Program_Chairs · 2024-01-16

Accept (poster)